# $D^3PM$: Diffusion Model Responds to the Duty Call from Causal Discovery

## Abstract

Causal discovery (CD) involves inferring cause-and-effect relationships as directed acyclic graphs (DAGs). In this work, we assume that the data is generated by an additive noise model (ANM). Recent work has formulated the problem as a continuous optimization problem, which consists of solving an inverse problem and satisfying an acyclicity constraint. However, solving the inverse problem in CD is often unstable, i.e. high sensitivity of the effects to perturbations in the causes. To address this instability, we formulate the inverse problem as a regularized optimization scheme and propose a novel variation-negotiation regularizer. Compared to traditional regularization techniques for the continuous optimization problem, e.g. $\ell_1$ penalty on graphs, the proposed regularizer exploits the variation variable in ANMs to stabilize the solutions (i.e. DAGs). This regularizer is advantageous as it does not rely on any hypotheses, such as graph sparsity, about true DAGs. The variation-negotiation regularizer regulates the DAG purely based on observed data.

Building on the proposed regularizer, a series of improvements to the regularized optimization scheme reveal the connections between solving the regularized optimization problem and learning a diffusion model, as they share comparable objective functions. This insight leads us to develop an equivalent diffusion model called DAG-invariant Denoising Diffusion Probabilistic Model. Extensive empirical experiments on synthetic and real datasets demonstrate that the proposed diffusion model achieves outstanding performance on all datasets.

## 1 Introduction

Identifying cause-and-effect relationships among variables is a challenging problem in various scientific fields such as economics (Hoover, 2017), biology (Sachs et al., 2005), and climate science (Zhang et al., 2011). Cause-and-effect relations can be represented as directed acyclic graphs (DAGs), where nodes are variables, and directed edges indicate direct causal effects. The objective of causal discovery (CD) is to recover DAGs from observed data. In this work, we assume the observational data follow additive noise models (ANMs), meaning each variable is defined as a function over a subset of the remaining variables, which are represented by a DAG, plus an unexplained variation variable[1] (Hoyer et al., 2008).

Traditional methods search the DAG space by testing conditional independence between variables (Spirtes et al., 2001) or by optimizing some goodness of fit measure (Chickering, 2002). A main challenge of these methods is that searching for true DAGs is extremely time-consuming (Chickering, 1996). To address it, Zheng et al. (2018) formulates the DAG search as a continuous optimization over the space of all graph adjacency matrices. The optimization objective comprises two parts: solving an inverse problem, where, given observational data, an adjacency matrix is solved according to ANMs, and satisfying an acyclicity constraint on the matrix. However, while promising, continuous optimization-based approaches struggle to combat instability in solving the inverse problem. The instability of an inverse problem refers to the high sensitivity of the effects to perturbations in the causes (Calvetti & Somersalo, 2018).

---

[1]In other work, they prefer referring to the unexplained variation as noise. However, in our work, we will introduce other noises later. To eliminate the ambiguity, following the naming system (Manzour et al., 2021), we use the notion of unexplained variation.

In this paper, we investigate how to trade the unstable inverse problem in CD with a relatively stable one using a regularization technique. We start by formulating the inverse problem as a regularized optimization problem that consists of a data consistency (recovery) term and a regularization term. Then, a novel **variation-negotiation regularizer** is proposed as the regularization term. Differing from previous regularization techniques for CD that explore the characteristics of DAGs, e.g. $\ell_1$ penalty on graphs (Zheng et al., 2018; Nazaret et al., 2024), the proposed regularizer alternatively exploits the unexplained variation variable in ANMs. This variation variable can be represented in terms of DAGs according to ANMs, so estimating the variation variable is equivalent to regularizing the solution (i.e., DAGs). We then use denoising techniques (Vincent et al., 2008) to estimate the value of the variation variable through a negotiation strategy. Regularizing DAGs through the variation variable has two main benefits. Firstly, the regularizer does not depend on any general hypothesis about true DAGs, such as the belief that real-life causal graphs are sparse. Instead, the variation-negotiation regularizer solely regulates DAGs based on observed data. Secondly, it paves the way for the connection between CD and diffusion models. With the proposed variation-negotiation regularizer, the regularized optimization objective can be reinterpreted as a single variation consistency (recovery) term without any regularization term. To probe the variation from diverse observations, we extend the single variation consistency term to multiple variation consistency terms by imposing diversified noise. With this extension, we find that **solving the proposed regularized optimization problem and training a Denoising Diffusion Probabilistic Model (DDPM) share comparable objective functions**.

The discovery motivates us to study diffusion models, such as DDPMs, which have recently emerged as powerful generative models (Cao et al., 2024). They use a sequence of probabilistic distributions to corrupt data in the forward process and learn a sequence of probabilistic models to reverse the forward process (Song et al., 2021). Although DDPMs achieve breakthrough performance in data generation, to our knowledge, only one work has studied applying DDPMs in CD tasks (Sanchez et al., 2023), where a diffusion model is used as a parameterized density estimator to replace a kernel-based estimation model in a CD algorithm (Rolland et al., 2022). Unlike this simple application of diffusion models, our work aims to explore the intrinsic relation between CD and diffusion models. **By posing the notion of DAG-invariance, where true DAGs remain invariant with any alteration to their corresponding observational data, we propose a diffusion model called DAG-invariant Denoising Diffusion Probabilistic Model ($D^3PM$), whose training objective is shown to be equivalent to the proposed regularized optimization objective**. In other words, $D^3PM$s are coined to respond to the duty call from continuous optimization-based CD approaches which suffer from instability.

We conducted a series of empirical studies to **evaluate the performance of $D^3PM$ on** 1040 **synthetic datasets with up to** 5000 **variables** and real-world datasets. The results demonstrate the superiority of $D^3PM$ over all baselines with reasonable training costs. The code is publicly available at https://anonymous.4open.science/r/D-3PM-07D1.

## 2 PRELIMINARIES

Here, we briefly review the prior knowledge about CD and DDPMs, respectively.

### 2.1 CAUSAL DISCOVERY

The CD problem is formally defined as follows: let $\mathbf{X} \in \mathbb{R}^{n \times d}$ be a data matrix representing $n$ i.i.d. observations of $d$ random variables. Let $\mathbb{G}$ be a space composed of DAGs with $d$ vertices and some directed edges. A DAG can be represented as a binary adjacency matrix. The goal of CD is, given $\mathbf{X}$, to derive a DAG $\mathcal{G} \in \mathbb{G}$ associated with the random variables, without access to ground truth DAGs (Koller & Friedman, 2009; Spirtes et al., 2001).

In this work, we focus on causal structure learning under ANMs:

$$\mathbf{X} \coloneqq \boldsymbol{f}(\mathbf{X}\boldsymbol{A}) + \mathbf{Z}, \tag{1}$$

where $\boldsymbol{f}$ is an arbitrary unknown function, and $\mathbf{Z}$ represents an $n \times d$ unexplained variation matrix. Here, $\mathbf{Z}$ is formulated as a random variable sampled from a distribution, but the distribution is unknown during the learning of $\boldsymbol{A}$.

## 2.2 DENOISING DIFFUSION PROBABILISTIC MODELS

DDPMs (Sohl-Dickstein et al., 2015; Ho et al., 2020; Song et al., 2021) follow a generative modelling paradigm that aims to approximate the target distribution $p_{\boldsymbol{\theta}}(\mathbf{X}_0) = \int p_{\boldsymbol{\theta}}(\mathbf{X}_{0:T})d\mathbf{X}_{1:T}$, where $\mathbf{X}_t$, $t = 1, ..., T$ are latent variables with identical dimensionality, given original data $\mathbf{X}_0 \sim q(\mathbf{X}_0)$. DDPMs consist of two steps: the forward Markov process and the reverse Markov process. The forward process gradually adds Gaussian noise to the data according to a variance schedule $\beta_1, ..., \beta_T$:

$$q(\mathbf{X}_{1:T}|\mathbf{X}_0) := \prod_{t=1}^{T} q(\mathbf{X}_t|\mathbf{X}_{t-1}), \qquad q(\mathbf{X}_t|\mathbf{X}_{t-1}) := \mathcal{N}(\mathbf{X}_t; \sqrt{1-\beta_t}\mathbf{X}_{t-1}, \beta_t \boldsymbol{I}). \quad (2)$$

The reverse process, in contrast to the forward process, is a Markov chain with learned Gaussian transitions $p_{\boldsymbol{\theta}}(\mathbf{X}_{t-1}|\mathbf{X}_t)$ starting at $p(\mathbf{X}_T) := \mathcal{N}(\mathbf{X}_T; \mathbf{0}, \boldsymbol{I})$:

$$p_{\boldsymbol{\theta}}(\mathbf{X}_{0:T}) := p(\mathbf{X}_T)\prod_{t=1}^{T} p_{\boldsymbol{\theta}}(\mathbf{X}_{t-1}|\mathbf{X}_t), \quad p_{\boldsymbol{\theta}}(\mathbf{X}_{t-1}|\mathbf{X}_t) := \mathcal{N}(\mathbf{X}_{t-1}; \boldsymbol{\mu}_{\boldsymbol{\theta}}(\mathbf{X}_t, t), \boldsymbol{\Sigma}_{\boldsymbol{\theta}}(\mathbf{X}_t, t)). \quad (3)$$

The reverse conditional probability $p_{\boldsymbol{\theta}}(\mathbf{X}_{t-1}|\mathbf{X}_t)$ is tractable when conditioned on $\mathbf{X}_0$: $q(\mathbf{X}_{t-1}|\mathbf{X}_t, \mathbf{X}_0) := \mathcal{N}(\mathbf{X}_{t-1}; \boldsymbol{\mu}_t(\mathbf{X}_t, \mathbf{X}_0), \hat{\beta}_t \boldsymbol{I})$ where

$$\boldsymbol{\mu}_t(\mathbf{X}_t, \mathbf{X}_0) := \frac{\sqrt{\bar{\alpha}_{t-1}}\beta_t}{1-\bar{\alpha}_t}\mathbf{X}_0 + \frac{\sqrt{\alpha_t}(1-\bar{\alpha}_{t-1})}{1-\bar{\alpha}_t}\mathbf{X}_t$$

$$= \frac{(1-\bar{\alpha}_{t-1})\sqrt{\alpha_t\bar{\alpha}_t} + \beta_t\sqrt{\bar{\alpha}_{t-1}}}{1-\bar{\alpha}_t}\mathbf{X}_0 + \frac{(1-\bar{\alpha}_{t-1})\sqrt{\alpha_t(1-\bar{\alpha}_t)}}{1-\bar{\alpha}_t}\boldsymbol{\Sigma},$$

$$\hat{\beta}_t := \frac{1-\bar{\alpha}_{t-1}}{1-\bar{\alpha}_t}\beta_t, \quad (4)$$

$\alpha_t := 1 - \beta_t$, $\bar{\alpha}_t := \prod_{s=1}^{t} \alpha_s$, and $\boldsymbol{\Sigma} \sim \mathcal{N}(\mathbf{0}, \boldsymbol{I})$. The ultimate training objective of DDPMs can be parameterized to learn approximator $\boldsymbol{\mu}_{\boldsymbol{\theta}}$ by minimizing the difference between $\boldsymbol{\mu}_t$ and $\boldsymbol{\mu}_{\boldsymbol{\theta}}$ (Sohl-Dickstein et al., 2015):

$$\mathcal{L} = \sum_{t \geq 1} \mathbb{E}_q[\|\boldsymbol{\mu}_t(\mathbf{X}_t, \mathbf{X}_0) - \boldsymbol{\mu}_{\boldsymbol{\theta}}(\mathbf{X}_t, t)\|^2]. \quad (5)$$

## 3 CONTINUOUS OPTIMIZATION BY DAG-INVARIANT DIFFUSION MODEL

In this section, we introduce a regularized optimization scheme with a novel variation-negotiation regularizer to address instability. Additionally, we propose a diffusion model, $\boldsymbol{D}^3PM$, which shares an equivalent training objective with the proposed regularized optimization objective.

### 3.1 CONTINUOUS PROGRAM WITH VARIATION-NEGOTIATION REGULARIZER

Continuous optimization-based approaches for CD involve modeling a continuous program (Zheng et al., 2018):

$$\boldsymbol{A}^*, \boldsymbol{\theta}^* = \underset{\boldsymbol{A}, \boldsymbol{\theta}}{\arg\min} \, D(\boldsymbol{f}_{\boldsymbol{\theta}}(\mathbf{X}\boldsymbol{A}), \mathbf{X}), \qquad \text{s.t. } \boldsymbol{A} \text{ is a DAG}, \quad (6)$$

where $D$ is a similarity measure, and $\boldsymbol{f}_{\boldsymbol{\theta}}$ is a parameterized function used to approximate $\boldsymbol{f}$ in Eq. (1). A high-quality solution to the continuous program is expected to satisfy two conditions: the minimization problem is solved and the DAG-ness constraint is satisfied. The focus of our work is on improving the solution to the minimization problem, which can be formulated as an inverse problem: given $\mathbf{X}$, $\boldsymbol{A}$ needs to be solved. Unfortunately, inverse problems always suffer from instability, where small variations in the space of $\mathbf{X}$ can correspond to very large variances in the matching parameters (Kasim et al., 2019; Calvetti & Somersalo, 2018).

To address the instability, we aim to reformulate the problem in a way that limits its instability and makes it possible to recover reasonably good solutions, a process known as regularization (Calvetti

& Somersalo, 2018). Our contribution is to pose the solution to the minimization problem in Eq. (6) as a regularized optimization scheme and propose a novel regularizer $R(A)$:

$$\min_{\boldsymbol{A},\boldsymbol{\theta}} \overbrace{D(\boldsymbol{f_\theta}(\mathbf{X}\boldsymbol{A}),\mathbf{X})}^{Data\ Consistency} + \lambda \overbrace{R(\boldsymbol{A})}^{Regularization} \quad , \tag{7}$$

where $R$ is designed to restrict the solutions to the space of desirable $\boldsymbol{A}$, and $\lambda$ is a positive scalar determining the balance between matching the data and minimizing $R(\boldsymbol{A})$.

**Variation-negotiation Regularizer** Traditional regularization methods uniformly explore the characteristics of DAGs, for example, by applying an $\ell_1$ penalty on graphs. In contrast, the proposed regularizer focuses on exploiting the variation variable $\mathbf{Z}$. The regularization effect of regulating $\mathbf{Z}$ on $\boldsymbol{A}$ can be found in the formula $\mathbf{Z} = \mathbf{X} - \boldsymbol{f}(\mathbf{X}\boldsymbol{A})$, which is derived from Eq. (1). Therefore, $\mathbf{Z}$ directly influences the measure of the data consistency term in Eq. (7). Without making general hypotheses about true DAGs, such as graph sparsity, the variation-negotiation regularizer aims to estimate the variation $\mathbf{Z}$ accurately. The estimation would consequently have a regularization effect on $\boldsymbol{A}$, purely based on the observed data $\mathbf{X}$. However, due to the inaccessibility of the variation $\mathbf{Z}$, we introduce two learnable counterparts, $\mathbf{Z_X}$ and $\mathbf{Z_N}$, with a negotiation strategy to collaboratively probe its value. Specifically, $\mathbf{Z_X}$ and $\mathbf{Z_N}$ function as two separate predictors from different viewpoints to estimate the variation $\mathbf{Z}$, to ensure consistency through negotiation.

We first describe the setting of the two counterparts. The design of $\mathbf{Z_X}$ is derived from formula $\mathbf{Z} = \mathbf{X} - \boldsymbol{f}(\mathbf{X}\boldsymbol{A})$. However, $\boldsymbol{f}$ is not accessible here. As a remedy, we use a parameterized estimator $\boldsymbol{f_\theta}$ to approximate $\boldsymbol{f}$. Then, $\mathbf{Z_X}$ is defined as $\mathbf{Z_X} := \mathbf{X} - \boldsymbol{f_\theta}(\mathbf{X}\boldsymbol{A}) \approx \mathbf{Z}$. Another counterpart, $\mathbf{Z_N}$, draws heavily from the philosophy of Denoising Autoencoders (Vincent et al., 2008), which suggests that partially destroyed data help reconstruct clean "repaired" data. Here, noisy data, symbolized as $\mathbf{X} + \mathbf{N}$, with artificial noise $\mathbf{N}$ drawn from some pre-defined distribution facilitate the recovery of $\mathbf{Z}$. Specifically, by inputting the noisy data, a parameterized estimator $\boldsymbol{g_\phi}$ is employed to predict $\mathbf{Z} + \mathbf{N}$. As a result, it holds that $\mathbf{Z_N} := \boldsymbol{g_\phi}(\mathbf{X}+\mathbf{N}) - \mathbf{N} \approx (\mathbf{Z}+\mathbf{N}) - \mathbf{N} = \mathbf{Z}$.

As we achieve $\mathbf{Z_X}$ and $\mathbf{Z_N}$, our objective is to encourage them to reach a consensus. This entails both $\mathbf{Z_X}$ approaching $\mathbf{Z_N}$ and vice versa. The level of agreement is assessed using the dot product for each observation. A higher positive value indicates a significant level of agreement. Finally, the regularized minimization objective with the variation-negotiation regularizer is formulated as:

$$\min_{\boldsymbol{A},\boldsymbol{\theta},\boldsymbol{\phi}} \overbrace{\underbrace{\|\mathbf{X} - \boldsymbol{f_\theta}(\mathbf{X}\boldsymbol{A})\|^2}_{\|\mathbf{Z_X}\|^2} + \underbrace{\|\boldsymbol{g_\phi}(\mathbf{X}+\mathbf{N}) - \mathbf{N}\|^2}_{\|\mathbf{Z_N}\|^2} - \underbrace{\lambda tr((\mathbf{X}-\boldsymbol{f_\theta}(\mathbf{X}\boldsymbol{A}))(\boldsymbol{g_\phi}(\mathbf{X}+\mathbf{N})-\mathbf{N})^T)}_{\lambda tr(\mathbf{Z_X}\mathbf{Z_N}^T)}}^{Data\ Consistency \qquad\qquad Regularization\ on\ \boldsymbol{A}},$$
$$\tag{8}$$

where the computation of the dot product is concisely expressed by calculating the matrix trace ($tr$), and $\lambda$ is a hyper-parameter that controls the strength of the negotiation agreement between $\mathbf{Z_X}$ and $\mathbf{Z_N}$.

**Optimization Objective as One Variation Consistency Term** If the value of $\lambda$ is set to 2, a specific expression of Eq. (8) can be derived:

$$\min_{\boldsymbol{A},\boldsymbol{\theta},\boldsymbol{\phi}} \| \overbrace{\underbrace{(\mathbf{X} - \boldsymbol{f_\theta}(\mathbf{X}\boldsymbol{A}))}_{\mathbf{Z_X}} - \underbrace{(\boldsymbol{g_\phi}(\mathbf{X}+\mathbf{N}) - \mathbf{N})}_{\mathbf{Z_N}}}^{Variation\ Consistency} \|^2, \tag{9}$$

which provides an alternative interpretation for the regularized minimization problem given in Eq. (7), where the problem is formulated as a data consistency term and a regularization term. The problem is now presented as a single variation consistency term without any regularization term. In this reformulation, $\mathbf{Z_X}$ plays a dual role — measuring variation ($\mathbf{X} - \boldsymbol{f_\theta}(\mathbf{X}\boldsymbol{A}) \approx \mathbf{Z}$) and optimizing $\boldsymbol{A}$.

Two valuable properties can be observed from the equivalent expression. Firstly, it is strictly non-negative, which facilitates optimization solvers. Another property is that the simple one-variation consistency term can be easily extended to multiple ones by diversifying the noise term $\mathbf{N}$. Diverse noises are beneficial for probing the true value of variation $\mathbf{Z}$, as diversified noisy data provide different observations for denoising techniques to recover variations.

**Optimization Objective as Multiple Variation Consistency Terms** The second property motivates us to further improve the minimization objective:

$$\min_{\boldsymbol{A},\boldsymbol{\theta},\boldsymbol{\phi}} \sum_t \|c_{data}(t)(\mathbf{X} - \boldsymbol{f_\theta}(\mathbf{X}\boldsymbol{A})) - c_{noise}(t)(\boldsymbol{g_\phi}(\mathbf{X} + \mathbf{N}_t) - \mathbf{N}_t)\|^2, \qquad (10)$$

where $\mathbf{N}_t$ denotes the $t$-th imposed noise. $c_{noise}(t)$ quantifies the noise magnitudes of $\mathbf{N}_t$, whereas $c_{data}(t)$ is set inversely proportional to $c_{noise}(t)$. The ranges of the two coefficients are $(0, 1)$. Intuitively, coefficients $c_{data}(t)$ and $c_{noise}(t)$ are designed to avoid the negative influence brought by the approximation error of variations for over-vast noises. Additionally, to ensure the negotiation effect between $\mathbf{Z_X}$ and $\mathbf{Z_N}$ is reserved with the added coefficients, there is a modification to $\boldsymbol{g_\phi}$ in Eq. (9). Given $\mathbf{X} + \mathbf{N}_t$, the estimation target of $\boldsymbol{g_\phi}$ is changed to $\frac{c_{data}(t)}{c_{noise}(t)}\mathbf{Z} + \mathbf{N}_t$, such that $c_{noise}(t)(\boldsymbol{g_\phi}(\mathbf{X} + \mathbf{N}_t) - \mathbf{N}_t) = c_{noise}(t)(\frac{c_{data}(t)}{c_{noise}(t)}\mathbf{Z_N} + \mathbf{N}_t - \mathbf{N}_t) = c_{data}(t)\mathbf{Z_N}$ holds and is thus able to negotiate with $c_{data}(t)(\mathbf{X} - \boldsymbol{f_\theta}(\mathbf{X}\boldsymbol{A})) = c_{data}(t)\mathbf{Z_X}$. This setting allows the evolved minimization objective to work effectively for varying noises. Given an instance, if the magnitude of $t$-th noise is vast, $c_{noise}(t)$ is large, then $c_{data}(t)$ becomes small. According to $c_{data}(t)(\mathbf{X} - \boldsymbol{f_\theta}(\mathbf{X}\boldsymbol{A})) - c_{noise}(t)(\boldsymbol{g_\phi}(\mathbf{X} + \mathbf{N}_t) - \mathbf{N}_t) = c_{data}(t)(\mathbf{Z_X} - \mathbf{Z_N})$, the effect of $t$-th noise is small. The mechanism prevents the performance of estimating $\boldsymbol{A}$ from degenerating for over-vast noises.

**Connection to Diffusion Models** Upon performing algebraic manipulations on Eq. (10), we arrive at the following expression:

$$\min_{\boldsymbol{A},\boldsymbol{\theta},\boldsymbol{\phi}} \sum_t \| \overbrace{(c_{data}(t)\mathbf{X} + c_{noise}(t)\mathbf{N}_t)}^{Blurred\ Data} - \overbrace{(c_{data}(t)\boldsymbol{f_\theta}(\mathbf{X}\boldsymbol{A}) + c_{noise}(t)\boldsymbol{g_\phi}(\mathbf{X} + \mathbf{N}_t))}^{Approximator} \|^2, \quad (11)$$

where $c_{data}(t)\mathbf{X} + c_{noise}(t)\mathbf{N}_t$ represents blurred data, which is the sum of faded clean data $c_{data}(t)\mathbf{X}$ and weighted noise $c_{noise}(t)\mathbf{N}_t$. The minimization objective follows the learning paradigm in which an approximator is trained to estimate blurred data with different noise magnitudes.

By setting all noises $\mathbf{N}_t$ to be independently drawn from standard Gaussian distribution, the connection between the proposed regularized minimization objective and diffusion models becomes apparent:

$$\min_{\boldsymbol{A},\boldsymbol{\theta},\boldsymbol{\phi}} \sum_t \| \overbrace{(c_{data}(t)\mathbf{X}_0 + c_{noise}(t)\mathbf{N}_t)}^{\boldsymbol{\mu}_t} - \overbrace{(c_{data}(t)\boldsymbol{f_\theta}(\mathbf{X}_0\boldsymbol{A}) + c_{noise}(t)\boldsymbol{g_\phi}(\mathbf{X}_t, t))}^{\boldsymbol{\mu_\theta}} \|^2. \quad (12)$$

Upon reviewing Eq. (5), the training objective of DDPMs consists of two terms: the mean of $t$-th noised data $\boldsymbol{\mu}_t$ and the corresponding approximator $\boldsymbol{\mu_\theta}$. The terms $\boldsymbol{\mu}_t$ and $\boldsymbol{\mu_\theta}$ in Eq. (5) conceptually resemble the blurred data term and the approximator term in Eq. (11), respectively. Therefore, the minimization objective is similar to the learning objective of DDPMs. The only difference between Eq. (11) and (12) is the input of $\boldsymbol{g_\phi}$. Here, the random variables $\mathbf{X} + \mathbf{N}_t$ are replaced by random variables $\mathbf{X}_t$, which are generated according to a Markov process $q(\mathbf{X}_t|\mathbf{X}_{t-1})$ in terms of $\mathbf{N}_t$, starting from $t = 1$ with $\mathbf{X}_0 := \mathbf{X}$. Nonetheless, this modification does not alter the nature of the input, since they are all noisy data generated along with noise $\mathbf{N}_t$, although in different manners.

Even though the resemblance between the proposed minimization objective and the training objective of DDPMs has been uncovered, there are challenges in transitioning the resemblance to strict equivalence. Firstly, existing diffusion models are designed to generate data without consideration for DAGs and ANMs. This results in no diffusion model instantiating $\boldsymbol{\mu_\theta}$ term in Eq. (12). Another one is how to specify the $t$-dependent coefficients $c_{data}(t)$ and $c_{noise}(t)$ such that the equivalence is strictly guaranteed. Once these challenges are overcome, the solution (i.e. a diffusion model) will be qualified to respond to the duty call from continuous optimization-based CD methods, combating instability.

## 3.2 DAG-invariant Denoising Diffusion Probabilistic Model

In this section, we introduce a novel diffusion model called $\boldsymbol{D}^3PM$ for CD. The learning objective of $\boldsymbol{D}^3PM$ is demonstrated to be completely equivalent to the proposed regularized continuous

program in Eq. (12). Sec. 3.2.1 describes the integration of DAG and ANMs into $\boldsymbol{D}^3PM$ by introducing the concept of DAG-invariance. In Sec. 3.2.2, we analytically determine the coefficients $c_{data}(t)$ and $c_{noise}(t)$ for the proposed minimization objective. With the determined coefficients, the equivalence between the proposed minimization objective and the training objective of $\boldsymbol{D}^3PM$ is established in Sec. 3.2.3. Furthermore, in Sec. 3.2.4, we illustrate how to estimate discrete DAGs via trained $\boldsymbol{D}^3PM$s.

### 3.2.1 DAG-INVARIANCE

We introduce the concept of *DAG-invariance* for $\boldsymbol{D}^3PM$. Let $\boldsymbol{A}$ be the DAG of given tabular data, then $\boldsymbol{A}$ remains invariant during the noising process on tabular data $\mathbf{X}$. This means that all immediately noised data of $\mathbf{X}$ share an identical DAG. For each step $t$, the immediately generated $\mathbf{X}_t$ can be expressed as $\mathbf{X}_t = \boldsymbol{f}(\mathbf{X}_0\boldsymbol{A}) + \mathbf{Z} + \mathbf{N}_t$, where $\mathbf{N}_t$ represents the $t$-th noise, and the DAG $\boldsymbol{A}$ remains constant for every $t$. An opposite notion which might facilitate the understanding of DAG-invariance is DAG-variance: $\mathbf{X}_t$ is suggested to be represented as $\mathbf{X}_t = \boldsymbol{f}(\mathbf{X}_0\boldsymbol{A}_t) + \mathbf{Z}_t$, where $\boldsymbol{A}_t$ (we assume DAG $\boldsymbol{A}_t$ always exists for each $\mathbf{X}_t$) and $\mathbf{Z}_t$ are different from $\boldsymbol{A}$ and $\mathbf{Z}$, respectively, at least at one time. The notion of DAG-invariance can be extended to the forward and reverse diffusion processes of DDPMs as follows: $\mathbf{X}_t = c_1(t)(\boldsymbol{f}(\mathbf{X}_0\boldsymbol{A}) + \mathbf{Z}) + c_2(t)\mathbf{N}_t$, where $c_1(t)$ and $c_2(t)$ are certain time-dependent coefficients involved in diffusion process.

The concept of DAG-invariance offers two main benefits. Firstly, it makes the technique of variable substitution $\mathbf{X}_0 = \boldsymbol{f}(\mathbf{X}_0\boldsymbol{A}) + \mathbf{Z}$ feasible throughout the diffusion process, allowing $\boldsymbol{A}$ to be explicitly involved in the forward and reverse processes of $\boldsymbol{D}^3PM$. Additionally, as $\boldsymbol{A}$ corresponding to the given data is shared across all noisy data generated in all timesteps, we can treat $\boldsymbol{A}$ as a trainable matrix (parameters), paving the way for modelling the optimization problem. Lastly, the fundamental mechanism of DDPMs remains unaffected. Specifically, with DAG-invariance, the forward transition kernel of $\boldsymbol{D}^3PM$, $q(\mathbf{X}_t|\mathbf{X}_{t-1})$, remains consistent with the one in Eq. (2), which follows a Gaussian distribution. It also ensures that the reverse transition kernel $q(\mathbf{X}_{t-1}|\mathbf{X}_t)$ is also a Gaussian distribution (Feller, 1949).

There might be a concern about DAG-invariance: whether optimizing $\boldsymbol{A}$ would be negatively influenced as the imposed noise is extremely large. It should be reassured, since, for $\boldsymbol{D}^3PM$, coefficients $c_{data}(t)$ and $c_{noise}(t)$ are designed to scale the weights of optimizing $\boldsymbol{A}$ for varying noise magnitudes, as mentioned for Eq. 10.

### 3.2.2 SOLVING OPTIMIZATION PROBLEMS IN DIFFUSION PROCESS

There is no difference in the forward diffusion process between $\boldsymbol{D}^3PM$ and DDPMs, despite the introduction of DAG-invariance. This means that the forward process of $\boldsymbol{D}^3PM$ is the same as $q(\mathbf{X}_{1:T}|\mathbf{X}_0)$ defined in Eq. (2). Therefore, our focus should be on the reverse diffusion process of $\boldsymbol{D}^3PM$ — $p_{\boldsymbol{\theta},\boldsymbol{\phi}}(\mathbf{X}_{0:T}|\mathbf{X}_0)$. We will start by discussing the reverse conditional Gaussian transition kernel of $\boldsymbol{D}^3PM$ conditioned on $\mathbf{X}_0$, $q(\mathbf{X}_{t-1}|\mathbf{X}_t, \mathbf{X}_0)$, with DAG-invariance. Then, we will design the reverse conditional Gaussian transition approximator $p_{\boldsymbol{\theta},\boldsymbol{\phi}}(\mathbf{X}_{t-1}|\mathbf{X}_t, \mathbf{X}_0)$. This process will involve establishing the values for $c_{data}(t)$ and $c_{noise}(t)$.

**Reverse Conditional Gaussian Transition with DAG-Invariance** The reverse conditional Gaussian transition kernel of $\boldsymbol{D}^3PM$ is defined as $q(\mathbf{X}_{t-1}|\mathbf{X}_t, \mathbf{X}_0) := \mathcal{N}(\mathbf{X}_{t-1}; \boldsymbol{\mu}_t(\mathbf{X}_t, \mathbf{X}_0), \hat{\beta}_t\boldsymbol{I})$. The variance $\hat{\beta}_t\boldsymbol{I}$ is set to untrained time-dependent constants as shown in Eq. (4). The mean of $\boldsymbol{D}^3PM$ and $\boldsymbol{\mu}_t$ in Eq. (4) share an identical expression. With DAG-invariance, it can be written as:

$$\boldsymbol{\mu}_t(\mathbf{X}_t, \mathbf{X}_0) = \frac{(1-\bar{\alpha}_{t-1})\sqrt{\alpha_t\bar{\alpha}_t} + \beta_t\sqrt{\bar{\alpha}_{t-1}}}{1-\bar{\alpha}_t}(\boldsymbol{f}(\mathbf{X}_0\boldsymbol{A}) + \mathbf{Z}) + \frac{(1-\bar{\alpha}_{t-1})\sqrt{\alpha_t(1-\bar{\alpha}_t)}}{1-\bar{\alpha}_t}\boldsymbol{\Sigma}$$

$$= c_{data}(t)\boldsymbol{f}(\mathbf{X}_0\boldsymbol{A}) + c_{noise}(t)(\frac{c_{data}(t)}{c_{noise}(t)}\mathbf{Z} + \boldsymbol{\Sigma}), \tag{13}$$

where $\boldsymbol{\Sigma} \sim \mathcal{N}(\mathbf{0}, \boldsymbol{I})$, $c_{data}(t) := \frac{(1-\bar{\alpha}_{t-1})\sqrt{\alpha_t\bar{\alpha}_t} + \beta_t\sqrt{\bar{\alpha}_{t-1}}}{1-\bar{\alpha}_t}$ and $c_{noise}(t) := \frac{(1-\bar{\alpha}_{t-1})\sqrt{\alpha_t(1-\bar{\alpha}_t)}}{1-\bar{\alpha}_t}$ hold. The detailed derivation process is provided in Appendix A.1. We will later verify whether the values of $c_{data}(t)$ and $c_{noise}(t)$ determined here secure the equivalence between optimizing DAG and training $\boldsymbol{D}^3PM$ in Sec. 3.2.3.

**Reverse Conditional Gaussian Transition Approximator** The approximator is defined as $p_{\boldsymbol{\theta},\boldsymbol{\phi}}(\mathbf{X}_{t-1}|\mathbf{X}_t,\mathbf{X}_0) := \mathcal{N}(\mathbf{X}_{t-1};\boldsymbol{\mu}_{\boldsymbol{\theta},\boldsymbol{\phi}}(\mathbf{X}_t,t,\mathbf{X}_0),\boldsymbol{\Sigma}_{\boldsymbol{\theta}}(\mathbf{X}_t,t))$. Once the approximator is obtained, the reverse process can be represented as a Markov chain $p_{\boldsymbol{\theta},\boldsymbol{\phi}}(\mathbf{X}_{0:T}|\mathbf{X}_0) := p(\mathbf{X}_T)\prod_{t=1}^{T}p_{\boldsymbol{\theta},\boldsymbol{\phi}}(\mathbf{X}_{t-1}|\mathbf{X}_t,\mathbf{X}_0)$. For the variance $\boldsymbol{\Sigma}_{\boldsymbol{\theta}}$, we choose the untrained parameterization $\hat{\beta}_t\boldsymbol{I}$. Regarding $\boldsymbol{\mu}_{\boldsymbol{\theta},\boldsymbol{\phi}}$, thanks to DAG-invariance, we can parameterize it according to the expression of $\boldsymbol{\mu}_{\boldsymbol{\theta}}$ term in Eq. (12):

$$\boldsymbol{\mu}_{\boldsymbol{\theta},\boldsymbol{\phi}}(\mathbf{X}_t,t,\mathbf{X}_0) := c_{data}(t)\boldsymbol{f}_{\boldsymbol{\theta}}(\mathbf{X}_0\boldsymbol{A}) + c_{noise}(t)\boldsymbol{g}_{\boldsymbol{\phi}}(\mathbf{X}_t,t), \tag{14}$$

where

$$\boldsymbol{g}_{\boldsymbol{\phi}}(\mathbf{X}_t,t) := \frac{\sqrt{\bar{\alpha}_{t-1}}\beta_t}{1-\bar{\alpha}_t}\hat{\mathbf{X}}_0 + \frac{\sqrt{\alpha_t}(1-\bar{\alpha}_{t-1})}{1-\bar{\alpha}_t}\mathbf{X}_t, \quad \hat{\mathbf{X}}_0 := \frac{X_t - \sqrt{1-\bar{\alpha}_t}\boldsymbol{\Sigma}_{\boldsymbol{\phi}}(\mathbf{X}_t,t)}{\sqrt{\bar{\alpha}_t}}. \tag{15}$$

The approximated objective of $\boldsymbol{f}_{\boldsymbol{\theta}}$ and $\boldsymbol{g}_{\boldsymbol{\phi}}$ is consistent with the setting of the optimization objective as multiple variation consistency terms in Sec. 3.1. The parameterization of $\boldsymbol{g}_{\boldsymbol{\phi}}$ matches the expression of $\boldsymbol{\mu}_t$ in Eq. (4) but with different prediction objectives. More detailed setting about approximators $\boldsymbol{f}_{\boldsymbol{\theta}}$ and $\boldsymbol{\Sigma}_{\boldsymbol{\phi}}$ is provided in Appendix C.1.

One thing to note is that original tabular data $\mathbf{X}_0$ is involved in the reverse Gaussian transition approximator, which is different from unconditional DDPMs. For unconditional DDPMs, taking $\mathbf{X}_0$ as input is not allowed for reverse transition approximators. Nonetheless, conditioning $\mathbf{X}_0$ is not inappropriate for $\boldsymbol{D}^3PM$, since, in CD, $\mathbf{X}_0$ plays the role as a condition for estimating DAGs. And, this difference does not deprive the generative ability of $\boldsymbol{D}^3PM$ at the sampling stage, due to the existence of $\boldsymbol{g}_{\boldsymbol{\phi}}$ and variance $\boldsymbol{\Sigma}_{\boldsymbol{\theta}}$. Since the generative ability is not the main focus of CD, we leave the discussion to Appendix B.

### 3.2.3 EQUIVALENCE BETWEEN TRAINING $D^3PM$ AND SOLVING OPTIMIZATION PROBLEM

To match the density $q(\mathbf{X}_0)$, the learned reverse transition $p_{\boldsymbol{\theta},\boldsymbol{\phi}}(\mathbf{X}_{t-1}|\mathbf{X}_t,\mathbf{X}_0)$ can be trained by minimizing cross entropy. Following previous work (Sohl-Dickstein et al., 2015), a lower bound can be expressed in terms of Kullback-Leibler divergence for $\boldsymbol{D}^3PM$ (See Appendix A.3 for a derivation). The loss function $\mathcal{L}$ for $\boldsymbol{D}^3PM$ is defined as:

$$\mathcal{L} = \sum_{t\geq 1}\mathbb{E}_q[\frac{1}{2\hat{\beta}_t}\|\boldsymbol{\mu}_t(\mathbf{X}_t,\mathbf{X}_0) - \boldsymbol{\mu}_{\boldsymbol{\theta},\boldsymbol{\phi}}(\mathbf{X}_t,t,\mathbf{X}_0)\|^2]. \tag{16}$$

By dropping the weights as per (Ho et al., 2020) and plugging in Eq. (13) and (14), we obtain:

$$\mathcal{L} = \sum_{t\geq 1}\mathbb{E}_q[\|(c_{data}(t)\mathbf{X}_0 + c_{noise}(t)\boldsymbol{\Sigma}) - (c_{data}(t)\boldsymbol{f}_{\boldsymbol{\theta}}(\mathbf{X}_0\boldsymbol{A}) + c_{noise}(t)\boldsymbol{g}_{\boldsymbol{\phi}}(\mathbf{X}_t,t))\|^2]. \tag{17}$$

This shows that the determined coefficients $c_{data}(t)$ and $c_{noise}(t)$ ensure training $\boldsymbol{D}^3PM$ is equivalent to solving the proposed minimization objection of Eq. (12).

### 3.2.4 ESTIMATION OF DAGS

While $\boldsymbol{D}^3PM$ addresses instability in inverse problems of continuous-optimization based CD approaches, the acyclicity constraint in Eq. (6) is missing. To measure the DAG-ness of $\boldsymbol{A}$, an additional loss term $\mathcal{L}_{dag} := tr(e^{\boldsymbol{A}\odot\boldsymbol{A}} - d)$ as proposed in (Zheng et al., 2018) is introduced, where $\odot$ denotes the Hadamard product, resulting in the final training objective for $\boldsymbol{D}^3PM$:

$$\boldsymbol{A}^*,\boldsymbol{\theta}^*,\boldsymbol{\phi}^* = \arg\min_{\boldsymbol{A},\boldsymbol{\theta},\boldsymbol{\phi}}\sum_{t\geq 1}\mathbb{E}_q[\mathcal{L}_{inv} + \mathcal{L}_{dag}], \tag{18}$$

where $\mathcal{L}_{inv}$ represents $\|(c_{data}(t)\mathbf{X}_0 + c_{noise}(t)\boldsymbol{\Sigma}) - (c_{data}(t)\boldsymbol{f}_{\boldsymbol{\theta}}(\mathbf{X}_0\boldsymbol{A}) + c_{noise}(t)\boldsymbol{g}_{\boldsymbol{\phi}}(\mathbf{X}_t,t))\|^2$.

After obtaining the optimal continuous-valued matrix $\boldsymbol{A}^*$ using Eq. (18), a heuristic strategy is employed to derive a DAG. This involves setting a small threshold $\gamma$ to remove edges from $\boldsymbol{A}^*$ with absolute weights smaller than $\gamma$ (Ng et al., 2020). If the resulting graph still contains cycles, edges are iteratively removed starting from the lowest absolute weights until a DAG is obtained.

## 4 RELATED WORK

The approaches for CD can be categorized into four branches as defined in (Hasan et al., 2023). The first category is the *constraint-based approaches*, such as PC (Kalisch & Bühlmann, 2007; Spirtes et al., 2001), FCI, and CD-NOD (Colombo et al., 2012; Zhang, 2008). These approaches detect causal relationships in observational data through conditional independence tests and then infer whether the data satisfies a DAG. They offer strong interpretability and the ability to incorporate domain knowledge but heavily rely on the quantity and quality of data. The second category is *Functional Causal Model (FCM) based approaches*, such as ANM (Hoyer et al., 2008), CAM (Bühlmann et al., 2014), PNL (Zhang et al., 2015), IGIC (Janzing et al., 2012), FOM (Cai et al., 2020), SCORE (Rolland et al., 2022), SAM (Kalainathan et al., 2022), and DiffAN (Sanchez et al., 2023). These approaches distinguish among different DAGs in the same equivalence class by imposing additional assumptions on the data distributions and/or function classes. They exhibit strong applicability and are adept at handling nonlinear relationships but come with strong assumptions imposed by the model.

The third category is *Score-based algorithms*, such as GES (Chickering, 2002; Hauser & Bühlmann, 2012), fGES (Ramsey et al., 2017), RL-BIC (Zhu et al., 2020), and CORL (Wang et al., 2021). These algorithms search over the space of all possible DAGs to find the graph that best explains the data. They increase the potential for searching for the correct causal graph while preserving sufficient interpretability but significantly reduce the efficiency of the model due to searching through all possible DAG spaces. The fourth category is the *continuous optimization-based approaches* such as NOTEARS (Zheng et al., 2018), which transforms the originally discrete and challenging-to-optimize DAG search space into a continuous and optimizable constraint space (Chen et al., 2023; Hasan et al., 2023). These methods leverage the powerful learning capabilities of deep learning to learn accurate causal graphs and improve optimization capabilities (Hasan et al., 2023), coupled with reduced computation time when utilizing GPUs.

**Concurrent work** The proposed method is closest to continuous optimization-based approaches. It explores the similarity between the proposed regularized continuous program for CD and diffusion models. Transitioning the resemblance to equivalence, $D^3PM$ is accordingly designed. A related work is DiffAN, which studies the link between CD and diffusion models. However, the difference between our work and DiffAN is vast. Firstly, DiffAN belongs to the category of FCM-based CD methods, which deviates from continuous optimization-based approaches. Secondly, DiffAN heavily relies on SCORE, which takes advantage of the Hessian of the data log-likelihood for topological ordering. For estimating the Hessian, SCORE utilizes a second-order Stein gradient estimator over a radial basis function kernel, while DiffAN replaces the kernel-based estimation with diffusion models. By contrast, our work focuses on revealing the inseparable connection between diffusion models and CD rather than treating diffusion models as a plug-and-play density estimation approach.

## 5 EXPERIMENTS

We use gradient-based optimization to train $D^3PM$s according to Eq. (18). For more information on the model architecture and hyper-parameter settings, please see Appendix C.1. We evaluate the performance of synthetic and real data and compare it to state-of-the-art CD methods from observational data.

**Baselines:** We compare $D^3PM$s with 10 baselines. More details can be found in Appendix C.2. Regarding FCM-based approaches, we consider CAM (Bühlmann et al., 2014), SAM (Kalainathan et al., 2022), and DiffAN (Sanchez et al., 2023) as references. For score-based and continuous optimization-based models, we select the following methods: CORL (Wang et al., 2021), NOTEARS (Zheng et al., 2018), GOLEM (Ng et al., 2020), GraN-DAG (Lachapelle et al., 2020), GAE (Ng et al., 2019), DAG-GNN (Yu et al., 2019), and SDCD (Nazaret et al., 2024). **Metrics:** The experiments' metrics are averaged over five randomly generated datasets of different seeds over causal graphs and variations. Following (Zhu et al., 2020; Ng et al., 2019; 2020), we evaluate the estimated graphs using three metrics: Structural Hamming Distance (SHD), False Discovery Rate (FDR), and True Positive Rate (TPR). SHD measures the smallest number of edge additions, deletions, and reversals required to convert the estimated graph into the true DAG, implicitly taking both FDR and TPR into account. Therefore, we take SHD as the primary metric for all experiments.

## 5.1 SYNTHETIC DATASET

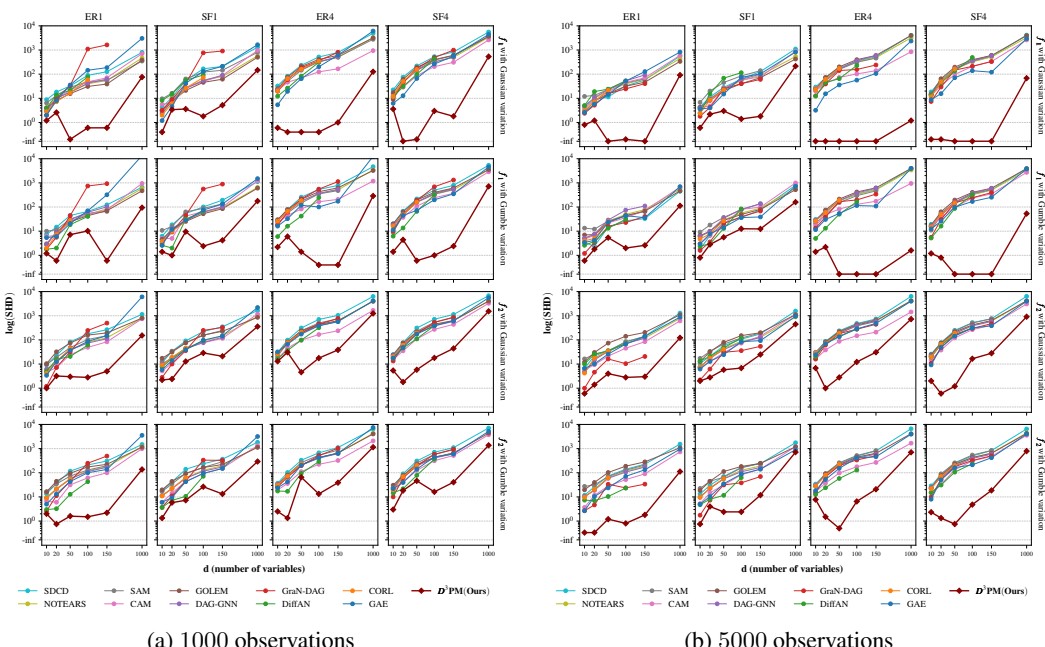

(a) 1000 observations      (b) 5000 observations

Figure 1: **Logarithm of SHD** for datasets with varying numbers of observations, generated by the function $\boldsymbol{f}_1$ or $\boldsymbol{f}_2$ with different variation distributions and causal graphs with increasing numbers of variables $d$ and varying edge numbers. ERi (SFi) means an ER (SF) graph whose number of edges is about $i \cdot d$.

In line with previous work (Sanchez et al., 2023; Wang et al., 2021; Lachapelle et al., 2020; Ng et al., 2019), we consider causal relationships with two functions: $\boldsymbol{f}_1(\mathbf{X}\boldsymbol{A}) := \boldsymbol{A}^T \cos(\mathbf{X} + \mathbf{1})$ and $\boldsymbol{f}_2(\mathbf{X}\boldsymbol{A}) := 2\sin(\boldsymbol{A}^T \cos(\mathbf{X} + \mathbf{1}) + 0.5 \cdot \mathbf{1}) + (\boldsymbol{A}^T \cos(\mathbf{X} + \mathbf{1}) + 0.5 \cdot \mathbf{1})$. The data is generated from ANMs using either function $\boldsymbol{f}_1$ or $\boldsymbol{f}_2$, with variation drawn from Gaussian or Gumbel distribution, and a causal graph. The causal graph $\boldsymbol{A}$ is constructed using either the Erdős–Rényi (ER) (Erdos et al., 1960) or the Scale Free (SF) (Bollobás et al., 2003) model. We conduct experiments with different sample sizes ($n \in \{1000, 5000\}$), graph sizes ($d \in \{10, 20, 50, 100, 150, 1000, 5000\}$), and numbers of edges ($1d$ or $4d$).

We classify causal graphs with less than or equal to 150 nodes as small and medium-scale graphs, and those with more than 150 nodes as large-scale.

**Datasets with Small and Medium-scale Causal Graphs** In Figure 1, the logarithm of SHD for $\boldsymbol{D}^3 PM$ and baselines is displayed. The corresponding SHD value can be found in the tables in Appendix D.1. Acorss all datasets, $\boldsymbol{D}^3 PM$ ranks first. The second-best positions vary depending on the observation number, variation type, causal relationship, and edge number. No baseline method secures the second place in at least $50\%$ of the datasets. This demonstrates the outstanding performance and robustness of $\boldsymbol{D}^3 PM$ across varying dataset settings. Furthermore, NOTEARS is modelled as solving a regularized optimization problem with the regularizer of $\ell_1$ penalty on graphs. However, NOTEARS dramatically falls behind our model, which shows the effectiveness of the proposed variation-based regularizer.

$\boldsymbol{D}^3 PM$ not only outperforms most baselines but also has a significant advantage over them, especially for causal graphs with a large number of nodes. The y-axis in Figure 1 represents log(SHD), so even a small gap in the figure implies a vast difference in SHD. For datasets of causal graphs containing 150 nodes and 150 edges, with causal relationship $\boldsymbol{f}_1$, $\boldsymbol{D}^3 PM$ outperforms the best baselines by an average of 49.68 SHD. As the number of edges increases to 600, this number rises to 197.88 SHD. The corresponding numbers for causal relationship $\boldsymbol{f}_2$ are 67.88 and 317.01, respectively. In addition to SHD, the metrics of FDR and TPR are documented in the tables in Appendix

D.1. When comparing the average FDR and TPR of $\boldsymbol{D}^3PM$ with those of the second-best baselines selected in terms of SHD, $\boldsymbol{D}^3PM$ achieves the best FDR in 168 out of 192 cases and the best TPR in 185 out of 192 cases. TPR measures actual positives, while FDR evaluates false positives.

**Datasets with Large-scale Causal Graphs** The scalability of our method and baselines is tested by increasing the number of nodes from 150 to 1000. The results are shown in the rightmost part of all sub-figures in Figure 1. As the graph size increases, some baselines are unable to run. DiffAN and CORL are unacceptably time-consuming for $d = 100$ and $d = 150$, respectively. Baselines SAM, GraN-DAG, and DAG-GNN fail to run for 1000 nodes. The figure demonstrates that the remaining baselines consistently lag behind $\boldsymbol{D}^3PM$s by a large margin. To further challenge all approaches, the number of vertices is significantly increased to 5000. In this case, only the baseline of SDCD is chosen, as it is the only work claiming to be qualified to run in the similiar data scale. The numerical results can be found in Table 18 and 19 in the appendix. $\boldsymbol{D}^3PM$ is the best-performing method in terms of SHD across all datasets. On average, the SHD of SDCD is 4.01 times larger than that of $\boldsymbol{D}^3PM$ for $\boldsymbol{f}_1$, and the number is 6.79 for $\boldsymbol{f}_2$.

Efficiency of $\boldsymbol{D}^3PM$ is also assessed in Appendix D.4. Among $\boldsymbol{D}^3PM$ and 4 baselines, $\boldsymbol{D}^3PM$ is ranked fourth when $d = 10$ and is moved to the second position as $d$ is increased to 1000, indicating that $\boldsymbol{D}^3PM$ is qualified to work on large-scale datasets.

## 5.2 REAL-WORLD DATASET

We compare $\boldsymbol{D}^3PM$s and baselines using a real dataset provided by (Sachs et al., 2005). This dataset pertains to a well-studied protein network problem and includes gene expression data consisting of 7466 observational data for 11 proteins. A signalling molecule causal graph, which is commonly accepted as ground truth, is used to evaluate the performance of CD methods. The results are shown in Figure 2 in the Appendix. Eight out of ten baselines produce SHDs greater than 23, GAE achieves an SHD of 18. Both $\boldsymbol{D}^3PM$ and GraN-DAG hold the top position with an SHD of 17.

## 6 CONCLUSION

To address the instability encountered by continuous optimization-based CD approaches, we propose the variation-negotiation regularizer, which eliminates any general hypotheses about true DAGs. Based on this regularizer, we identify a similarity between the regularized optimization problem and the training objective of diffusion models. This leads to the development of a novel diffusion model, called $\boldsymbol{D}^3PM$, whose training objective is equivalent to the regularized optimization problem. We demonstrate its superiority over various baselines with different dataset settings.

In terms of future work, it would be valuable to extend the assumption of data generation beyond ANMs. Additionally, exploring the adaptation of the variation-negotiation regularizer and $\boldsymbol{D}^3PM$ to observational time-series data represents a promising research direction.

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
