## A  EXTENDED DERIVATIONS

### A.1  DERIVATION OF REVERSE CONDITIONAL PROBABILITY CONDITIONED ON $\mathbf{X}_0$ WITH DAG-INVARIANCE

To establish the reverse process of $\boldsymbol{D}^3 PM$ using Bayes' rule, we begin with the equation:

$$q(\mathbf{X}_{t-1}|\mathbf{X}_t) = \frac{q(\mathbf{X}_t|\mathbf{X}_{t-1})q(\mathbf{X}_{t-1})}{q(\mathbf{X}_t)}. \tag{19}$$

Since $q(\mathbf{X}_t)$ and $q(\mathbf{X}_{t-1})$ are unknown, the reverse conditional probability is intractable. However, when conditioned on $\mathbf{X}_0$, the reverse conditional probability becomes tractable (Ho et al., 2020). Furthermore, with DAG-invariance, the reverse conditional probability can be represented as:

$$
\begin{aligned}
q(\mathbf{X}_{t-1}|\mathbf{X}_t, \mathbf{X}_0) &= \frac{q(\mathbf{X}_t|\mathbf{X}_{t-1}, \mathbf{X}_0)q(\mathbf{X}_{t-1}|\boldsymbol{f}(\mathbf{X}_0\boldsymbol{A}) + \mathbf{Z})}{q(\mathbf{X}_t|\boldsymbol{f}(\mathbf{X}_0\boldsymbol{A}) + \mathbf{Z})} \\
&= \frac{\mathcal{N}(\mathbf{X}_t; \sqrt{\alpha_t}\mathbf{X}_{t-1}, (1-\alpha_t)\boldsymbol{I})\mathcal{N}(\mathbf{X}_{t-1}; \sqrt{\bar{\alpha}_{t-1}}(\boldsymbol{f}(\mathbf{X}_0\boldsymbol{A}) + \mathbf{Z}), (1-\bar{\alpha}_{t-1})\boldsymbol{I})}{\mathcal{N}(\mathbf{X}_t; \sqrt{\bar{\alpha}_t}(\boldsymbol{f}(\mathbf{X}_0\boldsymbol{A}) + \mathbf{Z}), (1-\bar{\alpha}_t)\boldsymbol{I})}.
\end{aligned}
\tag{20}
$$

For simplicity, we take the vectors as scalars for the following illustration. It reads

$$
\begin{aligned}
q(\mathbf{X}_{t-1}|\mathbf{X}_t, \mathbf{X}_0) \propto \exp\Big\{ &\frac{(\mathbf{X}_t - \sqrt{\alpha_t}\mathbf{X}_{t-1})^2}{2(1-\alpha_t)} + \frac{(\mathbf{X}_{t-1} - \sqrt{\bar{\alpha}_{t-1}}(\boldsymbol{f}(\mathbf{X}_0\boldsymbol{A}) + \mathbf{Z}))^2}{2(1-\bar{\alpha}_{t-1})} \\
&- \frac{(\mathbf{X}_t - \sqrt{\bar{\alpha}_t}(\boldsymbol{f}(\mathbf{X}_0\boldsymbol{A}) + \mathbf{Z}))^2}{2(1-\bar{\alpha}_t)}\Big\}.
\end{aligned}
\tag{21}
$$

We can consider the exponential part as a quadratic function of the variable $\mathbf{X}_{t-1}$. The minimum value of the quadratic function corresponds to the mean of the resulting Gaussian. By taking the derivative of the quadratic function and setting it to 0, we obtain:

$$\boldsymbol{\mu}_t(\mathbf{X}_t, \mathbf{X}_0) = \frac{\sqrt{\bar{\alpha}_{t-1}}(1-\alpha_t)}{1-\bar{\alpha}_t}(\boldsymbol{f}(\mathbf{X}_0\boldsymbol{A}) + \mathbf{Z}) + \frac{\sqrt{\alpha_t}(1-\bar{\alpha}_{t-1})}{1-\bar{\alpha}_t}\mathbf{X}_t. \tag{22}$$

By reparameterizing the forward process as $\mathbf{X}_t = \sqrt{\bar{\alpha}_t}(\boldsymbol{f}(\mathbf{X}_0\boldsymbol{A}) + \mathbf{Z}) + \sqrt{1-\bar{\alpha}_t}\boldsymbol{\Sigma}$ for $\boldsymbol{\Sigma} \sim \mathcal{N}(\mathbf{0}, \boldsymbol{I})$ (See Appendix A.2), we obtain:

$$
\begin{aligned}
\boldsymbol{\mu}_t(\mathbf{X}_t, \mathbf{X}_0) &= \frac{\sqrt{\bar{\alpha}_{t-1}}(1-\alpha_t)}{1-\bar{\alpha}_t}(\boldsymbol{f}(\mathbf{X}_0\boldsymbol{A}) + \mathbf{Z}) + \frac{\sqrt{\alpha_t}(1-\bar{\alpha}_{t-1})}{1-\bar{\alpha}_t}(\sqrt{\bar{\alpha}_t}(\boldsymbol{f}(\mathbf{X}_0\boldsymbol{A}) + \mathbf{Z}) + \sqrt{1-\bar{\alpha}_t}\boldsymbol{\Sigma}) \\
&= \frac{(1-\bar{\alpha}_{t-1})\sqrt{\alpha_t\bar{\alpha}_t} + \beta_t\sqrt{\bar{\alpha}_{t-1}}}{1-\bar{\alpha}_t}(\boldsymbol{f}(\mathbf{X}_0\boldsymbol{A}) + \mathbf{Z}) + \frac{(1-\bar{\alpha}_{t-1})\sqrt{\alpha_t(1-\bar{\alpha}_t)}}{1-\bar{\alpha}_t}\boldsymbol{\Sigma}.
\end{aligned}
\tag{23}
$$

Similarly, when it comes to variance, we calculate the second derivative of the quadratic function and take its reciprocal. This gives us:

$$\hat{\beta}_t = \frac{1-\bar{\alpha}_{t-1}}{1-\bar{\alpha}_t}(1-\alpha_t). \tag{24}$$

### A.2  DERIVATION OF REPARAMETERIZED FORWARD PROCESS WITH DAG-INVARIANCE

By recursion, Eq. (2) gives us:

$$
\begin{aligned}
\mathbf{X}_t &= \sqrt{\alpha_t}\mathbf{X}_{t-1} + \sqrt{1-\alpha_t}\boldsymbol{\Sigma}_{t-1} \\
&= \sqrt{\alpha_t}(\sqrt{\alpha_{t-1}}\mathbf{X}_{t-2} + \sqrt{1-\alpha_{t-1}}\boldsymbol{\Sigma}_{t-2}) + \sqrt{1-\alpha_t}\boldsymbol{\Sigma}_{t-1} \\
&= \sqrt{\alpha_t\alpha_{t-1}}\mathbf{X}_{t-2} + \overbrace{\sqrt{\alpha_t}\sqrt{1-\alpha_{t-1}}\boldsymbol{\Sigma}_{t-2} + \sqrt{1-\alpha_t}\boldsymbol{\Sigma}_{t-1}}^{\vee}.
\end{aligned}
\tag{25}
$$

The sum of two Gaussians remains a Gaussian, motivating the computation of its new covariance:

$$\mathbb{E}[\mathbf{V}\mathbf{V}^T] = [(\sqrt{\alpha_t}\sqrt{1-\alpha_{t-1}})^2 + (\sqrt{1-\alpha_t})^2]\boldsymbol{I}$$
$$= [\alpha_t(1-\alpha_{t-1}) + 1 - \alpha_t]\boldsymbol{I}$$
$$= [1 - \alpha_t\alpha_{t-1}]\boldsymbol{I}. \tag{26}$$

Using the calculated covariance, we proceed to compute Eq. (25):

$$\mathbf{X}_t = \sqrt{\alpha_t\alpha_{t-1}}\mathbf{X}_{t-2} + \sqrt{1-\alpha_t\alpha_{t-1}}\boldsymbol{\Sigma}_{t-2}$$
$$= \sqrt{\alpha_t\alpha_{t-1}\alpha_{t-2}}\mathbf{X}_{t-3} + \sqrt{1-\alpha_t\alpha_{t-1}\alpha_{t-2}}\boldsymbol{\Sigma}_{t-3}$$
$$= ...$$
$$= \sqrt{\bar{\alpha}_t}\mathbf{X}_0 + \sqrt{1-\bar{\alpha}_t}\boldsymbol{\Sigma}_0$$
$$= \sqrt{\bar{\alpha}_t}(\boldsymbol{f}(\mathbf{X}_0\boldsymbol{A}) + \mathbf{Z}) + \sqrt{1-\bar{\alpha}_t}\boldsymbol{\Sigma}_0, \tag{27}$$

where $\{\boldsymbol{\Sigma}_i | i = t-1, ..., 0\}$ represent noise vectors. We simplify $\boldsymbol{\Sigma}_0$ as $\boldsymbol{\Sigma}$, and the simplified notation is used throughout the paper.

## A.3 DERIVATION OF TRAINING OBJECTIVE OF $D^3PM$

Based on the usual variational bound on negative log-likelihood, the loss function is defined as follows:

$$\mathcal{L} = \mathbb{E}_q[-\log\frac{p_{\boldsymbol{\theta},\boldsymbol{\phi}}(\mathbf{X}_{0:T}|\mathbf{X}_0)}{q(\mathbf{X}_{1:T}|\mathbf{X}_0)}]$$

$$= \mathbb{E}_q[-\log p(\mathbf{X}_T) - \sum_{t\geq 1}\log\frac{p_{\boldsymbol{\theta},\boldsymbol{\phi}}(\mathbf{X}_{t-1}|\mathbf{X}_t,\mathbf{X}_0)}{q(\mathbf{X}_t|\mathbf{X}_{t-1})}]$$

$$= \mathbb{E}_q[-\log p(\mathbf{X}_T) - \sum_{t>1}\log\frac{p_{\boldsymbol{\theta},\boldsymbol{\phi}}(\mathbf{X}_{t-1}|\mathbf{X}_t,\mathbf{X}_0)}{q(\mathbf{X}_t|\mathbf{X}_{t-1})} - \log\frac{p_{\boldsymbol{\theta},\boldsymbol{\phi}}(\mathbf{X}_0|\mathbf{X}_1,\mathbf{X}_0)}{q(\mathbf{X}_1|\mathbf{X}_0)}]$$

$$= \mathbb{E}_q[-\log p(\mathbf{X}_T) - \sum_{t>1}\log\frac{p_{\boldsymbol{\theta},\boldsymbol{\phi}}(\mathbf{X}_{t-1}|\mathbf{X}_t,\mathbf{X}_0)}{q(\mathbf{X}_{t-1}|\mathbf{X}_t,\mathbf{X}_0)}\cdot\frac{q(\mathbf{X}_{t-1}|\mathbf{X}_0)}{q(\mathbf{X}_t|\mathbf{X}_0)} - \log\frac{p_{\boldsymbol{\theta},\boldsymbol{\phi}}(\mathbf{X}_0|\mathbf{X}_1,\mathbf{X}_0)}{q(\mathbf{X}_1|\mathbf{X}_0)}]$$

$$= \mathbb{E}_q[-\log\frac{p(\mathbf{X}_T)}{q(\mathbf{X}_T|\mathbf{X}_0)} - \sum_{t>1}\log\frac{p_{\boldsymbol{\theta},\boldsymbol{\phi}}(\mathbf{X}_{t-1}|\mathbf{X}_t,\mathbf{X}_0)}{q(\mathbf{X}_{t-1}|\mathbf{X}_t,\mathbf{X}_0)} - \log p_{\boldsymbol{\theta},\boldsymbol{\phi}}(\mathbf{X}_0|\mathbf{X}_1,\mathbf{X}_0)]$$

$$= \mathbb{E}_q[\overbrace{D_{KL}(q(\mathbf{X}_T|\mathbf{X}_0)\|p(\mathbf{X}_T))}^{prior\ matching} + \overbrace{\sum_{t>1}D_{KL}(q(\mathbf{X}_{t-1}|\mathbf{X}_t,\mathbf{X}_0)\|p_{\boldsymbol{\theta},\boldsymbol{\phi}}(\mathbf{X}_{t-1}|\mathbf{X}_t,\mathbf{X}_0))}^{consistency} - \overbrace{\log p_{\boldsymbol{\theta},\boldsymbol{\phi}}(\mathbf{X}_0|\mathbf{X}_1,\mathbf{X}_0)}^{reconstruction}]. \tag{28}$$

Since there is no need to train for prior matching, we omit it and derive:

$$\mathcal{L} = \mathbb{E}_q[\overbrace{\sum_{t>1}D_{KL}(q(\mathbf{X}_{t-1}|\mathbf{X}_t,\mathbf{X}_0)\|p_{\boldsymbol{\theta},\boldsymbol{\phi}}(\mathbf{X}_{t-1}|\mathbf{X}_t,\mathbf{X}_0))}^{consistency} - \overbrace{\log p_{\boldsymbol{\theta},\boldsymbol{\phi}}(\mathbf{X}_0|\mathbf{X}_1,\mathbf{X}_0)}^{reconstruction}]. \tag{29}$$

Regarding the reconstruction term, the following analysis is performed. For $\boldsymbol{\mu}_t(\mathbf{X}_t, \mathbf{X}_0)$, by Eq. (4), considering $\alpha_0 = 1$ and $\bar{\alpha}_1 = \alpha_1$, we have:

$$\boldsymbol{\mu}_1(\mathbf{X}_1, \mathbf{X}_0) = \frac{(1-\bar{\alpha}_0)\sqrt{\alpha_1\bar{\alpha}_1} + \beta_1\sqrt{\bar{\alpha}_0}}{1-\bar{\alpha}_1}\mathbf{X}_0 + \frac{(1-\bar{\alpha}_0)\sqrt{\alpha_1(1-\bar{\alpha}_1)}}{1-\bar{\alpha}_1}\boldsymbol{\Sigma}$$

$$= \frac{\beta_1}{1-\alpha_1}\mathbf{X}_0$$

$$= \mathbf{X}_0. \tag{30}$$

Then, it can be deduced that:

$$\log p_{\boldsymbol{\theta},\boldsymbol{\phi}}(\mathbf{X}_0|\mathbf{X}_1,\mathbf{X}_0) = \log \mathcal{N}(\mathbf{X}_0; \boldsymbol{\mu}_{\boldsymbol{\theta},\boldsymbol{\phi}}(\mathbf{X}_1,1,\mathbf{X}_0), \hat{\beta}_1 \boldsymbol{I})$$

$$\propto -\frac{1}{2\hat{\beta}_1}\|\boldsymbol{\mu}_{\boldsymbol{\theta},\boldsymbol{\phi}}(\mathbf{X}_1,1,\mathbf{X}_0) - \mathbf{X}_0\|^2$$

$$= -\frac{1}{2\hat{\beta}_1}\|\boldsymbol{\mu}_{\boldsymbol{\theta},\boldsymbol{\phi}}(\mathbf{X}_1,1,\mathbf{X}_0) - \boldsymbol{\mu}_1(\mathbf{X}_1,\mathbf{X}_0)\|^2. \tag{31}$$

As for the consistency term, it results in:

$$D_{KL}(q(\mathbf{X}_{t-1}|\mathbf{X}_t,\mathbf{X}_0)\|p_{\boldsymbol{\theta},\boldsymbol{\phi}}(\mathbf{X}_{t-1}|\mathbf{X}_t,\mathbf{X}_0)) = D_{KL}(\mathcal{N}(\mathbf{X}_{t-1};\boldsymbol{\mu}_t(\mathbf{X}_t,\mathbf{X}_0),\hat{\beta}_t\boldsymbol{I})\|\mathcal{N}(\mathbf{X}_{t-1};\boldsymbol{\mu}_{\boldsymbol{\theta},\boldsymbol{\phi}}(\mathbf{X}_t,t,\mathbf{X}_0),\hat{\beta}_t\boldsymbol{I}))$$

$$= \frac{1}{2\hat{\beta}_t}\|\boldsymbol{\mu}_t(\mathbf{X}_t,\mathbf{X}_0) - \boldsymbol{\mu}_{\boldsymbol{\theta},\boldsymbol{\phi}}(\mathbf{X}_t,t,\mathbf{X}_0)\|^2. \tag{32}$$

Finally, substituting Eq. (31) and Eq. (32) into Eq. (29) simplifies $\mathcal{L}$ as:

$$\mathcal{L} = \sum_{t\geq 1}\mathbb{E}_q[\frac{1}{2\hat{\beta}_t}\|\boldsymbol{\mu}_t(\mathbf{X}_t,\mathbf{X}_0) - \boldsymbol{\mu}_{\boldsymbol{\theta},\boldsymbol{\phi}}(\mathbf{X}_t,t,\mathbf{X}_0)\|^2]. \tag{33}$$

## B  SAMPLING DATA BY $D^3PM$

In addition to estimating the DAG by discretizing $\boldsymbol{A}^*$ using Eq. (18), we can also sample new tabular data based on $\boldsymbol{A}^*$. The pseudo-code for the sampling procedure of $\boldsymbol{D}^3PM$ is described in Algorithm 1. Since $\boldsymbol{A}^*$ is utilized in each step of the reverse process, the generated tabular data reflects the learned causal relationship among variables.

---

**Algorithm 1** Sampling new tabular data.

---

1: Sample $\mathbf{X}_T \sim \mathcal{N}(\mathbf{0},\boldsymbol{I})$
2: **for** $t = T,...,1$ **do**
3:     $\boldsymbol{\Sigma} = \mathbf{0}$
4:     **if** $t > 1$ **then**
5:         $\boldsymbol{\Sigma} \sim \mathcal{N}(\mathbf{0},\boldsymbol{I})$
6:     **end if**
7:     $\mathbf{X}_{t-1} = c_{data}(t)\boldsymbol{f}_{\boldsymbol{\theta}}(\mathbf{X}_0\boldsymbol{A}^*) + c_{noise}(t)\boldsymbol{g}_{\boldsymbol{\phi}}(\mathbf{X}_t,t) + \sqrt{\hat{\beta}_t}\boldsymbol{\Sigma}$
8: **end for**

---

## C  EXPERIMENT SETTING

### C.1  DAG-INVARIANT DIFFUSION MODEL

In the neural architecture of $\boldsymbol{D}^3PM$, there are two main components: $\boldsymbol{f}_{\boldsymbol{\theta}}$ and $\boldsymbol{\Sigma}_{\boldsymbol{\phi}}$ in Eq. (14). For $\boldsymbol{f}_{\boldsymbol{\theta}}$, we use 2 multilayer perceptrons (MLPs), adapted from (Ng et al., 2019), with shared weights across all observed data. For $\boldsymbol{\Sigma}_{\boldsymbol{\phi}}$, we employ an MLP architecture adapted from (Gorishniy et al., 2021) with input noisy data $\mathbf{X}_t$ and diffusion time $t$. To encode $t$, we use the Transformer sinusoidal position embedding (Vaswani et al., 2017). The maximum value of $t$, i.e. $T$, is set to 1000.

To train $\boldsymbol{D}^3PM$, we use an Adam optimizer with a learning rate of $1e-3$ and apply mini-batch gradient-based optimization with a batch size of 32. The weight decay parameter for Adam is the only hyper-parameter that requires fine-tuning, with a search space of $\{1e-2, 1e-3, 1e-4\}$. For small and medium-scale datasets, we run 1000 epochs, while the number decreases to 500 for large-scale datasets.

The hyper-parameter for finally estimating discrete $\boldsymbol{A}$ is $\gamma$, set uniformly to 0.1 for all datasets without exception. Regarding the hardware environment, our GPU server consists of two 3.00 GHz Intel Xeon Gold 6248R CPUs, 384 GB of memory, and 4 Nvidia Quadro RTX 8000 GPUs.

## C.2 BASELINE METHODS

We indicate the implementations of baselines used for all experiments in the work in Table 1. The setting of their hyper-parameters completely follows either their original papers or the published codes.

| Methods | Implementation Codes |
|---|---|
| CAM | https://people.math.ethz.ch/~jopeters/code.html |
| SAM | https://github.com/FenTechSolutions/CausalDiscoveryToolbox |
| DiffAN | https://github.com/vios-s/DiffAN |
| CORL | https://github.com/huawei-noah/trustworthyAI |
| NOTEARS | https://github.com/xunzheng/notears |
| GOLEM | https://github.com/huawei-noah/trustworthyAI |
| GraN-DAG | https://github.com/kurowasan/GraN-DAG |
| GAE | https://github.com/huawei-noah/trustworthyAI |
| DAG-GNN | https://github.com/fishmoon1234/DAG-GNN |
| SDCD | https://github.com/azizilab/sdcd |

Table 1: Implementation codes for all baselines.

# D MORE EXPERIMENTAL RESULTS

## D.1 DATASETS WITH SMALL, MEDIUM, AND LARGE-SCALE CAUSAL GRAPHS

Below are some additional results related to the experiments discussed in Sec. 5.1. Firstly, we present the results of SHD metric to complement the information provided in Figure 1 expressed in terms of log(SHD). In addition to the SHD metric, we also include results for the TPR and FDR metrics. A better performance in estimating DAGs is indicated by smaller SHD and FDR values, as well as a higher TPR. The datasets that some baselines are unable to run on are denoted with mark "-".

## D.2 DATASETS WITH EXTREME LARGE-SCALE CAUSAL GRAPHS

We provide the detailed numerical results of extreme large-scale datasets ($d = 5000$) in Table 18 and 19. The related discussion can be found in Sec. 5.1.

## D.3 REAL-WORLD DATASET

We show the numerical results of the gene expression dataset in Figure 2. More discussion about the dataset and the results are given in Sec. 5.2.

## D.4 EFFICIENCY STUDY

We compare the training cost of $\boldsymbol{D}^3 PM$ with that of some representative baselines including NOTEARS, GAE, DiffAN, and SDCD. The hardware environment is described in Sec. C.1. In Figure 3, we record the total training time and corresponding performance (SHD) of each model. A model located at the bottom-left position indicates that it has a good performance in estimating DAGs with efficient computation. We choose three datasets with different scales: 1000 observations are generated based on causal relationship $\boldsymbol{f}_1$ with Gaussian variation and ER graphs having $d \in \{10, 100, 1000\}$ nodes and $4d$ edges.

We observe that, as the number of nodes increased, the training time of all approaches grows. In particular, DiffAN is not able to run when $d = 1000$. SDCD works efficiently in most cases. However, it comes at the cost of relatively poor estimation performance. GAE performs poorly in terms of efficiency. For efficiency, $\boldsymbol{D}^3 PM$ ranks 4th among all methods for small-scale data. However, its efficiency advantage becomes evident as the number of nodes increases. When $d = 1000$, $\boldsymbol{D}^3 PM$ achieves the second shortest training time. As for effectiveness, $\boldsymbol{D}^3 PM$ consistently

| d/e | Metrics | CAM | SAM | DiffAN | CORL | NOTEARS | GOLEM | GraN-DAG | GAE | DAG-GNN | SDCD | OURS |
|---|---|---|---|---|---|---|---|---|---|---|---|---|
| 10/1 | SHD↓ | 3.4 | 6.40 | 4.00 | 2.80 | 4.00 | 4.00 | 2.00 | 2.00 | 3.00 | 8.80 | **1.20** |
| | TPR↑ | 0.75 | 0.94 | 0.77 | 0.72 | 0.61 | 0.66 | 0.81 | 0.82 | 0.68 | 0.39 | **0.88** |
| | FDR↓ | 0.33 | 0.40 | 0.32 | 0.00 | 0.00 | 0.06 | 0.02 | 0.08 | 0.00 | 0.70 | **0.03** |
| 20/1 | SHD↓ | 9.2 | 13.20 | 13.75 | 7.20 | 10.60 | 8.20 | 9.80 | 8.00 | 10.20 | 18.20 | **2.60** |
| | TPR↑ | 0.73 | 0.66 | 0.68 | 0.72 | 0.56 | 0.71 | 0.58 | 0.69 | 0.58 | 0.52 | **0.88** |
| | FDR↓ | 0.35 | 0.44 | 0.47 | 0.08 | 0.09 | 0.10 | 0.01 | 0.07 | 0.04 | 0.04 | **0.00** |
| 50/1 | SHD↓ | 22.0 | 28.60 | 21.50 | 16.80 | 17.80 | 15.60 | 33.60 | 35.60 | 17.80 | 32.60 | **0.20** |
| | TPR↑ | 0.66 | 0.57 | 0.80 | 0.65 | 0.61 | 0.70 | 0.23 | 0.44 | 0.62 | 0.61 | **0.99** |
| | FDR↓ | 0.43 | 0.53 | 0.37 | 0.10 | 0.06 | 0.08 | 0.05 | 0.25 | 0.05 | 0.53 | **0.00** |
| 100/1 | SHD↓ | 47.8 | 48.80 | 76.50 | 58.00 | 40.00 | 31.00 | 1098.60 | 144.80 | 42.20 | 89.20 | **0.60** |
| | TPR↑ | 0.71 | 0.66 | 0.69 | 0.48 | 0.60 | 0.70 | 0.17 | 0.25 | 0.57 | 0.68 | **1.00** |
| | FDR↓ | 0.41 | 0.41 | 0.54 | 0.28 | 0.04 | 0.04 | 0.99 | 0.64 | 0.05 | 0.57 | **0.01** |
| 150/1 | SHD↓ | 70.8 | 55.00 | - | - | 54.60 | 39.00 | 1627.40 | 186.40 | 62.40 | 126.40 | **0.60** |
| | TPR↑ | 0.74 | 0.69 | - | - | 0.62 | 0.73 | 0.02 | 0.36 | 0.56 | 0.70 | **1.00** |
| | FDR↓ | 0.4 | 0.32 | - | - | 0.03 | 0.02 | 1.00 | 0.62 | 0.01 | 0.55 | **0.00** |
| 1000/1 | SHD↓ | 705.8 | 348.00 | - | - | 467.00 | 382.80 | - | 3030.50 | - | 798.00 | **75.80** |
| | TPR↑ | 0.75 | 0.69 | - | - | 0.57 | 0.66 | - | 0.43 | - | 0.73 | **0.97** |
| | FDR↓ | 0.47 | 0.16 | - | - | 0.06 | 0.06 | - | 0.83 | - | 0.50 | **0.05** |
| 10/4 | SHD↓ | 22.0 | 20.00 | 12.20 | 20.40 | 24.00 | 25.00 | 19.80 | 5.40 | 22.60 | 32.40 | **0.60** |
| | TPR↑ | 0.47 | 0.53 | 0.73 | 0.49 | 0.39 | 0.39 | 0.50 | 0.86 | 0.41 | 0.24 | **0.99** |
| | FDR↓ | 0.44 | 0.27 | 0.25 | 0.10 | 0.12 | 0.20 | 0.07 | 0.01 | 0.09 | 0.71 | **0.01** |
| 20/4 | SHD↓ | 46.2 | 50.20 | 26.50 | 65.40 | 64.80 | 75.40 | 59.20 | 19.40 | 64.40 | 81.80 | **0.40** |
| | TPR↑ | 0.52 | 0.51 | 0.79 | 0.35 | 0.27 | 0.22 | 0.24 | 0.76 | 0.31 | 0.31 | **0.99** |
| | FDR↓ | 0.36 | 0.35 | 0.25 | 0.39 | 0.34 | 0.51 | 0.05 | 0.02 | 0.34 | 0.69 | **0.00** |
| 50/4 | SHD↓ | 69.8 | 145.40 | 83.00 | 161.00 | 161.20 | 208.20 | 184.80 | 64.80 | 161.20 | 233.20 | **0.40** |
| | TPR↑ | 0.71 | 0.42 | 0.81 | 0.31 | 0.27 | 0.18 | 0.07 | 0.68 | 0.24 | 0.35 | **1.00** |
| | FDR↓ | 0.2 | 0.42 | 0.26 | 0.33 | 0.27 | 0.57 | 0.27 | 0.00 | 0.22 | 0.68 | **0.00** |
| 100/4 | SHD↓ | 124.80 | 324.80 | 309.75 | 345.00 | 321.60 | 408.20 | 357.00 | 202.40 | 330.40 | 515.00 | **0.40** |
| | TPR↑ | 0.74 | 0.38 | 0.73 | 0.21 | 0.26 | 0.08 | 0.33 | 0.48 | 0.21 | 0.38 | **1.00** |
| | FDR↓ | 0.16 | 0.50 | 0.46 | 0.31 | 0.23 | 0.62 | 0.36 | 0.06 | 0.17 | 0.70 | **0.00** |
| 150/4 | SHD↓ | 166.20 | 508.60 | - | - | 495.60 | 597.80 | 805.60 | 613.00 | 517.20 | 789.20 | **1.00** |
| | TPR↑ | 0.76 | 0.40 | - | - | 0.24 | 0.06 | 0.21 | 0.03 | 0.15 | 0.40 | **1.00** |
| | FDR↓ | 0.13 | 0.51 | - | - | 0.26 | 0.59 | 0.73 | 0.59 | 0.14 | 0.68 | **0.00** |
| 1000/4 | SHD↓ | 944.00 | 2766.00 | - | - | 3258.00 | 3167.60 | - | 6106.00 | - | 4891.20 | **126.40** |
| | TPR↑ | 0.81 | 0.38 | - | - | 0.25 | 0.31 | - | 0.08 | - | 0.41 | **0.97** |
| | FDR↓ | 0.11 | 0.26 | - | - | 0.19 | 0.24 | - | 0.88 | - | 0.64 | **0.00** |

Table 2: Average SHD, TPR, and FDR coming from 5 seeds. Each dataset comprises 1000 observations generated by ANMs with function $f_1$, Gaussian variation, and a causal graph. The causal graphs are generated by ER model with varying average degrees (e) and variable numbers (d). Bold and underlined font represent the first and second best results in terms of SHD, respectively.

| d/e | Metrics | CAM | SAM | DiffAN | CORL | NOTEARS | GOLEM | GraN-DAG | GAE | DAG-GNN | SDCD | OURS |
|---|---|---|---|---|---|---|---|---|---|---|---|---|
| 10/1 | SHD↓ | 5.6 | 9.60 | 8.20 | 2.00 | 3.20 | 2.40 | 3.20 | 1.20 | 3.00 | 4.40 | **0.40** |
| | TPR↑ | 0.73 | 1.00 | 0.69 | 0.82 | 0.67 | 0.78 | 0.69 | 0.89 | 0.69 | 0.67 | **0.96** |
| | FDR↓ | 0.4 | 0.51 | 0.46 | 0.10 | 0.09 | 0.08 | 0.06 | 0.02 | 0.08 | 0.34 | **0.00** |
| 20/1 | SHD↓ | 6.8 | 16.20 | 15.00 | 6.00 | 9.60 | 7.60 | 9.00 | 4.60 | 8.40 | 10.60 | **3.40** |
| | TPR↑ | 0.73 | 0.61 | 0.71 | 0.74 | 0.54 | 0.71 | 0.54 | 0.78 | 0.59 | 0.69 | **0.83** |
| | FDR↓ | 0.32 | 0.56 | 0.50 | 0.07 | 0.08 | 0.14 | 0.02 | 0.03 | 0.06 | 0.39 | **0.01** |
| 50/1 | SHD↓ | 27.2 | 63.60 | 55.25 | 26.20 | 28.00 | 21.20 | 41.00 | 42.00 | 25.60 | 55.20 | **3.60** |
| | TPR↑ | 0.71 | 0.40 | 0.61 | 0.58 | 0.51 | 0.64 | 0.16 | 0.48 | 0.53 | 0.52 | **0.93** |
| | FDR↓ | 0.38 | 0.72 | 0.60 | 0.21 | 0.14 | 0.11 | 0.03 | 0.43 | 0.09 | 0.62 | **0.00** |
| 100/1 | SHD↓ | 49.6 | 122.20 | 92.75 | 73.00 | 56.80 | 45.80 | 762.60 | 129.40 | 54.00 | 166.20 | **1.80** |
| | TPR↑ | 0.77 | 0.45 | 0.58 | 0.35 | 0.51 | 0.59 | 0.10 | 0.23 | 0.49 | 0.47 | **1.00** |
| | FDR↓ | 0.36 | 0.69 | 0.58 | 0.36 | 0.15 | 0.09 | 0.99 | 0.76 | 0.07 | 0.76 | **0.02** |
| 150/1 | SHD↓ | 87.8 | 144.80 | - | - | 79.40 | 61.80 | 906.80 | 210.80 | 90.20 | 209.60 | **5.20** |
| | TPR↑ | 0.71 | 0.45 | - | - | 0.53 | 0.64 | 0.02 | 0.30 | 0.41 | 0.49 | **0.98** |
| | FDR↓ | 0.42 | 0.64 | - | - | 0.12 | 0.08 | 1.00 | 0.55 | 0.04 | 0.72 | **0.02** |
| 1000/1 | SHD↓ | 959.2 | 801.00 | - | - | 571.00 | 500.00 | - | 1653.00 | - | 1304.20 | **147.20** |
| | TPR↑ | 0.66 | 0.33 | - | - | 0.50 | 0.58 | - | 0.31 | - | 0.53 | **0.91** |
| | FDR↓ | 0.56 | 0.42 | - | - | 0.14 | 0.12 | - | 0.76 | - | 0.67 | **0.07** |
| 10/4 | SHD↓ | 13.0 | 10.40 | 11.20 | 12.00 | 14.80 | 17.20 | 9.20 | 6.20 | 11.80 | 22.40 | **3.60** |
| | TPR↑ | 0.59 | 0.68 | 0.74 | 0.58 | 0.42 | 0.48 | 0.63 | 0.77 | 0.58 | 0.29 | **0.88** |
| | FDR↓ | 0.41 | 0.23 | 0.35 | 0.16 | 0.11 | 0.31 | 0.02 | 0.03 | 0.12 | 0.71 | **0.03** |
| 20/4 | SHD↓ | 28.8 | 45.20 | 31.00 | 49.80 | 50.20 | 61.40 | 43.40 | 12.80 | 48.00 | 75.00 | **0.00** |
| | TPR↑ | 0.66 | 0.50 | 0.76 | 0.41 | 0.32 | 0.27 | 0.33 | 0.82 | 0.34 | 0.31 | **1.00** |
| | FDR↓ | 0.27 | 0.43 | 0.35 | 0.34 | 0.28 | 0.47 | 0.03 | 0.03 | 0.23 | 0.73 | **0.00** |
| 50/4 | SHD↓ | 86.4 | 155.60 | 98.25 | 157.40 | 149.00 | 191.40 | 170.20 | 66.20 | 145.00 | 218.80 | **0.20** |
| | TPR↑ | 0.64 | 0.36 | 0.77 | 0.30 | 0.29 | 0.15 | 0.08 | 0.64 | 0.27 | 0.32 | **1.00** |
| | FDR↓ | 0.25 | 0.52 | 0.34 | 0.36 | 0.27 | 0.58 | 0.02 | 0.00 | 0.20 | 0.68 | **0.00** |
| 100/4 | SHD↓ | 205.40 | 381.40 | 433.75 | - | 326.60 | 401.00 | 487.60 | 262.80 | 320.80 | 531.40 | **3.00** |
| | TPR↑ | 0.63 | 0.30 | 0.60 | - | 0.25 | 0.11 | 0.24 | 0.32 | 0.23 | 0.33 | **1.00** |
| | FDR↓ | 0.27 | 0.61 | 0.59 | - | 0.30 | 0.59 | 0.68 | 0.10 | 0.22 | 0.73 | **0.01** |
| 150/4 | SHD↓ | 309.40 | 585.20 | - | - | 494.20 | 606.60 | 988.20 | 558.60 | 497.60 | 828.80 | **1.80** |
| | TPR↑ | 0.62 | 0.35 | - | - | 0.25 | 0.09 | 0.14 | 0.13 | 0.18 | 0.34 | **1.00** |
| | FDR↓ | 0.26 | 0.59 | - | - | 0.28 | 0.61 | 0.85 | 0.43 | 0.13 | 0.72 | **0.00** |
| 1000/4 | SHD↓ | 2618.20 | 3635.00 | - | - | 3361.50 | 3665.80 | - | 4407.50 | - | 5559.00 | **534.20** |
| | TPR↑ | 0.58 | 0.34 | - | - | 0.24 | 0.29 | - | 0.01 | - | 0.32 | **0.92** |
| | FDR↓ | 0.33 | 0.48 | - | - | 0.26 | 0.42 | - | 0.93 | - | 0.72 | **0.06** |

Table 3: Average SHD, TPR, and FDR coming from 5 seeds. Each dataset comprises 1000 observations generated by ANMs with function $f_1$ and Gaussian variation. The causal graphs are generated by SF model with varying average degrees (e) and variable numbers (d). Bold and underlined font represent the first and second best results in terms of SHD, respectively.

| d/e | Metrics | CAM | SAM | DiffAN | CORL | NOTEARS | GOLEM | GraN-DAG | GAE | DAG-GNN | SDCD | OURS |
|---|---|---|---|---|---|---|---|---|---|---|---|---|
| 10/1 | SHD↓ | 6.0 | 10.00 | _1.80_ | 2.00 | 2.80 | 3.00 | 1.80 | 5.40 | 2.80 | 7.40 | **1.20** |
| | TPR↑ | 0.51 | 0.98 | _0.89_ | 0.76 | 0.65 | 0.65 | 0.80 | 0.61 | 0.68 | 0.50 | **0.85** |
| | FDR↓ | 0.57 | 0.55 | _0.17_ | 0.00 | 0.00 | 0.02 | 0.03 | 0.15 | 0.06 | 0.61 | **0.00** |
| 20/1 | SHD↓ | 6.8 | 11.40 | _2.00_ | 7.25 | 9.60 | 9.00 | 5.60 | 6.00 | 8.80 | 14.80 | **0.60** |
| | TPR↑ | 0.73 | 0.74 | _0.98_ | 0.63 | 0.49 | 0.52 | 0.67 | 0.83 | 0.49 | 0.60 | **0.96** |
| | FDR↓ | 0.37 | 0.48 | _0.10_ | 0.09 | 0.08 | 0.10 | 0.01 | 0.09 | 0.05 | 0.57 | **0.00** |
| 50/1 | SHD↓ | 26.0 | 31.80 | _18.60_ | 27.50 | 25.60 | 24.40 | 44.00 | 20.80 | 29.60 | 44.80 | **7.20** |
| | TPR↑ | 0.7 | 0.60 | _0.83_ | 0.49 | 0.52 | 0.53 | 0.19 | 0.70 | 0.41 | 0.75 | **0.86** |
| | FDR↓ | 0.43 | 0.50 | _0.25_ | 0.15 | 0.07 | 0.05 | 0.37 | 0.14 | 0.01 | 0.52 | **0.00** |
| 100/1 | SHD↓ | 49.4 | 54.60 | _40.60_ | - | 50.60 | 43.80 | 742.20 | 70.00 | 60.60 | 65.60 | **10.20** |
| | TPR↑ | 0.72 | 0.61 | _0.87_ | - | 0.49 | 0.56 | 0.12 | 0.50 | 0.38 | 0.83 | **0.90** |
| | FDR↓ | 0.41 | 0.45 | _0.30_ | - | 0.04 | 0.02 | 0.98 | 0.30 | 0.01 | 0.41 | **0.01** |
| 150/1 | SHD↓ | 75.2 | _65.80_ | - | - | 83.80 | 70.20 | 927.00 | 321.60 | 102.00 | 123.40 | **0.60** |
| | TPR↑ | 0.75 | _0.65_ | - | - | 0.48 | 0.56 | 0.01 | 0.25 | 0.34 | 0.83 | **1.00** |
| | FDR↓ | 0.39 | _0.35_ | - | - | 0.06 | 0.04 | 1.00 | 0.76 | 0.01 | 0.47 | **0.00** |
| 1000/1 | SHD↓ | 948.4 | - | - | - | 585.00 | _470.00_ | - | 13514.50 | - | 660.40 | **94.40** |
| | TPR↑ | 0.75 | - | - | - | 0.46 | _0.55_ | - | 0.21 | - | 0.82 | **0.94** |
| | FDR↓ | 0.56 | - | - | - | 0.07 | _0.05_ | - | 0.98 | - | 0.41 | **0.04** |
| 10/4 | SHD↓ | 22.4 | 18.20 | _6.00_ | 25.75 | 27.80 | 30.00 | 18.00 | 16.20 | 26.60 | 28.40 | **2.20** |
| | TPR↑ | 0.44 | 0.55 | _0.85_ | 0.34 | 0.28 | 0.24 | 0.52 | 0.56 | 0.31 | 0.30 | **0.94** |
| | FDR↓ | 0.38 | 0.25 | _0.08_ | 0.09 | 0.10 | 0.25 | 0.03 | 0.07 | 0.08 | 0.55 | **0.00** |
| 20/4 | SHD↓ | 44.6 | 56.00 | _16.00_ | 66.00 | 68.00 | 78.40 | 61.40 | 32.80 | 66.20 | 78.40 | **6.00** |
| | TPR↑ | 0.53 | 0.42 | _0.84_ | 0.25 | 0.19 | 0.18 | 0.24 | 0.60 | 0.22 | 0.31 | **0.93** |
| | FDR↓ | 0.33 | 0.38 | _0.08_ | 0.29 | 0.25 | 0.52 | 0.11 | 0.04 | 0.22 | 0.63 | **0.00** |
| 50/4 | SHD↓ | 83.0 | 147.20 | _42.00_ | 184.75 | 173.60 | 208.40 | 214.60 | 117.60 | 175.40 | 255.40 | **1.40** |
| | TPR↑ | 0.66 | 0.40 | _0.87_ | 0.14 | 0.16 | 0.05 | 0.03 | 0.39 | 0.12 | 0.29 | **0.99** |
| | FDR↓ | 0.21 | 0.42 | _0.11_ | 0.34 | 0.24 | 0.73 | 0.66 | 0.01 | 0.14 | 0.73 | **0.00** |
| 100/4 | SHD↓ | 165.4 | 348.80 | 245.80 | - | 364.40 | 429.60 | 550.40 | _100.60_ | 365.60 | 533.80 | **0.40** |
| | TPR↑ | 0.66 | 0.34 | 0.74 | - | 0.15 | 0.02 | 0.11 | _0.76_ | 0.12 | 0.34 | **1.00** |
| | FDR↓ | 0.18 | 0.51 | 0.34 | - | 0.25 | 0.80 | 0.81 | _0.01_ | 0.12 | 0.69 | **0.00** |
| 150/4 | SHD↓ | 207.0 | 478.00 | - | - | 513.80 | 622.00 | 1127.60 | _172.80_ | 540.40 | 780.60 | **0.40** |
| | TPR↑ | 0.72 | 0.39 | - | - | 0.18 | 0.04 | 0.05 | _0.72_ | 0.11 | 0.40 | **1.00** |
| | FDR↓ | 0.16 | 0.44 | - | - | 0.16 | 0.67 | 0.95 | _0.00_ | 0.05 | 0.66 | **0.00** |
| 1000/4 | SHD↓ | _1187.20_ | - | - | - | 3427.00 | 3240.67 | - | 12488.00 | - | 4666.40 | **286.80** |
| | TPR↑ | _0.76_ | - | - | - | 0.17 | 0.23 | - | 0.06 | - | 0.40 | **0.93** |
| | FDR↓ | _0.13_ | - | - | - | 0.13 | 0.14 | - | 0.97 | - | 0.61 | **0.01** |

Table 4: Average SHD, TPR, and FDR coming from 5 seeds. Each dataset comprises 1000 observations generated by ANMs with function $f_1$ and Gumbel variation. The causal graphs are generated by ER model with varying average degrees (e) and variable numbers (d). Bold and underlined font represent the first and second best results in terms of SHD, respectively.

| d/e | Metrics | CAM | SAM | DiffAN | CORL | NOTEARS | GOLEM | GraN-DAG | GAE | DAG-GNN | SDCD | OURS |
|---|---|---|---|---|---|---|---|---|---|---|---|---|
| 10/1 | SHD↓ | 5.0 | 10.80 | _2.60_ | 3.75 | 4.40 | 4.60 | 3.20 | 2.60 | 3.80 | 6.20 | **1.40** |
| | TPR↑ | 0.64 | 0.98 | _0.84_ | 0.58 | 0.51 | 0.58 | 0.69 | 0.76 | 0.58 | 0.64 | **0.84** |
| | FDR↓ | 0.45 | 0.54 | _0.22_ | 0.05 | 0.04 | 0.15 | 0.08 | 0.04 | 0.04 | 0.49 | **0.00** |
| 20/1 | SHD↓ | 5.0 | 15.60 | _2.00_ | 9.75 | 12.80 | 11.00 | 9.20 | 11.80 | 12.80 | 18.40 | **1.00** |
| | TPR↑ | 0.79 | 0.68 | _0.93_ | 0.53 | 0.38 | 0.47 | 0.53 | 0.71 | 0.37 | 0.67 | **0.95** |
| | FDR↓ | 0.24 | 0.52 | _0.07_ | 0.08 | 0.12 | 0.09 | 0.02 | 0.33 | 0.12 | 0.53 | **0.00** |
| 50/1 | SHD↓ | 42.2 | 63.00 | 31.60 | 25.75 | 28.00 | _25.20_ | 44.80 | 27.40 | 27.60 | 53.80 | **9.60** |
| | TPR↑ | 0.59 | 0.34 | 0.78 | 0.57 | 0.50 | _0.53_ | 0.13 | 0.56 | 0.49 | 0.72 | **0.80** |
| | FDR↓ | 0.53 | 0.75 | 0.36 | 0.17 | 0.13 | _0.09_ | 0.24 | 0.18 | 0.11 | 0.53 | **0.00** |
| 100/1 | SHD↓ | 64.2 | 87.80 | 66.60 | - | 60.00 | _52.60_ | 556.40 | 75.00 | 62.40 | 101.00 | **2.40** |
| | TPR↑ | 0.7 | 0.46 | 0.79 | - | 0.44 | _0.50_ | 0.07 | 0.30 | 0.38 | 0.69 | **0.98** |
| | FDR↓ | 0.45 | 0.61 | 0.40 | - | 0.11 | _0.07_ | 0.98 | 0.25 | 0.03 | 0.54 | **0.00** |
| 150/1 | SHD↓ | 100.8 | 134.00 | - | - | 93.20 | _85.20_ | 887.00 | 132.80 | 101.00 | 192.60 | **4.20** |
| | TPR↑ | 0.67 | 0.45 | - | - | 0.42 | _0.46_ | 0.04 | 0.40 | 0.33 | 0.64 | **1.00** |
| | FDR↓ | 0.46 | 0.59 | - | - | 0.10 | _0.07_ | 0.99 | 0.39 | 0.02 | 0.63 | **0.03** |
| 1000/1 | SHD↓ | 1206.2 | - | - | - | 649.00 | _602.67_ | - | 1500.00 | - | 1122.80 | **176.60** |
| | TPR↑ | 0.64 | - | - | - | 0.40 | _0.48_ | - | 0.01 | - | 0.59 | **0.89** |
| | FDR↓ | 0.62 | - | - | - | 0.11 | _0.15_ | - | 0.97 | - | 0.58 | **0.07** |
| 10/4 | SHD↓ | 16.6 | 9.00 | _6.20_ | 16.25 | 17.80 | 20.20 | 10.00 | 11.40 | 17.80 | 20.00 | **1.40** |
| | TPR↑ | 0.44 | 0.74 | _0.82_ | 0.40 | 0.32 | 0.30 | 0.70 | 0.53 | 0.31 | 0.38 | **0.94** |
| | FDR↓ | 0.51 | 0.21 | _0.18_ | 0.15 | 0.17 | 0.36 | 0.16 | 0.00 | 0.14 | 0.57 | **0.00** |
| 20/4 | SHD↓ | 37.4 | 46.40 | _13.60_ | 47.80 | 50.40 | 62.80 | 47.60 | 45.00 | 50.00 | 67.40 | **4.40** |
| | TPR↑ | 0.53 | 0.45 | _0.85_ | 0.33 | 0.27 | 0.21 | 0.27 | 0.34 | 0.26 | 0.31 | **0.94** |
| | FDR↓ | 0.38 | 0.41 | _0.11_ | 0.21 | 0.19 | 0.50 | 0.04 | 0.21 | 0.16 | 0.67 | **0.01** |
| 50/4 | SHD↓ | 108.0 | 162.20 | _65.80_ | 160.25 | 157.40 | 194.40 | 183.20 | 72.60 | 157.80 | 215.60 | **0.60** |
| | TPR↑ | 0.55 | 0.32 | _0.81_ | 0.22 | 0.21 | 0.08 | 0.06 | 0.61 | 0.18 | 0.35 | **1.00** |
| | FDR↓ | 0.31 | 0.52 | _0.19_ | 0.31 | 0.24 | 0.66 | 0.44 | 0.00 | 0.16 | 0.65 | **0.00** |
| 100/4 | SHD↓ | 242.6 | 365.40 | 321.20 | - | 334.60 | 403.40 | 689.00 | _198.80_ | 343.40 | 496.40 | **1.00** |
| | TPR↑ | 0.55 | 0.30 | 0.65 | - | 0.18 | 0.04 | 0.09 | _0.49_ | 0.12 | 0.34 | **1.00** |
| | FDR↓ | 0.33 | 0.58 | 0.45 | - | 0.25 | 0.70 | 0.91 | _0.01_ | 0.09 | 0.68 | **0.00** |
| 150/4 | SHD↓ | 367.0 | 591.00 | - | - | 512.60 | 612.20 | 1314.20 | _350.20_ | 536.20 | 792.40 | **2.40** |
| | TPR↑ | 0.57 | 0.30 | - | - | 0.17 | 0.07 | 0.03 | _0.40_ | 0.09 | 0.34 | **1.00** |
| | FDR↓ | 0.32 | 0.59 | - | - | 0.23 | 0.64 | 0.97 | _0.04_ | 0.05 | 0.70 | **0.00** |
| 1000/4 | SHD↓ | _2870.20_ | - | - | - | 3493.00 | 3549.33 | - | 4574.50 | - | 5332.60 | **715.00** |
| | TPR↑ | _0.55_ | - | - | - | 0.16 | 0.22 | - | 0.01 | - | 0.32 | **0.87** |
| | FDR↓ | _0.37_ | - | - | - | 0.20 | 0.33 | - | 0.93 | - | 0.69 | **0.06** |

Table 5: Average SHD, TPR, and FDR coming from 5 seeds. Each dataset comprises 1000 observations generated by ANMs with function $f_1$ and Gumbel variation. The causal graphs are generated by SF model with varying average degrees (e) and variable numbers (d). Bold and underlined font represent the first and second best results in terms of SHD, respectively.

| d/e | Metrics | CAM | SAM | DiffAN | CORL | NOTEARS | GOLEM | GraN-DAG | GAE | DAG-GNN | SDCD | OURS |
|---|---|---|---|---|---|---|---|---|---|---|---|---|
| 10/1 | SHD↓ | 4.60 | 9.60 | 5.00 | 4.20 | 5.20 | 10.40 | _1.20_ | 3.40 | 6.20 | 9.20 | **1.00** |
|  | TPR↑ | 0.58 | 0.77 | 0.56 | 0.78 | 0.60 | 0.71 | _0.92_ | 0.74 | 0.65 | 0.36 | **0.96** |
|  | FDR↓ | 0.46 | 0.58 | 0.49 | 0.26 | 0.19 | 0.59 | 0.06 | 0.15 | 0.30 |  | **0.07** |
| 20/1 | SHD↓ | _7.20_ | 18.80 | 15.60 | 12.80 | 13.20 | 32.20 | 7.20 | 12.60 | 11.60 | 23.00 | **3.20** |
|  | TPR↑ | _0.69_ | 0.60 | 0.56 | 0.66 | 0.59 | 0.53 | 0.67 | 0.38 | 0.62 | 0.36 | **0.99** |
|  | FDR↓ | _0.33_ | 0.60 | 0.55 | 0.31 | 0.29 | 0.69 | 0.03 | 0.12 | 0.24 |  | **0.13** |
| 50/1 | SHD↓ | 24.40 | 48.20 | _20.60_ | 43.60 | 39.00 | 74.20 | 32.00 | 36.60 | 38.60 | 82.20 | **3.00** |
|  | TPR↑ | 0.62 | 0.42 | _0.76_ | 0.47 | 0.49 | 0.36 | 0.36 | 0.22 | 0.53 | 0.30 | **0.99** |
|  | FDR↓ | 0.40 | 0.67 | _0.32_ | 0.48 | 0.40 | 0.72 | 0.08 | 0.04 | 0.40 | 0.83 | **0.05** |
| 100/1 | SHD↓ | _49.00_ | 103.80 | 60.60 | - | 80.80 | 158.00 | 245.40 | 86.60 | 73.80 | 178.40 | **2.80** |
|  | TPR↑ | _0.67_ | 0.39 | 0.71 | - | 0.45 | 0.36 | 0.76 | 0.13 | 0.46 | 0.39 | **0.99** |
|  | FDR↓ | _0.38_ | 0.69 | 0.42 | - | 0.37 | 0.73 | 0.72 | 0.05 | 0.31 | 0.81 | **0.02** |
| 150/1 | SHD↓ | _82.80_ | 149.00 | - | - | 121.40 | 194.80 | 496.40 | 149.20 | 111.60 | 264.40 | **5.00** |
|  | TPR↑ | _0.66_ | 0.38 | - | - | 0.46 | 0.34 | 0.64 | 0.08 | 0.46 | 0.41 | **0.98** |
|  | FDR↓ | _0.41_ | 0.66 | - | - | 0.37 | 0.65 | 0.82 | 0.18 | 0.29 | 0.79 | **0.01** |
| 1000/1 | SHD↓ | _749.00_ | - | - | - | 794.00 | 797.00 | - | 6044.00 | - | 1136.00 | **152.50** |
|  | TPR↑ | _0.66_ | - | - | - | 0.47 | 0.52 | - | 0.12 | - | 0.50 | **0.89** |
|  | FDR↓ | _0.50_ | - | - | - | 0.36 | 0.38 | - | 0.98 | - | 0.63 | **0.04** |
| 10/4 | SHD↓ | _16.80_ | 21.20 | 16.80 | 26.60 | 28.20 | 28.20 | 26.80 | 32.00 | 29.20 | 29.40 | **13.00** |
|  | TPR↑ | _0.56_ | 0.49 | 0.60 | 0.34 | 0.28 | 0.31 | 0.27 | 0.14 | 0.29 | 0.26 | **0.63** |
|  | FDR↓ | _0.25_ | 0.32 | 0.29 | 0.24 | 0.24 | 0.26 | 0.09 | 0.25 | 0.32 | 0.66 | **0.16** |
| 20/4 | SHD↓ | 45.60 | 75.60 | _36.20_ | 76.60 | 76.80 | 82.20 | 73.20 | 65.80 | 75.00 | 97.60 | **30.60** |
|  | TPR↑ | 0.55 | 0.30 | _0.69_ | 0.20 | 0.17 | 0.15 | 0.09 | 0.20 | 0.17 | 0.18 | **0.64** |
|  | FDR↓ | 0.30 | 0.59 | _0.29_ | 0.48 | 0.47 | 0.57 | 0.13 | 0.15 | 0.45 | 0.81 | **0.06** |
| 50/4 | SHD↓ | 97.60 | 239.00 | 97.60 | 207.80 | 212.40 | 236.40 | 196.60 | 173.00 | 196.60 | 311.20 | **4.60** |
|  | TPR↑ | _0.62_ | 0.20 | 0.69 | 0.14 | 0.10 | 0.07 | 0.02 | 0.15 | 0.10 | 0.13 | **0.98** |
|  | FDR↓ | _0.23_ | 0.73 | 0.27 | 0.58 | 0.63 | 0.80 | 0.31 | 0.16 | 0.48 | 0.87 | **0.00** |
| 100/4 | SHD↓ | _166.60_ | 501.40 | 317.20 | - | 409.20 | 436.40 | 446.80 | 385.00 | 386.60 | 702.60 | **17.60** |
|  | TPR↑ | _0.68_ | 0.19 | 0.61 | - | 0.11 | 0.05 | 0.13 | 0.05 | 0.10 | 0.16 | **0.96** |
|  | FDR↓ | _0.20_ | 0.76 | 0.45 | - | 0.55 | 0.75 | 0.65 | 0.14 | 0.37 | 0.87 | **0.00** |
| 150/4 | SHD↓ | _238.20_ | 743.40 | - | - | 608.20 | 621.40 | 771.80 | 568.00 | 571.40 | 1031.60 | **38.40** |
|  | TPR↑ | _0.69_ | 0.19 | - | - | 0.11 | 0.04 | 0.12 | 0.03 | 0.08 | 0.16 | **0.93** |
|  | FDR↓ | _0.19_ | 0.75 | - | - | 0.59 | 0.72 | 0.79 | 0.06 | 0.42 | 0.87 | **0.00** |
| 1000/4 | SHD↓ | _1668.80_ | - | - | - | 4023.00 | 3942.00 | - | 4136.00 | - | 6264.20 | **1242.00** |
|  | TPR↑ | _0.68_ | - | - | - | 0.11 | 0.08 | - | 0.00 | - | 0.15 | **0.69** |
|  | FDR↓ | _0.18_ | - | - | - | 0.54 | 0.49 | - | 0.99 | - | 0.84 | **0.00** |

Table 6: Average SHD, TPR, and FDR coming from 5 seeds. Each dataset comprises 1000 observations generated by ANMs with function $f_2$ and Gaussian variation. The causal graphs are generated by ER model with varying average degrees (e) and variable numbers (d). Bold and underlined font represent the first and second best results in terms of SHD, respectively.

| d/e | Metrics | CAM | SAM | DiffAN | CORL | NOTEARS | GOLEM | GraN-DAG | GAE | DAG-GNN | SDCD | OURS |
|---|---|---|---|---|---|---|---|---|---|---|---|---|
| 10/1 | SHD↓ | 5.20 | 13.00 | 7.60 | 9.00 | 8.40 | 17.40 | _2.80_ | 5.80 | 10.00 | 11.40 | **2.20** |
|  | TPR↑ | 0.60 | 0.64 | 0.51 | 0.56 | 0.49 | 0.42 | _0.76_ | 0.60 | 0.44 | 0.36 | **0.91** |
|  | FDR↓ | 0.43 | 0.66 | 0.57 | 0.49 | 0.44 | 0.77 | _0.08_ | 0.18 | 0.52 | 0.75 | **0.13** |
| 20/1 | SHD↓ | 10.80 | 34.20 | 16.80 | 18.60 | 18.80 | 31.00 | _10.00_ | 14.00 | 18.40 | 33.40 | **2.40** |
|  | TPR↑ | 0.67 | 0.31 | 0.58 | 0.53 | 0.46 | 0.42 | _0.53_ | 0.28 | 0.52 | 0.27 | **1.00** |
|  | FDR↓ | 0.41 | 0.82 | 0.55 | 0.48 | 0.46 | 0.72 | _0.08_ | 0.07 | 0.47 | 0.83 | **0.11** |
| 50/1 | SHD↓ | 45.60 | 74.80 | 40.00 | 45.40 | 45.20 | 76.40 | _36.00_ | 39.60 | 43.00 | 91.00 | **12.80** |
|  | TPR↑ | 0.54 | 0.20 | 0.60 | 0.43 | 0.43 | 0.33 | _0.29_ | 0.24 | 0.44 | 0.35 | **0.93** |
|  | FDR↓ | 0.56 | 0.86 | 0.52 | 0.46 | 0.44 | 0.73 | _0.07_ | 0.06 | 0.42 | 0.81 | **0.17** |
| 100/1 | SHD↓ | _76.00_ | 153.00 | 92.40 | - | 89.60 | 159.00 | 247.20 | 97.00 | 78.00 | 222.40 | **28.80** |
|  | TPR↑ | _0.58_ | 0.29 | 0.52 | - | 0.44 | 0.34 | 0.50 | 0.08 | 0.41 | 0.29 | **0.96** |
|  | FDR↓ | _0.49_ | 0.80 | 0.59 | - | 0.44 | 0.73 | 0.78 | 0.21 | 0.32 | 0.87 | **0.20** |
| 150/1 | SHD↓ | _116.40_ | 246.40 | - | - | 145.20 | 226.20 | 326.20 | 148.00 | 128.20 | 349.00 | **21.00** |
|  | TPR↑ | _0.59_ | 0.29 | - | - | 0.41 | 0.32 | 0.44 | 0.05 | 0.39 | 0.30 | **0.97** |
|  | FDR↓ | _0.51_ | 0.81 | - | - | 0.49 | 0.72 | 0.71 | 0.42 | 0.39 | 0.87 | **0.10** |
| 1000/1 | SHD↓ | 1181.80 | - | - | - | 939.00 | _854.00_ | - | 2190.00 | - | 1644.60 | **357.00** |
|  | TPR↑ | 0.51 | - | - | - | 0.42 | _0.48_ | - | 0.18 | - | 0.39 | **0.84** |
|  | FDR↓ | 0.64 | - | - | - | 0.46 | _0.41_ | - | 0.89 | - | 0.76 | **0.19** |
| 10/4 | SHD↓ | 15.40 | 18.20 | 17.20 | 20.20 | 19.80 | 23.60 | _13.40_ | 17.60 | 22.00 | 24.80 | **5.40** |
|  | TPR↑ | 0.52 | 0.53 | 0.53 | 0.34 | 0.30 | 0.31 | _0.49_ | 0.28 | 0.27 | 0.31 | **0.82** |
|  | FDR↓ | 0.40 | 0.47 | 0.51 | 0.42 | 0.34 | 0.56 | _0.11_ | 0.03 | 0.45 | 0.69 | **0.08** |
| 20/4 | SHD↓ | _35.20_ | 62.00 | 45.80 | 60.00 | 61.80 | 74.80 | 56.40 | 48.40 | 60.60 | 75.40 | **1.80** |
|  | TPR↑ | _0.57_ | 0.32 | 0.50 | 0.26 | 0.19 | 0.18 | 0.14 | 0.26 | 0.21 | 0.17 | **0.98** |
|  | FDR↓ | _0.31_ | 0.61 | 0.47 | 0.47 | 0.48 | 0.67 | 0.12 | 0.09 | 0.44 | 0.79 | **0.01** |
| 50/4 | SHD↓ | _108.80_ | 227.60 | 111.40 | 193.20 | 191.00 | 211.00 | 181.40 | 174.00 | 183.80 | 311.00 | **5.80** |
|  | TPR↑ | _0.57_ | 0.21 | 0.64 | 0.15 | 0.13 | 0.08 | 0.02 | 0.07 | 0.12 | 0.16 | **0.97** |
|  | FDR↓ | _0.30_ | 0.74 | 0.35 | 0.58 | 0.56 | 0.73 | 0.33 | 0.26 | 0.50 | 0.87 | **0.01** |
| 100/4 | SHD↓ | _272.00_ | 539.00 | 383.60 | - | 409.20 | 448.40 | 529.00 | 384.80 | 380.40 | 715.40 | **18.00** |
|  | TPR↑ | _0.54_ | 0.14 | 0.51 | - | 0.11 | 0.07 | 0.11 | 0.04 | 0.09 | 0.15 | **0.96** |
|  | FDR↓ | _0.37_ | 0.83 | 0.54 | - | 0.62 | 0.77 | 0.75 | 0.41 | 0.46 | 0.89 | **0.00** |
| 150/4 | SHD↓ | _438.80_ | 877.20 | - | - | 634.00 | 626.40 | 842.20 | 620.00 | 579.20 | 1128.00 | **44.20** |
|  | TPR↑ | _0.53_ | 0.12 | - | - | 0.10 | 0.03 | 0.08 | 0.02 | 0.07 | 0.14 | **0.93** |
|  | FDR↓ | _0.38_ | 0.86 | - | - | 0.66 | 0.78 | 0.86 | 0.57 | 0.46 | 0.90 | **0.00** |
| 1000/4 | SHD↓ | _3269.80_ | - | - | - | 4049.00 | 4037.00 | - | 5565.00 | - | 6750.40 | **1531.00** |
|  | TPR↑ | _0.49_ | - | - | - | 0.12 | 0.07 | - | 0.03 | - | 0.13 | **0.62** |
|  | FDR↓ | _0.43_ | - | - | - | 0.53 | 0.55 | - | 0.93 | - | 0.87 | **0.01** |

Table 7: Average SHD, TPR, and FDR coming from 5 seeds. Each dataset comprises 1000 observations generated by ANMs with function $f_2$ and Gaussian variation. The causal graphs are generated by SF model with varying average degrees (e) and variable numbers (d). Bold and underlined font represent the first and second best results in terms of SHD, respectively.

| d/e | Metrics | CAM | SAM | DiffAN | CORL | NOTEARS | GOLEM | GraN-DAG | GAE | DAG-GNN | SDCD | OURS |
|---|---|---|---|---|---|---|---|---|---|---|---|---|
| 10/1 | SHD↓ | 6.60 | 16.00 | 3.00 | 8.80 | 9.40 | 16.40 | 2.60 | 5.00 | 9.00 | 13.40 | **2.00** |
|  | TPR↑ | 0.36 | 0.41 | 0.73 | 0.21 | 0.15 | 0.20 | 0.88 | 0.57 | 0.17 | 0.15 | **1.00** |
|  | FDR↓ | 0.56 | 0.81 | 0.22 | 0.39 | 0.47 | 0.85 | 0.14 | 0.07 | 0.42 | 0.90 | **0.15** |
| 20/1 | SHD↓ | 6.00 | 34.25 | 3.25 | 21.00 | 21.75 | 44.25 | 11.25 | 13.25 | 21.00 | 36.25 | **0.75** |
|  | TPR↑ | 0.73 | 0.14 | 0.84 | 0.16 | 0.08 | 0.10 | 0.52 | 0.35 | 0.08 | 0.03 | **1.00** |
|  | FDR↓ | 0.25 | 0.94 | 0.12 | 0.70 | 0.82 | 0.94 | 0.21 | 0.10 | 0.80 | 0.99 | **0.04** |
| 50/1 | SHD↓ | 31.40 | 72.80 | 12.80 | 54.25 | 57.00 | 95.80 | 45.20 | 43.40 | 55.20 | 116.60 | **1.60** |
|  | TPR↑ | 0.54 | 0.17 | 0.88 | 0.07 | 0.08 | 0.03 | 0.20 | 0.13 | 0.04 | 0.08 | **1.00** |
|  | FDR↓ | 0.44 | 0.85 | 0.14 | 0.78 | 0.78 | 0.96 | 0.36 | 0.14 | 0.83 | 0.96 | **0.03** |
| 100/1 | SHD↓ | 61.50 | 130.25 | 42.75 | - | 120.75 | 173.75 | 248.00 | 100.00 | 111.25 | 210.25 | **1.50** |
|  | TPR↑ | 0.57 | 0.14 | 0.88 | - | 0.05 | 0.03 | 0.55 | 0.08 | 0.03 | 0.08 | **0.99** |
|  | FDR↓ | 0.43 | 0.83 | 0.28 | - | 0.85 | 0.97 | 0.77 | 0.31 | 0.85 | 0.96 | **0.01** |
| 150/1 | SHD↓ | 99.40 | 195.25 | - | - | 176.20 | 223.00 | 490.40 | 137.20 | 168.80 | 305.40 | **2.20** |
|  | TPR↑ | 0.57 | 0.11 | - | - | 0.07 | 0.02 | 0.43 | 0.13 | 0.05 | 0.06 | **0.99** |
|  | FDR↓ | 0.45 | 0.84 | - | - | 0.79 | 0.98 | 0.86 | 0.17 | 0.81 | 0.96 | **0.00** |
| 1000/1 | SHD↓ | 991.00 | - | - | - | 1186.00 | 1149.00 | - | 3541.00 | - | 1479.50 | **137.50** |
|  | TPR↑ | 0.59 | - | - | - | 0.08 | 0.10 | - | 0.05 | - | 0.08 | **0.88** |
|  | FDR↓ | 0.57 | - | - | - | 0.78 | 0.72 | - | 0.99 | - | 0.93 | **0.01** |
| 10/4 | SHD↓ | 19.50 | 26.50 | 17.50 | 30.50 | 32.50 | 32.50 | 26.00 | 24.00 | 32.50 | 36.50 | **2.50** |
|  | TPR↑ | 0.48 | 0.35 | 0.57 | 0.21 | 0.15 | 0.18 | 0.32 | 0.31 | 0.15 | 0.14 | **0.98** |
|  | FDR↓ | 0.18 | 0.53 | 0.23 | 0.34 | 0.39 | 0.44 | 0.12 | 0.29 | 0.39 | 0.80 | **0.05** |
| 20/4 | SHD↓ | 36.00 | 81.00 | 17.00 | 72.00 | 72.67 | 79.00 | 65.67 | 42.67 | 71.00 | 102.00 | **1.33** |
|  | TPR↑ | 0.55 | 0.18 | 0.79 | 0.09 | 0.06 | 0.11 | 0.06 | 0.39 | 0.02 | 0.08 | **1.00** |
|  | FDR↓ | 0.23 | 0.78 | 0.07 | 0.61 | 0.66 | 0.71 | 0.22 | 0.00 | 0.79 | 0.93 | **0.02** |
| 50/4 | SHD↓ | 111.50 | 262.50 | 94.00 | 234.00 | 210.50 | 215.50 | 196.50 | 205.00 | 208.00 | 330.00 | **65.00** |
|  | TPR↑ | 0.56 | 0.12 | 0.67 | 0.04 | 0.04 | 0.04 | 0.00 | 0.03 | 0.03 | 0.09 | **0.66** |
|  | FDR↓ | 0.25 | 0.84 | 0.25 | 0.80 | 0.77 | 0.80 | 0.90 | 0.75 | 0.78 | 0.92 | **0.04** |
| 100/4 | SHD↓ | 223.67 | 533.67 | 289.67 | - | 421.33 | 422.00 | 514.67 | 427.00 | 400.33 | 668.67 | **13.33** |
|  | TPR↑ | 0.57 | 0.08 | 0.61 | - | 0.02 | 0.01 | 0.05 | 0.05 | 0.01 | 0.06 | **0.97** |
|  | FDR↓ | 0.26 | 0.88 | 0.40 | - | 0.84 | 0.92 | 0.89 | 0.56 | 0.85 | 0.94 | **0.00** |
| 150/4 | SHD↓ | 322.25 | 852.75 | - | - | 653.50 | 655.50 | 1030.00 | 636.25 | 617.25 | 1103.00 | **38.75** |
|  | TPR↑ | 0.59 | 0.08 | - | - | 0.02 | 0.01 | 0.04 | 0.03 | 0.01 | 0.07 | **0.94** |
|  | FDR↓ | 0.23 | 0.88 | - | - | 0.85 | 0.90 | 0.95 | 0.75 | 0.82 | 0.94 | **0.00** |
| 1000/4 | SHD↓ | 2085.25 | - | - | - | 4289.00 | 4036.00 | - | 7474.00 | - | 6446.25 | **1145.75** |
|  | TPR↑ | 0.62 | - | - | - | 0.01 | 0.00 | - | 0.03 | - | 0.02 | **0.71** |
|  | FDR↓ | 0.23 | - | - | - | 0.92 | 0.89 | - | 0.97 | - | 0.98 | **0.00** |

Table 8: Average SHD, TPR, and FDR coming from 5 seeds. Each dataset comprises 1000 observations generated by ANMs with function $f_2$ and Gumbel variation. The causal graphs are generated by ER model with varying average degrees (e) and variable numbers (d). Bold and underlined font represent the first and second best results in terms of SHD, respectively.

| d/e | Metrics | CAM | SAM | DiffAN | CORL | NOTEARS | GOLEM | GraN-DAG | GAE | DAG-GNN | SDCD | OURS |
|---|---|---|---|---|---|---|---|---|---|---|---|---|
| 10/1 | SHD↓ | 6.00 | 16.67 | 3.67 | 11.00 | 11.00 | 19.67 | 3.67 | 6.00 | 11.33 | 18.00 | **1.33** |
|  | TPR↑ | 0.59 | 0.52 | 0.67 | 0.11 | 0.11 | 0.11 | 0.74 | 0.56 | 0.07 | 0.04 | **1.00** |
|  | FDR↓ | 0.45 | 0.77 | 0.31 | 0.74 | 0.74 | 0.92 | 0.18 | 0.39 | 0.82 | 0.98 | **0.10** |
| 20/1 | SHD↓ | 14.20 | 34.40 | 7.20 | 22.40 | 22.40 | 43.80 | 11.00 | 9.40 | 22.00 | 42.00 | **5.80** |
|  | TPR↑ | 0.45 | 0.08 | 0.68 | 0.09 | 0.06 | 0.05 | 0.48 | 0.56 | 0.06 | 0.01 | **0.91** |
|  | FDR↓ | 0.41 | 0.94 | 0.19 | 0.72 | 0.76 | 0.97 | 0.12 | 0.10 | 0.76 | 0.99 | **0.19** |
| 50/1 | SHD↓ | 43.75 | 94.75 | 10.50 | 62.00 | 62.25 | 92.50 | 45.50 | 42.50 | 58.75 | 139.50 | **7.25** |
|  | TPR↑ | 0.47 | 0.08 | 0.89 | 0.04 | 0.03 | 0.02 | 0.15 | 0.16 | 0.03 | 0.03 | **0.98** |
|  | FDR↓ | 0.53 | 0.94 | 0.13 | 0.88 | 0.92 | 0.98 | 0.36 | 0.14 | 0.89 | 0.99 | **0.11** |
| 100/1 | SHD↓ | 92.40 | 166.60 | 71.40 | - | 126.60 | 164.20 | 332.60 | 94.20 | 115.80 | 236.20 | **26.40** |
|  | TPR↑ | 0.45 | 0.05 | 0.77 | - | 0.03 | 0.00 | 0.38 | 0.06 | 0.01 | 0.04 | **0.87** |
|  | FDR↓ | 0.55 | 0.95 | 0.41 | - | 0.90 | 0.99 | 0.87 | 0.08 | 0.95 | 0.98 | **0.14** |
| 150/1 | SHD↓ | 153.50 | 286.00 | - | - | 195.00 | 219.75 | 325.25 | 149.25 | 173.25 | 364.75 | **13.25** |
|  | TPR↑ | 0.48 | 0.04 | - | - | 0.05 | 0.00 | 0.32 | 0.03 | 0.01 | 0.04 | **0.97** |
|  | FDR↓ | 0.58 | 0.96 | - | - | 0.88 | 0.99 | 0.83 | 0.28 | 0.93 | 0.98 | **0.06** |
| 1000/1 | SHD↓ | 1334.75 | - | - | - | 1227.00 | 1121.00 | - | 3171.00 | - | 1826.00 | **291.50** |
|  | TPR↑ | 0.46 | - | - | - | 0.05 | 0.04 | - | 0.07 | - | 0.05 | **0.80** |
|  | FDR↓ | 0.68 | - | - | - | 0.85 | 0.81 | - | 0.94 | - | 0.96 | **0.10** |
| 10/4 | SHD↓ | 17.50 | 23.50 | 14.50 | 19.50 | 21.00 | 30.50 | 10.00 | 22.00 | 23.00 | 24.50 | **3.00** |
|  | TPR↑ | 0.33 | 0.35 | 0.48 | 0.23 | 0.17 | 0.12 | 0.67 | 0.08 | 0.08 | 0.21 | **0.98** |
|  | FDR↓ | 0.58 | 0.70 | 0.45 | 0.14 | 0.14 | 0.77 | 0.11 | 0.00 | 0.25 | 0.80 | **0.10** |
| 20/4 | SHD↓ | 40.50 | 73.25 | 21.75 | 69.00 | 70.50 | 77.75 | 61.00 | 39.75 | 67.00 | 90.50 | **18.25** |
|  | TPR↑ | 0.52 | 0.21 | 0.73 | 0.08 | 0.06 | 0.09 | 0.07 | 0.39 | 0.05 | 0.14 | **0.75** |
|  | FDR↓ | 0.34 | 0.74 | 0.18 | 0.66 | 0.73 | 0.79 | 0.24 | 0.01 | 0.65 | 0.86 | **0.20** |
| 50/4 | SHD↓ | 125.00 | 252.40 | 78.00 | 200.00 | 200.80 | 214.40 | 194.00 | 141.00 | 192.20 | 307.40 | **46.20** |
|  | TPR↑ | 0.51 | 0.10 | 0.72 | 0.03 | 0.02 | 0.02 | 0.00 | 0.28 | 0.01 | 0.08 | **0.77** |
|  | FDR↓ | 0.33 | 0.86 | 0.20 | 0.80 | 0.85 | 0.90 | 0.92 | 0.34 | 0.84 | 0.93 | **0.18** |
| 100/4 | SHD↓ | 314.60 | 568.60 | 366.20 | - | 429.40 | 437.80 | 581.40 | 448.20 | 405.80 | 705.40 | **16.20** |
|  | TPR↑ | 0.47 | 0.05 | 0.54 | - | 0.02 | 0.01 | 0.05 | 0.03 | 0.01 | 0.07 | **0.96** |
|  | FDR↓ | 0.41 | 0.93 | 0.50 | - | 0.88 | 0.95 | 0.91 | 0.50 | 0.89 | 0.94 | **0.00** |
| 150/4 | SHD↓ | 517.75 | 900.33 | - | - | 654.75 | 659.50 | 940.25 | 597.00 | 610.50 | 1097.00 | **40.50** |
|  | TPR↑ | 0.44 | 0.05 | - | - | 0.01 | 0.01 | 0.04 | 0.02 | 0.01 | 0.05 | **0.94** |
|  | FDR↓ | 0.45 | 0.93 | - | - | 0.91 | 0.95 | 0.94 | 0.62 | 0.90 | 0.95 | **0.00** |
| 1000/4 | SHD↓ | 3741.25 | - | - | - | - | 4359.00 | - | 5196.00 | - | 7113.50 | **1368.25** |
|  | TPR↑ | 0.42 | - | - | - | - | 0.01 | - | 0.03 | - | 0.02 | **0.66** |
|  | FDR↓ | 0.48 | - | - | - | - | 0.89 | - | 0.91 | - | 0.98 | **0.00** |

Table 9: Average SHD, TPR, and FDR coming from 5 seeds. Each dataset comprises 1000 observations generated by ANMs with function $f_2$ and Gumbel variation. The causal graphs are generated by SF model with varying average degrees (e) and variable numbers (d). Bold and underlined font represent the first and second best results in terms of SHD, respectively.

| d/e | Metrics | CAM | SAM | DiffAN | CORL | NOTEARS | GOLEM | GraN-DAG | GAE | DAG-GNN | SDCD | OURS |
|---|---|---|---|---|---|---|---|---|---|---|---|---|
| 10/1 | SHD↓ | 4.2 | 12.00 | 5.00 | 3.00 | 3.40 | 2.80 | _2.40_ | 2.60 | 4.80 | 4.80 | **0.80** |
|  | TPR↑ | 0.58 | 0.81 | 0.62 | 0.67 | 0.62 | 0.67 | _0.77_ | 0.80 | 0.49 | 0.58 | **0.90** |
|  | FDR↓ | 0.49 | 0.67 | 0.50 | 0.06 | 0.04 | 0.05 | _0.07_ | 0.09 | 0.25 | 0.45 | **0.03** |
| 20/1 | SHD↓ | 8.0 | 13.20 | 19.00 | 7.20 | 10.00 | 7.60 | _5.20_ | 5.80 | 10.60 | 9.20 | **1.20** |
|  | TPR↑ | 0.72 | 0.70 | 0.64 | 0.72 | 0.55 | 0.69 | _0.76_ | 0.73 | 0.48 | 0.59 | **0.94** |
|  | FDR↓ | 0.36 | 0.47 | 0.53 | 0.11 | 0.09 | 0.11 | _0.01_ | 0.00 | 0.04 | 0.36 | **0.00** |
| 50/1 | SHD↓ | 18.6 | 23.20 | 24.00 | 21.60 | 19.80 | 15.60 | 15.60 | 14.40 | 23.20 | _11.40_ | **0.00** |
|  | TPR↑ | 0.73 | 0.64 | 0.79 | 0.61 | 0.60 | 0.72 | 0.67 | 0.69 | 0.51 | _0.79_ | **1.00** |
|  | FDR↓ | 0.35 | 0.43 | 0.39 | 0.20 | 0.05 | 0.08 | 0.01 | 0.00 | 0.03 | _0.16_ | **0.00** |
| 100/1 | SHD↓ | 45.6 | 43.80 | 52.25 | - | 39.80 | 31.80 | _24.60_ | 51.60 | 52.80 | 32.60 | **0.20** |
|  | TPR↑ | 0.73 | 0.64 | 0.83 | - | 0.62 | 0.71 | _0.74_ | 0.53 | 0.46 | 0.73 | **1.00** |
|  | FDR↓ | 0.4 | 0.38 | 0.40 | - | 0.05 | 0.06 | _0.00_ | 0.06 | 0.02 | 0.24 | **0.00** |
| 150/1 | SHD↓ | 71.0 | 64.00 | - | - | 61.40 | 46.40 | _40.40_ | 128.00 | 92.80 | 57.20 | **0.00** |
|  | TPR↑ | 0.74 | 0.63 | - | - | 0.59 | 0.70 | _0.72_ | 0.29 | 0.35 | 0.73 | **1.00** |
|  | FDR↓ | 0.41 | 0.39 | - | - | 0.05 | 0.05 | _0.01_ | 0.53 | 0.01 | 0.31 | **0.00** |
| 1000/1 | SHD↓ | 599.2 | - | - | - | 442.50 | _337.00_ | - | 824.00 | - | 560.60 | **91.40** |
|  | TPR↑ | 0.76 | - | - | - | 0.59 | _0.69_ | - | 0.19 | - | 0.85 | **0.98** |
|  | FDR↓ | 0.44 | - | - | - | 0.05 | _0.04_ | - | 0.07 | - | 0.38 | **0.08** |
| 10/4 | SHD↓ | 20.2 | 20.80 | 12.40 | 22.60 | 26.20 | 24.40 | 12.60 | _3.20_ | 24.80 | 29.40 | **0.00** |
|  | TPR↑ | 0.49 | 0.50 | 0.75 | 0.43 | 0.30 | 0.41 | 0.65 | _0.91_ | 0.35 | 0.24 | **1.00** |
|  | FDR↓ | 0.42 | 0.37 | 0.32 | 0.18 | 0.17 | 0.23 | 0.02 | _0.00_ | 0.14 | 0.63 | **0.00** |
| 20/4 | SHD↓ | 41.4 | 51.20 | 45.00 | 60.80 | 60.20 | 72.80 | 39.20 | _15.40_ | 59.40 | 70.80 | **0.00** |
|  | TPR↑ | 0.56 | 0.58 | 0.74 | 0.34 | 0.29 | 0.25 | 0.47 | _0.80_ | 0.29 | 0.21 | **1.00** |
|  | FDR↓ | 0.37 | 0.43 | 0.42 | 0.33 | 0.25 | 0.49 | 0.01 | _0.01_ | 0.22 | 0.70 | **0.00** |
| 50/4 | SHD↓ | 72.8 | 156.60 | 65.00 | 173.50 | 169.60 | 201.00 | 141.40 | _35.80_ | 166.00 | 179.00 | **0.00** |
|  | TPR↑ | 0.72 | 0.42 | 0.92 | 0.27 | 0.25 | 0.17 | 0.29 | _0.82_ | 0.21 | 0.22 | **1.00** |
|  | FDR↓ | 0.2 | 0.46 | 0.25 | 0.40 | 0.31 | 0.54 | 0.01 | _0.00_ | 0.22 | 0.58 | **0.00** |
| 100/4 | SHD↓ | 102.2 | 311.80 | 229.25 | - | 319.20 | 398.40 | 156.00 | _56.60_ | 359.80 | 347.00 | **0.00** |
|  | TPR↑ | 0.79 | 0.36 | 0.87 | - | 0.25 | 0.09 | 0.61 | _0.85_ | 0.09 | 0.21 | **1.00** |
|  | FDR↓ | 0.13 | 0.46 | 0.39 | - | 0.22 | 0.57 | 0.00 | _0.00_ | 0.07 | 0.55 | **0.00** |
| 150/4 | SHD↓ | 139.6 | 460.20 | - | - | 481.00 | 593.00 | 239.60 | _106.40_ | 590.00 | 522.60 | **0.00** |
|  | TPR↑ | 0.81 | 0.36 | - | - | 0.25 | 0.08 | 0.60 | _0.82_ | 0.00 | 0.22 | **1.00** |
|  | FDR↓ | 0.12 | 0.44 | - | - | 0.20 | 0.52 | 0.01 | _0.00_ | 0.04 | 0.55 | **0.00** |
| 1000/4 | SHD↓ | _844.40_ | - | - | - | 3197.75 | 3988.60 | - | 2349.00 | - | 4064.80 | **1.20** |
|  | TPR↑ | _0.83_ | - | - | - | 0.26 | 0.05 | - | 0.41 | - | 0.35 | **1.00** |
|  | FDR↓ | _0.10_ | - | - | - | 0.19 | 0.50 | - | 0.00 | - | 0.59 | **0.00** |

Table 10: Average SHD, TPR, and FDR coming from 5 seeds. Each dataset comprises 5000 observations generated by ANMs with function $f_1$ and Gaussian variation. The causal graphs are generated by ER model with varying average degrees (e) and variable numbers (d). Bold and underlined font represent the first and second best results in terms of SHD, respectively.

| d/e | Metrics | CAM | SAM | DiffAN | CORL | NOTEARS | GOLEM | GraN-DAG | GAE | DAG-GNN | SDCD | OURS |
|---|---|---|---|---|---|---|---|---|---|---|---|---|
| 10/1 | SHD↓ | 4.2 | 6.80 | 2.40 | 2.40 | 3.80 | 4.20 | _1.80_ | 3.60 | 3.80 | 4.40 | **0.60** |
|  | TPR↑ | 0.69 | 0.84 | 0.82 | 0.78 | 0.62 | 0.64 | _0.82_ | 0.64 | 0.62 | 0.58 | **0.93** |
|  | FDR↓ | 0.37 | 0.44 | 0.24 | 0.07 | 0.10 | 0.16 | _0.02_ | 0.10 | 0.07 | 0.45 | **0.00** |
| 20/1 | SHD↓ | 4.8 | 20.00 | 15.50 | 8.40 | 9.80 | 8.20 | 4.20 | _4.00_ | 9.40 | 11.40 | **2.20** |
|  | TPR↑ | 0.85 | 0.65 | 0.71 | 0.64 | 0.55 | 0.64 | 0.78 | _0.80_ | 0.56 | 0.58 | **0.88** |
|  | FDR↓ | 0.23 | 0.55 | 0.49 | 0.12 | 0.11 | 0.11 | 0.00 | _0.01_ | 0.10 | 0.43 | **0.00** |
| 50/1 | SHD↓ | 24.6 | 45.20 | 68.25 | 22.40 | 22.60 | 20.60 | 24.40 | _15.00_ | 23.80 | 28.20 | **3.00** |
|  | TPR↑ | 0.8 | 0.40 | 0.68 | 0.64 | 0.60 | 0.70 | 0.50 | _0.71_ | 0.55 | 0.58 | **0.94** |
|  | FDR↓ | 0.33 | 0.67 | 0.64 | 0.19 | 0.09 | 0.14 | 0.00 | _0.03_ | 0.07 | 0.37 | **0.00** |
| 100/1 | SHD↓ | 58.0 | 84.40 | 115.00 | - | 54.20 | _40.40_ | 40.40 | 73.80 | 61.20 | 62.20 | **1.40** |
|  | TPR↑ | 0.76 | 0.48 | 0.74 | - | 0.52 | _0.62_ | 0.59 | 0.29 | 0.41 | 0.55 | **0.99** |
|  | FDR↓ | 0.43 | 0.57 | 0.60 | - | 0.12 | _0.04_ | 0.00 | 0.06 | 0.06 | 0.42 | **0.01** |
| 150/1 | SHD↓ | 76.4 | 139.40 | - | - | 89.40 | 71.20 | _60.60_ | 103.00 | 99.60 | 128.20 | **1.80** |
|  | TPR↑ | 0.77 | 0.45 | - | - | 0.50 | 0.59 | _0.59_ | 0.31 | 0.34 | 0.49 | **1.00** |
|  | FDR↓ | 0.39 | 0.61 | - | - | 0.18 | 0.10 | _0.00_ | 0.03 | 0.03 | 0.56 | **0.01** |
| 1000/1 | SHD↓ | 899.4 | - | - | - | 536.50 | _422.80_ | - | 817.00 | - | 1103.00 | **215.00** |
|  | TPR↑ | 0.67 | - | - | - | 0.53 | _0.62_ | - | 0.18 | - | 0.58 | **0.92** |
|  | FDR↓ | 0.55 | - | - | - | 0.12 | _0.07_ | - | 0.01 | - | 0.61 | **0.17** |
| 10/4 | SHD↓ | 13.2 | 14.00 | 13.60 | 11.60 | 14.80 | 15.20 | _7.40_ | 8.80 | 14.20 | 18.40 | **0.20** |
|  | TPR↑ | 0.57 | 0.57 | 0.72 | 0.59 | 0.43 | 0.53 | _0.70_ | 0.70 | 0.46 | 0.34 | **0.99** |
|  | FDR↓ | 0.41 | 0.38 | 0.43 | 0.12 | 0.12 | 0.25 | _0.01_ | 0.10 | 0.15 | 0.56 | **0.00** |
| 20/4 | SHD↓ | 33.6 | 48.20 | 44.00 | 48.60 | 49.40 | 64.80 | 29.40 | _15.60_ | 48.20 | 58.00 | **0.20** |
|  | TPR↑ | 0.64 | 0.57 | 0.74 | 0.39 | 0.33 | 0.24 | 0.56 | _0.76_ | 0.32 | 0.24 | **1.00** |
|  | FDR↓ | 0.36 | 0.49 | 0.46 | 0.31 | 0.24 | 0.54 | 0.03 | _0.01_ | 0.20 | 0.63 | **0.00** |
| 50/4 | SHD↓ | 96.8 | 160.00 | 133.25 | 163.75 | 158.00 | 195.20 | 146.60 | _71.00_ | 169.00 | 167.20 | **0.00** |
|  | TPR↑ | 0.65 | 0.41 | 0.85 | 0.26 | 0.24 | 0.11 | 0.20 | _0.62_ | 0.12 | 0.21 | **1.00** |
|  | FDR↓ | 0.31 | 0.50 | 0.43 | 0.40 | 0.32 | 0.61 | 0.00 | _0.01_ | 0.15 | 0.59 | **0.00** |
| 100/4 | SHD↓ | 210.6 | 356.40 | 491.75 | - | 324.40 | 397.40 | 211.00 | _138.80_ | 348.20 | 342.00 | **0.00** |
|  | TPR↑ | 0.62 | 0.29 | 0.79 | - | 0.25 | 0.11 | 0.45 | _0.64_ | 0.11 | 0.20 | **1.00** |
|  | FDR↓ | 0.29 | 0.55 | 0.60 | - | 0.28 | 0.59 | 0.01 | _0.01_ | 0.08 | 0.55 | **0.00** |
| 150/4 | SHD↓ | 333.4 | 605.20 | - | - | 514.00 | 588.00 | 333.40 | _120.80_ | 573.00 | 527.40 | **0.00** |
|  | TPR↑ | 0.64 | 0.26 | - | - | 0.21 | 0.08 | 0.43 | _0.80_ | 0.02 | 0.19 | **1.00** |
|  | FDR↓ | 0.3 | 0.62 | - | - | 0.30 | 0.53 | 0.02 | _0.00_ | 0.00 | 0.56 | **0.00** |
| 1000/4 | SHD↓ | _2591.60_ | - | - | - | 3254.33 | 3948.20 | - | 2943.00 | - | 4141.00 | **68.20** |
|  | TPR↑ | _0.61_ | - | - | - | 0.25 | 0.07 | - | 0.29 | - | 0.30 | **0.99** |
|  | FDR↓ | _0.35_ | - | - | - | 0.21 | 0.48 | - | 0.08 | - | 0.60 | **0.01** |

Table 11: Average SHD, TPR, and FDR coming from 5 seeds. Each dataset comprises 5000 observations generated by ANMs with function $f_1$ and Gaussian variation. The causal graphs are generated by SF model with varying average degrees (e) and variable numbers (d). Bold and underlined font represent the first and second best results in terms of SHD, respectively.

| d/e | Metrics | CAM | SAM | DiffAN | CORL | NOTEARS | GOLEM | GraN-DAG | GAE | DAG-GNN | SDCD | OURS |
|---|---|---|---|---|---|---|---|---|---|---|---|---|
| 10/1 | SHD↓ | 4.6 | 13.40 | 2.60 | 4.00 | 5.00 | 7.00 | _1.20_ | 3.40 | 5.20 | 3.40 | **0.60** |
|  | TPR↑ | 0.63 | 0.90 | 0.82 | 0.64 | 0.51 | 0.47 | _0.90_ | 0.68 | 0.51 | 0.73 | **0.94** |
|  | FDR↓ | 0.4 | 0.60 | 0.23 | 0.05 | 0.03 | 0.26 | _0.02_ | 0.04 | 0.04 | 0.21 | **0.00** |
| 20/1 | SHD↓ | 6.8 | 12.20 | 3.80 | 5.00 | 7.40 | 7.20 | 3.40 | 4.20 | 7.40 | _2.40_ | **1.80** |
|  | TPR↑ | 0.72 | 0.73 | 0.89 | 0.69 | 0.57 | 0.58 | 0.80 | 0.80 | 0.55 | _0.86_ | **0.90** |
|  | FDR↓ | 0.35 | 0.49 | 0.17 | 0.00 | 0.01 | 0.01 | 0.00 | 0.06 | 0.00 | _0.03_ | **0.00** |
| 50/1 | SHD↓ | 22.0 | 26.00 | _13.20_ | 25.00 | 25.80 | 23.60 | 15.80 | 23.60 | 29.00 | 14.80 | **5.40** |
|  | TPR↑ | 0.74 | 0.63 | _0.91_ | 0.53 | 0.50 | 0.54 | 0.68 | 0.52 | 0.43 | 0.75 | **0.89** |
|  | FDR↓ | 0.38 | 0.44 | _0.22_ | 0.11 | 0.04 | 0.03 | 0.01 | 0.00 | 0.04 | 0.15 | **0.00** |
| 100/1 | SHD↓ | 36.2 | 44.00 | 31.20 | - | 49.60 | 43.60 | _22.80_ | 43.80 | 74.60 | 27.00 | **2.00** |
|  | TPR↑ | 0.77 | 0.62 | 0.90 | - | 0.49 | 0.55 | _0.76_ | 0.51 | 0.22 | 0.77 | **0.97** |
|  | FDR↓ | 0.33 | 0.39 | 0.26 | - | 0.03 | 0.02 | _0.00_ | 0.07 | 0.00 | 0.09 | **0.00** |
| 150/1 | SHD↓ | 71.0 | 64.80 | - | - | 78.40 | 68.00 | 40.80 | 36.20 | 110.20 | _33.20_ | **2.60** |
|  | TPR↑ | 0.74 | 0.61 | - | - | 0.47 | 0.53 | 0.71 | 0.75 | 0.22 | _0.82_ | **1.00** |
|  | FDR↓ | 0.4 | 0.39 | - | - | 0.04 | 0.02 | 0.00 | 0.01 | 0.00 | _0.09_ | **0.02** |
| 1000/1 | SHD↓ | 665.6 | - | - | - | 551.50 | 462.00 | - | 699.50 | - | _449.20_ | **112.80** |
|  | TPR↑ | 0.77 | - | - | - | 0.47 | 0.55 | - | 0.43 | - | _0.90_ | **0.97** |
|  | FDR↓ | 0.46 | - | - | - | 0.05 | 0.02 | - | 0.29 | - | _0.29_ | **0.10** |
| 10/4 | SHD↓ | 17.0 | 21.20 | _5.00_ | 26.25 | 28.20 | 26.60 | 13.40 | 11.60 | 28.60 | 30.60 | **1.40** |
|  | TPR↑ | 0.59 | 0.49 | _0.89_ | 0.36 | 0.29 | 0.34 | 0.65 | 0.69 | 0.27 | 0.22 | **0.96** |
|  | FDR↓ | 0.27 | 0.27 | _0.09_ | 0.15 | 0.10 | 0.16 | 0.00 | 0.00 | 0.09 | 0.54 | **0.00** |
| 20/4 | SHD↓ | 48.6 | 48.40 | _13.20_ | 58.80 | 61.40 | 75.20 | 38.60 | 30.60 | 61.20 | 66.40 | **2.20** |
|  | TPR↑ | 0.5 | 0.55 | _0.90_ | 0.29 | 0.26 | 0.18 | 0.55 | 0.60 | 0.23 | 0.23 | **0.97** |
|  | FDR↓ | 0.43 | 0.36 | _0.13_ | 0.18 | 0.19 | 0.52 | 0.09 | 0.00 | 0.12 | 0.48 | **0.00** |
| 50/4 | SHD↓ | 82.4 | 152.60 | _50.20_ | 190.25 | 181.60 | 213.20 | 166.40 | 54.20 | 186.40 | 183.00 | **0.00** |
|  | TPR↑ | 0.69 | 0.40 | _0.90_ | 0.13 | 0.15 | 0.04 | 0.19 | 0.73 | 0.09 | 0.17 | **1.00** |
|  | FDR↓ | 0.22 | 0.41 | _0.16_ | 0.35 | 0.26 | 0.73 | 0.06 | 0.00 | 0.14 | 0.45 | **0.00** |
| 100/4 | SHD↓ | 128.8 | 294.80 | 165.60 | - | 338.40 | 419.20 | 196.40 | _115.20_ | 379.20 | 337.40 | **0.00** |
|  | TPR↑ | 0.74 | 0.36 | 0.87 | - | 0.17 | 0.03 | 0.54 | _0.71_ | 0.04 | 0.23 | **1.00** |
|  | FDR↓ | 0.17 | 0.40 | 0.27 | - | 0.16 | 0.77 | 0.07 | _0.00_ | 0.04 | 0.35 | **0.00** |
| 150/4 | SHD↓ | 173.6 | 464.40 | - | - | 510.40 | 620.60 | 341.80 | _109.40_ | 595.80 | 524.80 | **0.00** |
|  | TPR↑ | 0.77 | 0.32 | - | - | 0.17 | 0.04 | 0.43 | _0.82_ | 0.00 | 0.23 | **1.00** |
|  | FDR↓ | 0.14 | 0.46 | - | - | 0.15 | 0.68 | 0.02 | _0.00_ | 0.00 | 0.38 | **0.00** |
| 1000/4 | SHD↓ | _947.60_ | - | - | - | 3410.50 | 3969.00 | - | 3983.00 | - | 3908.60 | **1.60** |
|  | TPR↑ | _0.80_ | - | - | - | 0.17 | 0.05 | - | 0.03 | - | 0.35 | **1.00** |
|  | FDR↓ | _0.10_ | - | - | - | 0.12 | 0.41 | - | 0.42 | - | 0.50 | **0.00** |

Table 12: Average SHD, TPR, and FDR coming from 5 seeds. Each dataset comprises 5000 observations generated by ANMs with function $f_1$ and Gumbel variation. The causal graphs are generated by ER model with varying average degrees (e) and variable numbers (d). Bold and underlined font represent the first and second best results in terms of SHD, respectively.

| d/e | Metrics | CAM | SAM | DiffAN | CORL | NOTEARS | GOLEM | GraN-DAG | GAE | DAG-GNN | SDCD | OURS |
|---|---|---|---|---|---|---|---|---|---|---|---|---|
| 10/1 | SHD↓ | 3.0 | 9.20 | 2.40 | 4.75 | 5.80 | 6.60 | _1.60_ | 3.00 | 6.20 | 1.80 | **0.80** |
|  | TPR↑ | 0.78 | 0.84 | 0.89 | 0.50 | 0.38 | 0.38 | _0.84_ | 0.69 | 0.31 | 0.89 | **0.91** |
|  | FDR↓ | 0.26 | 0.53 | 0.21 | 0.05 | 0.05 | 0.30 | _0.03_ | 0.02 | 0.10 | 0.15 | **0.00** |
| 20/1 | SHD↓ | 6.8 | 18.00 | _3.00_ | 7.75 | 10.00 | 9.80 | 3.60 | 7.80 | 9.80 | 6.60 | **2.80** |
|  | TPR↑ | 0.79 | 0.69 | _0.92_ | 0.61 | 0.48 | 0.51 | 0.81 | 0.60 | 0.49 | 0.71 | **0.85** |
|  | FDR↓ | 0.3 | 0.56 | _0.12_ | 0.02 | 0.01 | 0.03 | 0.00 | 0.05 | 0.04 | 0.15 | **0.00** |
| 50/1 | SHD↓ | 19.2 | 37.20 | _13.00_ | 26.67 | 29.40 | 28.00 | 20.80 | 14.40 | 34.20 | 22.00 | **5.60** |
|  | TPR↑ | 0.79 | 0.48 | _0.90_ | 0.49 | 0.43 | 0.44 | 0.58 | 0.71 | 0.30 | 0.65 | **0.89** |
|  | FDR↓ | 0.33 | 0.59 | _0.19_ | 0.11 | 0.06 | 0.04 | 0.00 | 0.01 | 0.00 | 0.24 | **0.00** |
| 100/1 | SHD↓ | 80.8 | 77.60 | 83.40 | - | 56.00 | 49.00 | 39.40 | _36.60_ | 80.40 | 36.80 | **12.60** |
|  | TPR↑ | 0.67 | 0.46 | 0.83 | - | 0.45 | 0.51 | 0.60 | _0.65_ | 0.19 | 0.70 | **1.00** |
|  | FDR↓ | 0.5 | 0.57 | 0.45 | - | 0.04 | 0.02 | 0.00 | _0.03_ | 0.00 | 0.13 | **0.11** |
| 150/1 | SHD↓ | 98.2 | 136.40 | - | - | 89.40 | 79.80 | 68.20 | _38.80_ | 135.60 | 76.20 | **12.40** |
|  | TPR↑ | 0.72 | 0.38 | - | - | 0.44 | 0.48 | 0.54 | _0.74_ | 0.09 | 0.67 | **1.00** |
|  | FDR↓ | 0.45 | 0.64 | - | - | 0.08 | 0.03 | 0.00 | _0.00_ | 0.00 | 0.23 | **0.07** |
| 1000/1 | SHD↓ | 995.8 | - | - | - | 596.50 | _522.00_ | - | 718.00 | - | 795.60 | **159.80** |
|  | TPR↑ | 0.67 | - | - | - | 0.43 | _0.50_ | - | 0.28 | - | 0.66 | **0.96** |
|  | FDR↓ | 0.57 | - | - | - | 0.06 | _0.04_ | - | 0.00 | - | 0.44 | **0.12** |
| 10/4 | SHD↓ | 15.6 | 11.00 | _5.20_ | 15.00 | 16.00 | 17.00 | 5.60 | 12.40 | 16.20 | 19.20 | **1.20** |
|  | TPR↑ | 0.5 | 0.68 | _0.86_ | 0.42 | 0.35 | 0.40 | 0.78 | 0.56 | 0.36 | 0.29 | **0.95** |
|  | FDR↓ | 0.52 | 0.27 | _0.17_ | 0.09 | 0.06 | 0.22 | 0.02 | 0.09 | 0.08 | 0.50 | **0.00** |
| 20/4 | SHD↓ | 37.2 | 44.80 | _16.20_ | 49.40 | 54.60 | 63.40 | 30.20 | 25.80 | 53.00 | 53.40 | **0.80** |
|  | TPR↑ | 0.57 | 0.57 | _0.88_ | 0.31 | 0.23 | 0.24 | 0.58 | 0.62 | 0.19 | 0.29 | **0.99** |
|  | FDR↓ | 0.37 | 0.44 | _0.20_ | 0.21 | 0.27 | 0.51 | 0.09 | 0.04 | 0.07 | 0.41 | **0.00** |
| 50/4 | SHD↓ | 99.4 | 143.20 | _87.20_ | 148.00 | 151.80 | 194.40 | 147.60 | 98.20 | 161.20 | 151.60 | **0.00** |
|  | TPR↑ | 0.61 | 0.39 | _0.87_ | 0.24 | 0.22 | 0.09 | 0.20 | 0.47 | 0.15 | 0.24 | **1.00** |
|  | FDR↓ | 0.31 | 0.44 | _0.29_ | 0.20 | 0.19 | 0.66 | 0.04 | 0.01 | 0.13 | 0.33 | **0.00** |
| 100/4 | SHD↓ | 237.4 | 361.20 | 325.40 | - | 326.80 | 404.60 | 242.60 | _169.20_ | 376.00 | 327.40 | **0.00** |
|  | TPR↑ | 0.58 | 0.28 | 0.80 | - | 0.19 | 0.05 | 0.39 | _0.56_ | 0.02 | 0.22 | **1.00** |
|  | FDR↓ | 0.33 | 0.56 | 0.46 | - | 0.18 | 0.70 | 0.06 | _0.00_ | 0.01 | 0.35 | **0.00** |
| 150/4 | SHD↓ | 347.0 | 521.00 | - | - | 498.40 | 600.00 | 394.60 | _253.40_ | 583.40 | 512.60 | **0.00** |
|  | TPR↑ | 0.61 | 0.27 | - | - | 0.19 | 0.07 | 0.33 | _0.57_ | 0.00 | 0.23 | **1.00** |
|  | FDR↓ | 0.32 | 0.53 | - | - | 0.18 | 0.61 | 0.01 | _0.01_ | 0.00 | 0.38 | **0.00** |
| 1000/4 | SHD↓ | _2714.00_ | - | - | - | 3337.00 | 3881.00 | - | 3742.00 | - | 3935.80 | **53.40** |
|  | TPR↑ | _0.59_ | - | - | - | 0.19 | 0.08 | - | 0.11 | - | 0.30 | **0.99** |
|  | FDR↓ | _0.36_ | - | - | - | 0.13 | 0.42 | - | 0.31 | - | 0.52 | **0.01** |

Table 13: Average SHD, TPR, and FDR coming from 5 seeds. Each dataset comprises 5000 observations generated by ANMs with function $f_1$ and Gumbel variation. The causal graphs are generated by SF model with varying average degrees (e) and variable numbers (d). Bold and underlined font represent the first and second best results in terms of SHD, respectively.

| d/e | Metrics | CAM | SAM | DiffAN | CORL | NOTEARS | GOLEM | GraN-DAG | GAE | DAG-GNN | SDCD | OURS |
|---|---|---|---|---|---|---|---|---|---|---|---|---|
| 10/1 | SHD↓ | 5.60 | 16.40 | 10.00 | 4.20 | 4.60 | 12.00 | _1.00_ | 6.60 | 4.60 | 6.60 | **0.60** |
| | TPR↑ | 0.55 | 0.60 | 0.49 | 0.76 | 0.68 | 0.69 | _0.95_ | 0.37 | 0.66 | 0.56 | **1.00** |
| | FDR↓ | 0.51 | 0.76 | 0.63 | 0.18 | 0.19 | 0.60 | _0.05_ | 0.34 | 0.19 | 0.52 | **0.06** |
| 20/1 | SHD↓ | 9.40 | 25.80 | 25.60 | 17.60 | 16.40 | 30.20 | _4.60_ | 10.20 | 14.80 | 21.80 | **1.40** |
| | TPR↑ | 0.68 | 0.50 | 0.49 | 0.56 | 0.48 | 0.39 | _0.83_ | 0.51 | 0.46 | 0.40 | **1.00** |
| | FDR↓ | 0.36 | 0.68 | 0.67 | 0.38 | 0.32 | 0.68 | _0.05_ | 0.00 | 0.26 | 0.51 | **0.06** |
| 50/1 | SHD↓ | 21.40 | 33.60 | 35.60 | 34.80 | 32.20 | 73.40 | _16.20_ | 25.80 | 34.40 | 35.60 | **4.00** |
| | TPR↑ | 0.68 | 0.56 | 0.67 | 0.54 | 0.54 | 0.45 | _0.70_ | 0.44 | 0.39 | 0.57 | **0.99** |
| | FDR↓ | 0.41 | 0.52 | 0.53 | 0.38 | 0.33 | 0.71 | _0.08_ | 0.01 | 0.29 | 0.51 | **0.08** |
| 100/1 | SHD↓ | 45.20 | 84.40 | 70.00 | - | 74.00 | 141.00 | _10.40_ | 66.80 | 69.40 | 81.40 | **2.80** |
| | TPR↑ | 0.69 | 0.42 | 0.73 | - | 0.51 | 0.38 | _0.90_ | 0.29 | 0.41 | 0.57 | **0.99** |
| | FDR↓ | 0.40 | 0.61 | 0.50 | - | 0.37 | 0.70 | _0.02_ | 0.02 | 0.27 | 0.54 | **0.02** |
| 150/1 | SHD↓ | 84.00 | 137.40 | - | - | 119.60 | 203.60 | _20.60_ | 140.20 | 119.00 | 141.60 | **3.00** |
| | TPR↑ | 0.68 | 0.39 | - | - | 0.49 | 0.37 | _0.88_ | 0.09 | 0.32 | 0.55 | **0.99** |
| | FDR↓ | 0.43 | 0.61 | - | - | 0.36 | 0.66 | _0.02_ | 0.07 | 0.24 | 0.57 | **0.01** |
| 1000/1 | SHD↓ | _609.80_ | - | - | - | 778.00 | 1099.00 | - | 874.00 | - | 1289.40 | **121.80** |
| | TPR↑ | _0.71_ | - | - | - | 0.48 | 0.28 | - | 0.11 | - | 0.57 | **0.93** |
| | FDR↓ | _0.45_ | - | - | - | 0.37 | 0.60 | - | 0.03 | - | 0.66 | **0.05** |
| 10/4 | SHD↓ | _16.00_ | 27.60 | 20.20 | 28.40 | 29.60 | 28.40 | 16.20 | 24.20 | 30.00 | 30.60 | **6.80** |
| | TPR↑ | _0.60_ | 0.35 | 0.59 | 0.27 | 0.23 | 0.27 | 0.57 | 0.28 | 0.21 | 0.23 | **0.80** |
| | FDR↓ | _0.30_ | 0.54 | 0.46 | 0.33 | 0.37 | 0.34 | 0.09 | 0.06 | 0.36 | 0.66 | **0.11** |
| 20/4 | SHD↓ | _38.20_ | 75.40 | 71.20 | 72.60 | 73.20 | 83.00 | 61.20 | 69.60 | 75.00 | 86.40 | **1.00** |
| | TPR↑ | _0.59_ | 0.45 | 0.57 | 0.23 | 0.19 | 0.15 | 0.23 | 0.10 | 0.13 | 0.14 | **0.99** |
| | FDR↓ | _0.25_ | 0.59 | 0.58 | 0.44 | 0.43 | 0.63 | 0.11 | 0.03 | 0.41 | 0.76 | **0.00** |
| 50/4 | SHD↓ | _86.80_ | 225.60 | 141.80 | 188.40 | 192.60 | 213.60 | 181.80 | 134.80 | 185.20 | 237.60 | **2.80** |
| | TPR↑ | _0.67_ | 0.23 | 0.71 | 0.16 | 0.12 | 0.10 | 0.07 | 0.31 | 0.07 | 0.11 | **0.99** |
| | FDR↓ | _0.24_ | 0.69 | 0.47 | 0.46 | 0.49 | 0.66 | 0.12 | 0.02 | 0.23 | 0.82 | **0.00** |
| 100/4 | SHD↓ | _148.60_ | 445.40 | 282.00 | - | 389.40 | 414.00 | 276.20 | 287.00 | 380.40 | 476.80 | **12.20** |
| | TPR↑ | _0.71_ | 0.17 | 0.74 | - | 0.12 | 0.04 | 0.31 | 0.24 | 0.04 | 0.11 | **0.97** |
| | FDR↓ | _0.20_ | 0.72 | 0.46 | - | 0.53 | 0.74 | 0.08 | 0.00 | 0.32 | 0.82 | **0.00** |
| 150/4 | SHD↓ | _209.00_ | 687.60 | - | - | 593.40 | 629.80 | 441.20 | 481.80 | 579.80 | 733.00 | **30.80** |
| | TPR↑ | _0.73_ | 0.16 | - | - | 0.11 | 0.05 | 0.26 | 0.18 | 0.02 | 0.10 | **0.95** |
| | FDR↓ | _0.17_ | 0.72 | - | - | 0.54 | 0.74 | 0.07 | 0.00 | 0.23 | 0.83 | **0.00** |
| 1000/4 | SHD↓ | _1433.20_ | - | - | - | 4198.00 | 4116.00 | - | 3981.00 | - | 6337.40 | **724.60** |
| | TPR↑ | _0.73_ | - | - | - | 0.09 | 0.03 | - | 0.00 | - | 0.13 | **0.82** |
| | FDR↓ | _0.17_ | - | - | - | 0.62 | 0.68 | - | 0.75 | - | 0.87 | **0.00** |

Table 14: Average SHD, TPR, and FDR coming from 5 seeds. Each dataset comprises 5000 observations generated by ANMs with function $f_2$ and Gaussian variation. The causal graphs are generated by ER model with varying average degrees (e) and variable numbers (d). Bold and underlined font represent the first and second best results in terms of SHD, respectively.

| d/e | Metrics | CAM | SAM | DiffAN | CORL | NOTEARS | GOLEM | GraN-DAG | GAE | DAG-GNN | SDCD | OURS |
|---|---|---|---|---|---|---|---|---|---|---|---|---|
| 10/1 | SHD↓ | 7.00 | 17.20 | 13.40 | 7.60 | 7.20 | 12.60 | _2.20_ | 6.20 | 7.60 | 10.60 | **2.00** |
| | TPR↑ | 0.67 | 0.67 | 0.40 | 0.60 | 0.51 | 0.56 | _0.78_ | 0.38 | 0.44 | 0.44 | **0.84** |
| | FDR↓ | 0.47 | 0.72 | 0.75 | 0.40 | 0.31 | 0.63 | _0.03_ | 0.17 | 0.32 | 0.64 | **0.06** |
| 20/1 | SHD↓ | 12.60 | 33.40 | 20.80 | 17.60 | 18.00 | 32.00 | _6.20_ | 13.00 | 17.20 | 20.80 | **2.80** |
| | TPR↑ | 0.61 | 0.38 | 0.56 | 0.61 | 0.49 | 0.31 | _0.68_ | 0.39 | 0.47 | 0.46 | **1.00** |
| | FDR↓ | 0.51 | 0.82 | 0.64 | 0.47 | 0.48 | 0.77 | _0.02_ | 0.12 | 0.42 | 0.61 | **0.12** |
| 50/1 | SHD↓ | 29.40 | 67.20 | 46.80 | 41.00 | 44.00 | 79.00 | 30.60 | _24.40_ | 40.00 | 57.00 | **5.80** |
| | TPR↑ | 0.72 | 0.29 | 0.71 | 0.53 | 0.46 | 0.38 | 0.45 | _0.52_ | 0.47 | 0.41 | **1.00** |
| | FDR↓ | 0.42 | 0.79 | 0.56 | 0.42 | 0.44 | 0.72 | 0.14 | _0.02_ | 0.37 | 0.67 | **0.10** |
| 100/1 | SHD↓ | 73.00 | 119.40 | 119.80 | - | 87.40 | 150.40 | _35.80_ | 84.20 | 79.60 | 107.40 | **6.80** |
| | TPR↑ | 0.65 | 0.33 | 0.59 | - | 0.47 | 0.35 | _0.66_ | 0.16 | 0.36 | 0.43 | **0.98** |
| | FDR↓ | 0.50 | 0.70 | 0.65 | - | 0.43 | 0.71 | _0.03_ | 0.07 | 0.31 | 0.63 | **0.05** |
| 150/1 | SHD↓ | 143.20 | 193.00 | - | - | 127.00 | 197.40 | _54.00_ | 93.00 | 126.20 | 176.00 | **24.80** |
| | TPR↑ | 0.58 | 0.28 | - | - | 0.46 | 0.28 | _0.64_ | 0.39 | 0.26 | 0.40 | **0.98** |
| | FDR↓ | 0.58 | 0.75 | - | - | 0.41 | 0.69 | _0.00_ | 0.07 | 0.26 | 0.69 | **0.12** |
| 1000/1 | SHD↓ | 1111.00 | - | - | - | _897.00_ | 1119.00 | - | 942.00 | - | 1566.40 | **445.80** |
| | TPR↑ | 0.56 | - | - | - | _0.43_ | 0.18 | - | 0.06 | - | 0.44 | **0.83** |
| | FDR↓ | 0.63 | - | - | - | _0.43_ | 0.63 | - | 0.00 | - | 0.74 | **0.25** |
| 10/4 | SHD↓ | 15.20 | 20.80 | 19.40 | 19.60 | 20.80 | 24.20 | 10.80 | _9.20_ | 21.80 | 25.80 | **2.00** |
| | TPR↑ | 0.53 | 0.50 | 0.54 | 0.42 | 0.33 | 0.34 | 0.62 | _0.64_ | 0.30 | 0.24 | **0.98** |
| | FDR↓ | 0.47 | 0.58 | 0.57 | 0.40 | 0.42 | 0.54 | 0.08 | _0.04_ | 0.46 | 0.72 | **0.06** |
| 20/4 | SHD↓ | _37.80_ | 67.00 | 63.60 | 65.60 | 65.00 | 74.60 | 47.20 | 49.40 | 63.80 | 76.40 | **0.60** |
| | TPR↑ | _0.59_ | 0.46 | 0.54 | 0.21 | 0.17 | 0.15 | 0.28 | 0.23 | 0.16 | 0.18 | **0.99** |
| | FDR↓ | _0.36_ | 0.61 | 0.61 | 0.53 | 0.54 | 0.70 | 0.07 | 0.04 | 0.51 | 0.76 | **0.00** |
| 50/4 | SHD↓ | _121.00_ | 232.60 | 149.60 | 190.20 | 195.60 | 214.00 | 170.00 | 143.80 | 186.00 | 249.00 | **1.20** |
| | TPR↑ | _0.59_ | 0.21 | 0.73 | 0.15 | 0.12 | 0.07 | 0.09 | 0.22 | 0.06 | 0.08 | **0.99** |
| | FDR↓ | _0.36_ | 0.74 | 0.50 | 0.58 | 0.61 | 0.76 | 0.16 | 0.15 | 0.52 | 0.88 | **0.00** |
| 100/4 | SHD↓ | _265.40_ | 500.20 | 322.20 | - | 407.40 | 413.80 | 298.00 | 310.80 | 380.80 | 514.80 | **16.60** |
| | TPR↑ | _0.58_ | 0.13 | 0.75 | - | 0.09 | 0.04 | 0.26 | 0.19 | 0.02 | 0.10 | **0.96** |
| | FDR↓ | _0.37_ | 0.80 | 0.49 | - | 0.63 | 0.77 | 0.11 | 0.00 | 0.37 | 0.85 | **0.00** |
| 150/4 | SHD↓ | 403.40 | 748.40 | - | - | 597.20 | 637.40 | 469.00 | _384.00_ | 579.00 | 752.20 | **28.00** |
| | TPR↑ | 0.58 | 0.13 | - | - | 0.12 | 0.05 | 0.21 | _0.34_ | 0.01 | 0.10 | **0.95** |
| | FDR↓ | 0.37 | 0.79 | - | - | 0.54 | 0.75 | 0.05 | _0.00_ | 0.17 | 0.84 | **0.00** |
| 1000/4 | SHD↓ | _3068.60_ | - | - | - | 4053.00 | 4140.00 | - | 3986.00 | - | 6358.00 | **908.80** |
| | TPR↑ | _0.53_ | - | - | - | 0.12 | 0.04 | - | 0.00 | - | 0.12 | **0.78** |
| | FDR↓ | _0.41_ | - | - | - | 0.53 | 0.66 | - | 0.58 | - | 0.88 | **0.01** |

Table 15: Average SHD, TPR, and FDR coming from 5 seeds. Each dataset comprises 5000 observations generated by ANMs with function $f_2$ and Gaussian variation. The causal graphs are generated by SF model with varying average degrees (e) and variable numbers (d). Bold and underlined font represent the first and second best results in terms of SHD, respectively.

| d/e | Metrics | CAM | SAM | DiffAN | CORL | NOTEARS | GOLEM | GraN-DAG | GAE | DAG-GNN | SDCD | OURS |
|---|---|---|---|---|---|---|---|---|---|---|---|---|
| 10/1 | SHD↓ | 3.67 | 27.33 | 7.33 | 9.67 | 9.67 | 19.67 | 2.67 | 2.67 | 10.33 | 11.67 | **0.33** |
| | TPR↑ | 0.66 | 0.22 | 0.48 | 0.16 | 0.12 | 0.08 | 0.89 | 0.79 | 0.02 | 0.06 | **1.00** |
| | FDR↓ | 0.32 | 0.92 | 0.54 | 0.42 | 0.39 | 0.92 | 0.11 | 0.06 | 0.50 | 0.94 | **0.06** |
| 20/1 | SHD↓ | 11.67 | 31.33 | 7.00 | 20.33 | 20.33 | 39.67 | 4.67 | 10.33 | 18.67 | 26.33 | **0.33** |
| | TPR↑ | 0.54 | 0.04 | 0.82 | 0.05 | 0.05 | 0.04 | 0.82 | 0.38 | 0.04 | 0.00 | **1.00** |
| | FDR↓ | 0.55 | 0.98 | 0.26 | 0.83 | 0.83 | 0.97 | 0.08 | 0.00 | 0.83 | 1.00 | **0.02** |
| 50/1 | SHD↓ | 28.00 | 69.00 | 10.40 | 61.33 | 62.20 | 102.60 | 33.40 | 23.60 | 59.00 | 73.20 | **1.20** |
| | TPR↑ | 0.62 | 0.14 | 0.94 | 0.07 | 0.04 | 0.02 | 0.39 | 0.56 | 0.03 | 0.07 | **1.00** |
| | FDR↓ | 0.42 | 0.84 | 0.17 | 0.82 | 0.89 | 0.98 | 0.13 | 0.06 | 0.89 | 0.94 | **0.02** |
| 100/1 | SHD↓ | 52.00 | 127.60 | 23.20 | - | 116.00 | 186.80 | 23.60 | 71.00 | 106.20 | 139.80 | **0.80** |
| | TPR↑ | 0.65 | 0.08 | 0.93 | - | 0.05 | 0.02 | 0.86 | 0.26 | 0.01 | 0.08 | **0.99** |
| | FDR↓ | 0.42 | 0.87 | 0.19 | - | 0.82 | 0.98 | 0.10 | 0.01 | 0.91 | 0.93 | **0.00** |
| 150/1 | SHD↓ | 90.20 | 209.00 | - | - | 186.80 | 282.60 | 33.80 | 132.60 | 172.00 | 226.60 | **1.80** |
| | TPR↑ | 0.65 | 0.06 | - | - | 0.06 | 0.03 | 0.82 | 0.15 | 0.02 | 0.13 | **0.99** |
| | FDR↓ | 0.44 | 0.91 | - | - | 0.83 | 0.97 | 0.05 | 0.17 | 0.85 | 0.90 | **0.00** |
| 1000/1 | SHD↓ | 731.60 | - | - | - | - | 1080.50 | - | 948.00 | - | 1531.40 | **112.00** |
| | TPR↑ | 0.66 | - | - | - | - | 0.14 | - | 0.03 | - | 0.16 | **0.91** |
| | FDR↓ | 0.50 | - | - | - | - | 0.66 | - | 0.16 | - | 0.88 | **0.03** |
| 10/4 | SHD↓ | 16.75 | 28.50 | 12.50 | 28.25 | 29.00 | 28.00 | 14.50 | 18.25 | 30.50 | 34.00 | **7.75** |
| | TPR↑ | 0.53 | 0.30 | 0.71 | 0.22 | 0.19 | 0.24 | 0.60 | 0.50 | 0.15 | 0.13 | **0.78** |
| | FDR↓ | 0.25 | 0.56 | 0.29 | 0.20 | 0.18 | 0.21 | 0.04 | 0.09 | 0.24 | 0.77 | **0.01** |
| 20/4 | SHD↓ | 37.00 | 92.50 | 24.00 | 79.50 | 79.50 | 88.50 | 67.00 | 52.50 | 79.00 | 90.50 | **1.50** |
| | TPR↑ | 0.63 | 0.26 | 0.83 | 0.09 | 0.06 | 0.07 | 0.14 | 0.32 | 0.04 | 0.12 | **1.00** |
| | FDR↓ | 0.23 | 0.76 | 0.23 | 0.62 | 0.68 | 0.79 | 0.15 | 0.00 | 0.71 | 0.81 | **0.02** |
| 50/4 | SHD↓ | 97.50 | 259.25 | 57.75 | 224.00 | 215.00 | 219.75 | 195.75 | 151.75 | 202.75 | 242.75 | **0.50** |
| | TPR↑ | 0.64 | 0.12 | 0.86 | 0.03 | 0.03 | 0.01 | 0.02 | 0.23 | 0.01 | 0.03 | **1.00** |
| | FDR↓ | 0.24 | 0.82 | 0.23 | 0.82 | 0.79 | 0.91 | 0.25 | 0.12 | 0.82 | 0.94 | **0.00** |
| 100/4 | SHD↓ | 177.00 | 515.00 | 129.50 | - | 427.00 | 442.00 | 337.50 | 405.50 | 402.00 | 481.50 | **6.50** |
| | TPR↑ | 0.66 | 0.08 | 0.86 | - | 0.02 | 0.02 | 0.17 | 0.06 | 0.00 | 0.02 | **0.98** |
| | FDR↓ | 0.21 | 0.86 | 0.25 | - | 0.85 | 0.89 | 0.11 | 0.42 | 0.81 | 0.94 | **0.00** |
| 150/4 | SHD↓ | 267.50 | 825.67 | - | - | 641.00 | 636.50 | 503.50 | 458.00 | 589.50 | 802.25 | **20.75** |
| | TPR↑ | 0.66 | 0.03 | - | - | 0.01 | 0.01 | 0.16 | 0.27 | 0.00 | 0.02 | **0.96** |
| | FDR↓ | 0.21 | 0.94 | - | - | 0.91 | 0.92 | 0.08 | 0.17 | 0.50 | 0.97 | **0.00** |
| 1000/4 | SHD↓ | 1674.00 | - | - | - | - | 4134.50 | - | 3921.50 | - | 6600.80 | **704.00** |
| | TPR↑ | 0.69 | - | - | - | - | 0.02 | - | 0.02 | - | 0.02 | **0.82** |
| | FDR↓ | 0.18 | - | - | - | - | 0.72 | - | 0.14 | - | 0.98 | **0.00** |

Table 16: Average SHD, TPR, and FDR coming from 5 seeds. Each dataset comprises 5000 observations generated by ANMs with function $f_2$ and Gumbel variation. The causal graphs are generated by ER model with varying average degrees (e) and variable numbers (d). Bold and underlined font represent the first and second best results in terms of SHD, respectively.

| d/e | Metrics | CAM | SAM | DiffAN | CORL | NOTEARS | GOLEM | GraN-DAG | GAE | DAG-GNN | SDCD | OURS |
|---|---|---|---|---|---|---|---|---|---|---|---|---|
| 10/1 | SHD↓ | 5.25 | 22.50 | 4.75 | 9.50 | 10.00 | 18.50 | 1.75 | 5.00 | 9.75 | 13.00 | **0.75** |
| | TPR↑ | 0.53 | 0.17 | 0.61 | 0.19 | 0.11 | 0.17 | 0.94 | 0.53 | 0.06 | 0.14 | **1.00** |
| | FDR↓ | 0.48 | 0.94 | 0.38 | 0.54 | 0.52 | 0.87 | 0.12 | 0.11 | 0.62 | 0.89 | **0.07** |
| 20/1 | SHD↓ | 14.67 | 44.33 | 8.00 | 23.67 | 22.67 | 44.33 | 7.33 | 10.33 | 22.33 | 30.33 | **4.00** |
| | TPR↑ | 0.60 | 0.05 | 0.65 | 0.04 | 0.04 | 0.00 | 0.67 | 0.46 | 0.00 | 0.05 | **0.86** |
| | FDR↓ | 0.46 | 0.98 | 0.33 | 0.89 | 0.88 | 1.00 | 0.08 | 0.00 | 1.00 | 0.96 | **0.09** |
| 50/1 | SHD↓ | 31.40 | 65.00 | 11.60 | 56.50 | 57.40 | 112.40 | 31.60 | 35.20 | 56.00 | 69.80 | **2.40** |
| | TPR↑ | 0.58 | 0.06 | 0.87 | 0.08 | 0.02 | 0.01 | 0.41 | 0.28 | 0.02 | 0.05 | **0.99** |
| | FDR↓ | 0.45 | 0.90 | 0.19 | 0.76 | 0.89 | 0.99 | 0.13 | 0.00 | 0.93 | 0.95 | **0.04** |
| 100/1 | SHD↓ | 81.80 | 159.20 | 62.80 | - | 126.80 | 182.00 | 38.20 | 88.00 | 109.80 | 165.80 | **2.40** |
| | TPR↑ | 0.59 | 0.04 | 0.85 | - | 0.02 | 0.01 | 0.65 | 0.12 | 0.01 | 0.07 | **0.99** |
| | FDR↓ | 0.52 | 0.95 | 0.37 | - | 0.93 | 0.99 | 0.05 | 0.06 | 0.96 | 0.95 | **0.01** |
| 150/1 | SHD↓ | 142.20 | 249.25 | - | - | 190.80 | 242.80 | 69.60 | 140.40 | 165.40 | 239.40 | **11.80** |
| | TPR↑ | 0.53 | 0.02 | - | - | 0.03 | 0.01 | 0.57 | 0.07 | 0.01 | 0.08 | **0.97** |
| | FDR↓ | 0.57 | 0.97 | - | - | 0.90 | 0.99 | 0.06 | 0.11 | 0.96 | 0.94 | **0.05** |
| 1000/1 | SHD↓ | 1203.60 | - | - | - | - | 1191.50 | - | 895.50 | - | 1747.40 | **711.60** |
| | TPR↑ | 0.51 | - | - | - | - | 0.02 | - | 0.10 | - | 0.13 | **0.50** |
| | FDR↓ | 0.65 | - | - | - | - | 0.91 | - | 0.00 | - | 0.91 | **0.51** |
| 10/4 | SHD↓ | 15.33 | 23.00 | 15.00 | 21.67 | 23.00 | 23.67 | 10.00 | 8.00 | 23.67 | 28.67 | **2.33** |
| | TPR↑ | 0.50 | 0.36 | 0.58 | 0.19 | 0.12 | 0.25 | 0.75 | 0.67 | 0.10 | 0.10 | **0.97** |
| | FDR↓ | 0.44 | 0.67 | 0.43 | 0.37 | 0.47 | 0.51 | 0.16 | 0.02 | 0.43 | 0.90 | **0.06** |
| 20/4 | SHD↓ | 32.33 | 90.00 | 31.00 | 70.33 | 70.67 | 76.00 | 54.33 | 50.67 | 66.00 | 83.33 | **1.33** |
| | TPR↑ | 0.65 | 0.27 | 0.78 | 0.07 | 0.05 | 0.07 | 0.29 | 0.24 | 0.03 | 0.09 | **0.99** |
| | FDR↓ | 0.27 | 0.80 | 0.34 | 0.70 | 0.77 | 0.81 | 0.29 | 0.23 | 0.77 | 0.88 | **0.02** |
| 50/4 | SHD↓ | 133.00 | 258.00 | 105.75 | 201.00 | 207.00 | 226.00 | 178.50 | 162.75 | 190.00 | 247.50 | **0.75** |
| | TPR↑ | 0.52 | 0.09 | 0.83 | 0.02 | 0.03 | 0.03 | 0.04 | 0.27 | 0.01 | 0.05 | **1.00** |
| | FDR↓ | 0.38 | 0.88 | 0.36 | 0.84 | 0.85 | 0.90 | 0.22 | 0.19 | 0.90 | 0.92 | **0.00** |
| 100/4 | SHD↓ | 309.80 | 541.40 | 236.40 | - | 423.00 | 426.00 | 321.40 | 212.40 | 390.00 | 494.00 | **4.80** |
| | TPR↑ | 0.50 | 0.05 | 0.82 | - | 0.02 | 0.01 | 0.19 | 0.45 | 0.00 | 0.02 | **0.99** |
| | FDR↓ | 0.43 | 0.92 | 0.37 | - | 0.88 | 0.93 | 0.11 | 0.00 | 0.59 | 0.96 | **0.00** |
| 150/4 | SHD↓ | 477.20 | 834.75 | - | - | 652.80 | 637.40 | 501.40 | 418.60 | 584.20 | 832.20 | **18.60** |
| | TPR↑ | 0.50 | 0.04 | - | - | 0.01 | 0.01 | 0.16 | 0.28 | 0.00 | 0.03 | **0.97** |
| | FDR↓ | 0.43 | 0.94 | - | - | 0.92 | 0.95 | 0.09 | 0.00 | 0.20 | 0.96 | **0.00** |
| 1000/4 | SHD↓ | 3558.00 | - | - | - | 4414.00 | 4224.50 | - | 3975.00 | - | 6491.20 | **788.80** |
| | TPR↑ | 0.48 | - | - | - | 0.01 | 0.01 | - | 0.00 | - | 0.02 | **0.80** |
| | FDR↓ | 0.47 | - | - | - | 0.95 | 0.88 | - | 0.24 | - | 0.97 | **0.00** |

Table 17: Average SHD, TPR, and FDR coming from 5 seeds. Each dataset comprises 5000 observations generated by ANMs with function $f_2$ and Gumbel variation. The causal graphs are generated by SF model with varying average degrees (e) and variable numbers (d). Bold and underlined font represent the first and second best results in terms of SHD, respectively.

| Methods | Metrics | Gaussian | | | | Gumbel | | | |
|---------|---------|------|------|------|------|------|------|------|------|
| | | ER1 | SF1 | ER4 | SF4 | ER1 | SF1 | ER4 | SF4 |
| OURS | SHD↓ | **2294** | **1899** | **3063** | **5854** | **1182** | **2535** | **2504** | **5178** |
| | TPR↑ | **0.63** | **0.77** | **0.89** | **0.79** | **0.89** | **0.67** | **0.91** | **0.81** |
| | FDR↓ | **0.15** | **0.16** | **0.05** | **0.10** | **0.12** | **0.23** | **0.04** | **0.08** |
| SDCD | SHD↓ | 3692 | 5877 | 21697 | 22836 | 2579 | 4839 | 20529 | 21326 |
| | TPR↑ | 0.88 | 0.65 | 0.46 | 0.38 | 0.93 | 0.72 | 0.48 | 0.38 |
| | FDR↓ | 0.44 | 0.60 | 0.58 | 0.61 | 0.33 | 0.50 | 0.53 | 0.55 |

Table 18: Average SHD, TPR, and FDR coming from 5 seeds. Each dataset comprises 5000 observations generated by ANMs with function $f_1$ and different variation distributions. The causal graphs are generated by either ER or SF model with 5000 variables and varying edge numbers. ERi (SFi) means an ER (SF) graph whose edge number is $5000i$. Bold font represents the best results in terms of SHD.

| Methods | Metrics | Gaussian | | | | Gumbel | | | |
|---------|---------|------|------|------|------|------|------|------|------|
| | | ER1 | SF1 | ER4 | SF4 | ER1 | SF1 | ER4 | SF4 |
| OURS | SHD↓ | **1200** | **3840** | **2793** | **6693** | **747** | **4640** | **2736** | **6364** |
| | TPR↑ | **0.96** | **0.87** | **0.90** | **0.80** | **0.95** | **0.78** | **0.88** | **0.78** |
| | FDR↓ | **0.17** | **0.42** | **0.04** | **0.14** | **0.091** | **0.48** | **0.02** | **0.11** |
| SDCD | SHD↓ | 6418 | 7639 | 32776 | 33423 | 7616 | 8495 | 34797 | 35123 |
| | TPR↑ | 0.61 | 0.47 | 0.16 | 0.14 | 0.18 | 0.11 | 0.02 | 0.02 |
| | FDR↓ | 0.64 | 0.72 | 0.85 | 0.86 | 0.87 | 0.91 | 0.98 | 0.98 |

Table 19: Average SHD, TPR, and FDR coming from 5 seeds. Each dataset comprises 5000 observations generated by ANMs with function $f_2$ and different variation distributions. The causal graphs are generated by either ER or SF model with 5000 variables and varying edge numbers. ERi (SFi) means an ER (SF) graph whose edge number is $5000i$. Bold font represents the best results in terms of SHD.

outperforms all baselines in all cases. In conclusion, $D^3PM$ achieves a great balance between effectiveness and efficiency.

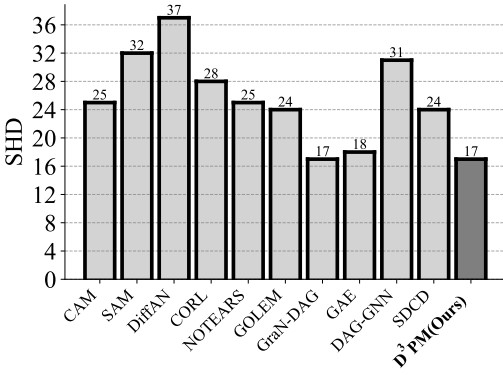

Figure 2: SHD on real-world gene expression dataset.

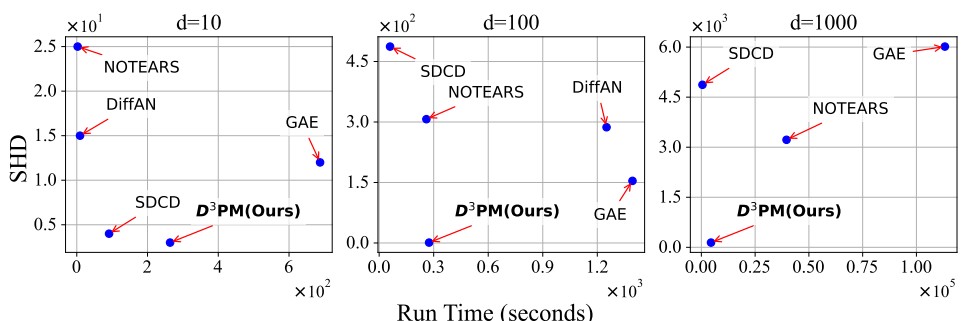

Figure 3: Training cost of $D^3PM$ and some baselines.