# OpenReview forum: "$D^3PM$: Diffusion Model Responds to the Duty Call from Causal Discovery"
_ICLR.cc/2025/Conference — ICLR 2025 Conference Withdrawn Submission_

### Official Review · Reviewer_ndsK · 2024-10-30

**Soundness:** 2
**Presentation:** 2
**Contribution:** 3
**Rating:** 3
**Confidence:** 3

**Summary:**

This paper introduces a new type of regularizer for differentiable causal discovery with additive noise models, called a "variation-negotiation regularizer," as an alternative to existing regularizers, e.g., an $\ell_1$ penalty. This regularizer estimates the noise of an additive noise model (referred to as a "variation") in two different ways, and regularizes by minimizing the estimated variations and the disagreement between estimates. From this, a loss function connected to diffusion models is derived. The diffusion model is applied to causal problems by (1) making the causal graph $A$ be a time-invariant variable to the diffusion model, a property the authors call "DAG-Invariance" and (2) regularizing with the NOTEARS constraint function. If the resulting graph is not a DAG at the end of optimization, it is iteratively pruned until it is a DAG. The proposed model is tested on synthetic datasets, showing very high performance in reconstructing the causal graph.

**Strengths:**

**The Proposed Approach is Interesting**
Many difficulties in differentiable causal discovery are optimization-based, and drawing connections with generative models to aid in optimization is interesting, and I find the proposed approach novel and creative. This general direction is exciting, and the results are promising. Ng et al. (2024) may be a relevant work to reference when adding to this discussion.

**Reported Results are Impressive**
The results reported in Figures 1(a) and 1(b) are extremely impressive, outperforming essentially the entire state of the art by several orders of magnitude. While I have some doubts about the extent of the experiments (outlined in Weaknesses), these results show considerable promise.

**Weaknesses:**

**The Presentation Can Be Improved**

I found the article somewhat difficult to read, and think rewriting to improve linearity and readability would significantly strengthen the paper.

To give a concrete example, the introduction of the variation-negotiation regularizer was somewhat disorienting to me. To arrive at the final regularization term, (1) a general "negotiation" regularizer is introduced, (2) a specific hyperparameter of the general regularizer is used, (3) additional terms are added to generalize the (simplified) regularizer, (4) the regularizer is algebraically manipulated, and yet another specific hyperparameter is chosen, and finally, (5) the regularizer is again generalized. I think it would significantly improve the clarity of the paper to start at, say, Equation (10), rather than Equations (8) and (9).

**Doubts About Experiments**

As mentioned in the strengths, the presented results are indeed extremely impressive. However, I don't find the functions tested ($f_1$ and $f_2$) particularly convincing. The article states that "In line with previous work (Sanchez et al., 2023; Wang et al., 2021; Lachapelle et al., 2020; Ng et al., 2019), we consider [...] two functions [...]". However, as far as I can tell, these functions are only used by Ng et al. (2019) -- at a quick glance, all other mentioned papers use data generated by Gaussian Processes (GPs).

I ran some crude experiments comparing to DAGMA (Bello et al., 2022) using a commonly chosen set of hyperparameters, using the provided code and finetuning as specified in Appendix C.1. On ER-1 graphs with 100 nodes and 1000 observations generated from GPs, DAGMA seemed to obtain similar or better performance than D3PM. In particular, to generate data:
```
n, d, s0, graph_type, sem_type = 1000, 100, 100, 'ER', 'gp'
true_DAG = simulate_dag(d, s0, graph_type)
data = simulate_nonlinear_sem(true_DAG, n, sem_type)
d = data.shape[1]
```
and to run DAGMA:
```
eq_model = DagmaMLP(dims=[d, 16, 1], bias=True, dtype=torch.double).to('cuda:0')
model = DagmaNonlinear(eq_model, dtype=torch.double)
W_est = model.fit(torch.from_numpy(data).to('cuda:0'), lambda1=2e-2, lambda2=5e-3,
	lr=2e-3, mu_factor=0.1, mu_init=0.1, w_threshold=0.2)
B_result = (np.abs(W_est) > 0.0).astype(int)
met = MetricsDAG(B_result, true_DAG)
print(met.metrics)
```
Of course, my experiments were quite brief, and it is possible I've made a mistake here. Nevertheless, I believe it would significantly improve the manuscript if more extensive tests with more common additive models are performed and reported.

**DAGness is Enforced in a Usual Way**

The paper repeatedly criticizes regularization which relies on a hypothesis about graphs, e.g., $\ell_1$ regularization, juxtaposed to the "data-driven" negotiation regularizer. However, the "DAGness" regularization in Section 3.2.4 is itself a hypothesis-based regularizer enforcing DAGness. To this end, I'm skeptical that the paper explores the "intrinsic relation between CD and diffusion models," as the actual connection to CD (juxtaposed to general graph learning) is fairly heuristic, especially with iterative pruning.

## Other Minor Notes
- (Equation 8, Line 368, and some others) Please use $\mathrm{tr}$ rather than $tr$. This is repeated in a few places.
- (Line 289) Should this be $X_t = \cdots$, rather than $X_0 = \cdots$?
- (Lines 857, 859) Please use $10^{-k}$ rather than $1e-k$.
- `requirements.txt` is not quite correct. It must use `==`, and `delu` should use `0.0.25`. Here is a corrected version.
```networkx==3.2.1
	numpy==1.23.4
	pandas==2.2.2
	scikit-learn==1.5.2
	scipy==1.13.1
	tomli==2.0.1
	gcastle==1.0.3
	delu==0.0.25
```
- It seems like some pieces of code are taken from other projects without credit. For example, `betas_for_alpha_bar(...)` and `get_named_beta_schedule(...)` in `gaussian_diffuser.py` are not credited, but are from OpenAI's ImprovedDiffusion repository. Please make sure to provide credit for reused code.
- Some tables in the appendix incorrectly bold D3PM as having the best performance. For example, in Table 2, NOTEARS has a lower FPR than D3PM, but is underlined rather than bolded.

**Questions:**

- All plots in Figures 1(a) and 1(b) show a common "dip" pattern, where performance improves then once again degrades. This is not visibly present in any of the baselines. Do the authors have any thoughts on why this may happen?
- How does the proposed method perform on other ANMs? For example, ones generated by random MLPs, GPs, or additive GPs? I would consider these to be the standard in the literature.

## References
Bello, K., Aragam, B., & Ravikumar, P. (2022). DAGMA: Learning DAGS via M-matrices and a log-determinant acyclicity characterization. _Advances in Neural Information Processing Systems_, _35_, 8226-8239.

Lachapelle, S., Brouillard, P., Deleu, T., & Lacoste-Julien, S. (2019). Gradient-based neural DAG learning. arXiv preprint arXiv:1906.02226.

Ng, I., Zhu, S., Chen, Z., & Fang, Z. (2019). A graph autoencoder approach to causal structure learning. arXiv preprint arXiv:1911.07420.

Ng, I., Huang, B., & Zhang, K. (2024). Structure learning with continuous optimization: A sober look and beyond. In Causal Learning and Reasoning (pp. 71-105). PMLR.

Sanchez, P., Liu, X., O'Neil, A. Q., & Tsaftaris, S. A. (2022). Diffusion models for causal discovery via topological ordering. arXiv preprint arXiv:2210.06201.

Wang, X., Du, Y., Zhu, S., Ke, L., Chen, Z., Hao, J., & Wang, J. (2021). Ordering-based causal discovery with reinforcement learning. arXiv preprint arXiv:2105.06631.

---

### Official Review · Reviewer_X8DA · 2024-11-02

**Soundness:** 2
**Presentation:** 2
**Contribution:** 1
**Rating:** 3
**Confidence:** 3

**Summary:**

This paper develops a variation-negotiation regularizer for causal discovery with continuous optimization. Different from traditional sparsity regularizer, the proposed regularizer exploits the variation variable in additive noise models to stabilize the solutions. The paper then discusses the connection with diffusion models, and accordingly proposes a diffusion model for causal discovery.

**Strengths:**

Diffusion models have gained considerable attention in the machine learning community, whose connection with causal discovery is interesting and remains under-explored.

**Weaknesses:**

- Many parts of the setting/methods/contributions are unclear.
- The functional causal model considered is rather restrictive.
- There is no discussion about the theoretical guarantee of the method, or what assumption is required to achieve identifiability.

**Questions:**

- In Eq. (1), the paper should give a proper definition of function $f$. Is it a element-wise function that is applied to each scalar?
- If the answer to the above question is "yes", then the causal model considered in Eq. (1) is rather restrictive, which is much more restrictive than the standard additive noise models, and is similar to a "generalized linear model".
- The paper mentions several times (e.g. in abstract, introduction, or Section 3) that one of its contributions is to "pose the solution to the minimization problem in Eq. (6) as a regularized optimization scheme". However, regularized optimization (that consists of data consistency and regularization) is commonly used in causal discovery, e.g., GOLEM, ENCO, NODAGS-Flow.
- Since least squares loss is used for data consistency part, does the paper requires the error terms $Z$ to have equal variances?
- After reading the explanations in L154-163, it is still unclear how Eq. (7) resolves the instability issue of Eq. (6), which the paper claims. It is also unclear why stability is an issue with the original optimization formulation in Eq. (6)--I would suggest the authors to provide more explanations.

---

### Official Review · Reviewer_oQgH · 2024-11-04

**Soundness:** 3
**Presentation:** 3
**Contribution:** 2
**Rating:** 1
**Confidence:** 4

**Summary:**

This paper addresses the causal discovery problem through a regularized optimization approach that incorporates a variation-negotiation regularizer. Building on the well-known NOTEARS method for DAG learning, the paper identifies a challenge with NOTEARS: it produces unstable solutions under high data perturbations. The author then demonstrates that the proposed variation-negotiation regularizer aligns with the objective of diffusion models, enhancing the robustness of the solution.

**Strengths:**

This paper addresses a significant issue in causal discovery, specifically within the context of continuous structural learning. It provides a thorough background and clear technical details, making the problem accessible and well-defined for readers. The objectives are well-formulated, guiding the study with a strong sense of direction and purpose. The paper includes extensive experiments, evaluating the proposed method on both simulated and real-world datasets, which adds credibility to the results and demonstrates the approach’s versatility. Additionally, the paper makes a valuable connection between causal learning and diffusion models, offering a novel perspective that could open avenues for future research in integrating causal discovery with diffusion-based techniques. This interdisciplinary link enhances the paper’s impact and showcases the broader relevance of the proposed method.

**Weaknesses:**

(1) The results would benefit from a stronger theoretical foundation. The formulation of causal discovery as a continuous optimization problem lacks sufficient justification, and there is no guarantee of the uniqueness or reliability of the discovered causal structure. While the paper aims to address the instability of the model under highly perturbed datasets, it does not provide a theoretical guarantee that the proposed variation-negotiation regularizer effectively mitigates this issue. Specifically, it remains unclear why the addition of this regularizer should be expected to stabilize the results. Although the paper includes a number of equations and models, the absence of a formal theorem or proof leaves the effectiveness of the method without solid theoretical support.

(2) The focus of the paper could be more concentrated. While the main objective appears to be addressing the instability of causal discovery algorithms, a substantial portion of the discussion is devoted to the connection with diffusion models. The relationship between diffusion modeling and the stated goal of enhancing stability could be more clearly articulated to show how this linkage supports the primary objective.

(3) In the experiments section, the design could be more directly aligned with the goal of addressing algorithm instability under data perturbations. It would be helpful to see experiments specifically structured to test the robustness of the method in scenarios with varying levels of data perturbation, as this would provide more targeted evidence on the stability improvements claimed in the paper.

**Questions:**

(1) How is the proposed method theoretically justified to address the instability problem? Could you clarify how “instability” is defined in this context, and why users should trust of your method for downstream tasks? Why is it reliable?

(2) How does the connection with diffusion models contribute to solving the instability problem?

(3) How do the experiments demonstrate that your method effectively addresses the instability problem? Could you elaborate on how the experimental design is aligned with this objective?

---

### Official Review · Reviewer_HhqC · 2024-11-04

**Soundness:** 3
**Presentation:** 4
**Contribution:** 3
**Rating:** 6
**Confidence:** 3

**Summary:**

The paper formulates a class of recent causal discovery methods as a regularized optimization scheme and propose a novel variation-negotiation regularizer. The regularizer does not rely on any hypotheses about true DAGs, which is important and useful to causal discovery in practice. Authors discover the connections between solving the regularized optimization problem and learning a diffusion model, and further develop an equivalent diffusion model called DAG-invariant Denoising Diffusion Probabilistic Model. Extensive empirical experiments validate the effectiveness of the proposed method.

**Strengths:**

- The proposed regularizer is pratically important and usefule to causal discovery.
- Interesting connection between the regularized optimizaiton problem and learning a diffusion model is obtained.
- The proposed method achieves good empirical performances.
- Clarity is good.

**Weaknesses:**

- No identifiability discussion.
- Not clear if the proposed method is performed as desired (see below).

**Questions:**

I generally like the paper where the proposed regularizer is pratically important and useful to causal discovery. In many real applications, I do find L_1 terms have a large effect on the final result, but there is no general rule to select its weight.

I have the following concerns regarding the current version:


- While the paper mentions the proposed regularizer does not depend on any general hypothesis about true DAG, I am wondering if there is any relation to existing regularizations, e.g., is it very indepedent of existing regularizations like L_1 penalty? Moreover, if such knowledge is available, can the method take advantages of this information? For example, in the following paper, authors show that a low rank assumption on the graph structure could enhance empricial performances for a class of problems with dense graphs. How can we modify the proposed method to take into account this low-rank assumption?

     [1] On Low-Rank Directed Acyclic Graphs and Causal Structure Learning, TNNLS 2024


- For the ANM model, there seems no constraints placed on the noise variables. Does it matter here?

- Importantly, I'm not sure if the proposed method works as desired. Take Eq. (9) as example, where the objective is $\||Z_X-Z_N\||^2$. Even if this  objective can be perfectly optimized, $Z_X$ may still not be the underlying true residual. That is, we can obtain $Z_X' = Z_X+ n$ and $Z_N' = Z_N + n$ for any $n$. In this sense, I am not sure if the proposed method has an identifiability/identification gurantee. Notice that identifiability is an important property to a causal discovery method.

I am willing to raise my score if these concerns are addressed, particularly the last one. Look forward to authors' response.

---

### Note · Authors · 2024-12-01

**Comment:**

We thank all 4 reviewers for the time they spent reviewing our work, especially Reviewer HhqC and Reviewer ndsK, who offered additional suggestions during the rebuttal stage. We will refine our work further based on your valuable comments.

**Withdrawal Confirmation:**

I have read and agree with the venue's withdrawal policy on behalf of myself and my co-authors.